# DAM: Towards A Foundation Model for Time Series Forecasting

**Luke Darlow, Qiwen Deng, Ahmed Hassan,**
**Martin Asenov, Rajkarn Singh, Artjom Joosen,**
**Adam Barker**[*]
Systems Infrastructure Research
Edinburgh Research Centre
Central Software Institute
Huawei
Edinburgh, UK
`sirlab@huawei.com`

**Amos Storkey**
School of Informatics
University of Edinburgh
Edinburgh, UK
`a.storkey@ed.ac.uk`

## Abstract

It is challenging to scale time series forecasting models such that they forecast accurately for multiple distinct domains and datasets, all with potentially different underlying collection procedures (e.g., sample resolution), patterns (e.g., periodicity), and prediction requirements (e.g., reconstruction vs. forecasting). We call this *general task* universal forecasting. Existing methods usually assume that input data is regularly sampled, and they forecast to pre-determined horizons, resulting in failure to generalise outside of the scope of their training. We propose the DAM – a neural model that takes randomly sampled histories and outputs an adjustable basis composition as a continuous function of time for forecasting to non-fixed horizons. It involves three key components: (1) a flexible approach for using randomly sampled histories from a long-tail distribution, that enables an efficient global perspective of the underlying temporal dynamics while retaining focus on the recent history; (2) a transformer backbone that is trained on these actively sampled histories to produce, as representational output, (3) the basis coefficients of a continuous function of time. We show that a *single univariate DAM*, trained on 25 time series datasets, either outperformed or closely matched existing SoTA models at multivariate long-term forecasting across 18 datasets, including 8 held-out for zero-shot transfer, even though these models were trained to specialise for each dataset-horizon combination. This single DAM excels at zero-shot transfer and very-long-term forecasting, performs well at imputation, is interpretable via basis function composition and attention, can be tuned for different inference-cost requirements, is robust to missing and irregularly sampled data by design.

## 1 Introduction

Time series forecasting can have a positive impact in a number of domains, including weather, traffic, finance, electricity, and cloud resource management (Wu et al., 2021; Lai et al., 2018). Most state-of-the-art (SoTA) forecasting methods assume fixed-length common-interval sequences (Nie et al., 2022; Zeng et al., 2023; Lim & Zohren, 2021), otherwise known as 'regular time series' (Rubanova et al., 2019). However, this does not scale well for many practical applications, particularly where the data generating mechanism is complex and varies over time. One example application is workload forecasting for cloud computing, where a single cloud provider can have tens or hundreds of thousands of time series workloads with diverse characteristics, differing length, and discontinuous data due to monitoring failures and outages (Sloss et al., 2019; Taylor & Letham, 2018; Darlow et al., 2023; Joosen et al., 2023). Predicting at this scale requires more generalised forecasting methods as it is infeasible to train or tune a model for each time series.

Existing methods fail to generalise outside the scope of their training. We argue that some of the key reasons for the ubiquitous poor generalisation in time series forecasting are that existing methods as-

---

[*]Also working at the School of Computer Science, University of St Andrews, UK.

sume (1) that input data is fixed-length and regularly sampled (i.e., evenly spaced and ordered), and (2) a pre-determined forecast horizon. It is common for existing methods to model future values directly as a vector output of fixed-length. Relaxing these assumptions enables *universal forecasting* – an approach to scaling forecasting methods such that they are applicable across domains and generalise to new datasets. Universal forecasting must be robust to the underlying collection processes of data (e.g., resolution or continuity) and cross-domain differences (e.g., seasonality or stationarity). In this paper, we aim to solve the challenge of designing a single model that can forecast accurately for a variety of time series datasets.

We present the **deep data-dependant approximate analytical model (DAM)** as a significant step towards a *foundation model* (Bommasani et al., 2021) for universal forecasting. The DAM is a neural model that takes in *randomly sampled histories* (Section 3.2) and outputs time-function forecasts through an adjustable basis decomposition (Section 3.3). It uses a transformer backbone (Section 3.1) to ingest time series that are irregularly sampled, and forecasts via a continuous function of time, meaning that it can be applied across diverse domains with differing forecast requirements.

To the best of our knowledge, the DAM is the first model for universal time series forecasting that can be trained simultaneously across many diverse datasets, of different resolutions, and for various horizons, such that it generalises well both within and outwith the training set. We trained it on 25 publicly available datasets, totalling 2280 univariate time series (over 44 mil. samples). A *single DAM* outperformed most specialised SoTA methods at long-term forecasting, was superior at very long-term forecasting and at imputation, and outperformed SoTA methods on held-out datasets **even when those methods were trained on those datasets**. Our contributions can be summarised as:

1. The design and implementation of the DAM (Section 3).

2. A new and flexible approach for using actively sampled histories (Section 3.2): a long-tail sampling method that enables efficient access to the distant past for a global perspective of the underlying signal, while maintaining focus on the recent past.

3. Forecasting via continuous basis functions, where the coefficients are the output of the DAM (Section 3.3). Such a function is not constrained by a pre-determined horizon, thus enabling longer term forecasts (Section 4.3) and past reconstruction (Section 4.4).

4. Demonstrations of: stable and performant forecasting in very long-term settings, flexible inference cost, transfer to held-out data, and interpretability (Section 5).

## 2 RELATED WORK

PatchTST (Nie et al., 2022) is a modern and performant transformer method that operates by encoding regularly sampled patches of channel-independent (i.e., univariate) time series data into tokens for attention. DLinear (Zeng et al., 2023) decomposes the time series signal into trend and seasonal components, applies linear layers to these, and sums the resultant forecast. N-HiTS (Challu et al., 2023) is an extension of NBEATS (Oreshkin et al., 2020), both of which effectively forecast by way of multi-scale neural basis composition. The *neural basis* that NBEATS and N-HiTS use is different from the basis functions that the DAM uses; *neural basis* are learned weighted connections between past and future as opposed to a continuous function of time.

**Multi-scale modelling.** TimesNet (Wu et al., 2023) and MICN (Wang et al., 2022) both use explicit multi-scale mechanisms for effective forecasting, breaking the task up into multiple scales over which convolutional neural networks can operate. Pyraformer (Liu et al., 2021) and Crossformer (Zhang & Yan, 2022) use custom hierarchical attention mechanisms for multi-scale modelling. Informer (Zhou et al., 2021) uses sparse attention to improve efficiency. LightTS (Zhang et al., 2022) applies careful down sampling strategies and MLP layers to model at multiple scales. The performance of explicit multi-scale methods is evidence that multi-scale modelling is paramount for accurate forecasting. The DAM also operates at multiple scales via its basis function composition, but can also access the distant past to model at longer scales because of the history sampling regime.

**Frequency-domain modelling.** Autoformer (Wu et al., 2021) uses an autocorrelation-based attention mechanism for forecasting. Zhou et al. (2022b) argued that frequency domain information enables a more global perspective of time series, proposing Fedformer that uses a 'frequency enhanced

block' and Fourier projections to act in the frequency domain. Zhou et al. (2022a) used Legrende polynomials and Fourier projections to model and denoise historical data for FiLM. ETSFormer (Woo et al., 2022) uses two attention mechanisms that utilise (1) exponential decay to explicitly bias toward recent history and (2) high amplitude fourier components. The DAM operates in both frequency and time domains simultaneously and can utilise distant history (Fourier projections are not suited to irregular data (Lomb, 1976)). More related work can be found in Appendix A.

## 3 THE DAM, EXPLAINED

The DAM is a model for universal forecasting. We designed it such that a single model can be used for many time series datasets and across domains. It uses a transformer to ingest context data sampled from a long-tail distribution, called the history sampling regime (HSR), and returns the coefficients of basis functions. These define the shape of a continuous function of time, $f(t)$. The DAM is trained to estimate this function from actively sampled *past data* for any past or future $t$.

### 3.1 BACKBONE

Figure 1: ① Context time-value samples from the HSR (Section 3.2) are sent to a ② linear solver to initialise the basis coefficients, $\theta_0$. These are ③ embedded into ④ B-tokens. Context data is also ⑤ embedded into ⑥ TV-tokens and processed through ⑦ 4 layers of MHSA, ToME, and feed-forward blocks, with layer-norm, and used as ⑧ keys and values for cross attention, where the queries are ⑨ the B-tokens. Both TV- and B-tokens are ⑩ passed to proceeding layers. The ⑪ B-tokens from the final layer are projected into ⑫ basis coefficients for ⑬ forecasting and backcasting.

Figure 1 shows the DAM architecture. $D_{model}$ is the latent dimension of the model. The DAM takes as input: (1) **univariate** time-value tuples sampled from the HSR (Section 3.2), with time units of days (i.e., $\delta t = 1$ is a 1 day interval), and (2) initialised basis coefficients with corresponding frequency information (Section 3.3). **T**ime-**V**alue pairs are embedded into what we call '**TV-tokens**' (using transformer nomenclature) and initialised **B**asis coefficients are embedded into '**B-tokens**'. Another TV-token is initialised with 50 evenly-spaced percentiles of the values for affine adjustments. Appendix B gives a code listing for the full model architecture.

**Model structure.** After embedding, the DAM uses 4 layers of processing, each consisting of 4 heads of multi-head self-attention (MHSA) (Vaswani et al., 2017) for TV-tokens; 4 heads of cross-attention for B-tokens where keys and values are TV-tokens; 3 separate feed-forward blocks (linear → GeLU (Hendrycks & Gimpel, 2016) → linear, across $D_{model}$) for TV-tokens, the affine token, and for B-tokens; an additional feed-forward block acting across B-tokens (FF$_{B,cross}$ in Figure 1); multiple layer normalization (LN) layers; and token merging (ToME) (Bolya et al., 2022). ToME is used to reduce the number of TV-tokens during each layer of processing. This backbone is simple compared to earlier methods: it uses standard MHSA, not a time series-specific variant (Wu et al., 2021; Zhou et al., 2021); data need not be regularly sampled to yield continuous 'blocks' (Nie et al., 2022); no explicit multi-scale mechanisms are required (Wu et al., 2021; 2023; Wang et al., 2022).

## 3.2 HISTORY SAMPLING REGIME: A NEW TREATMENT OF TIME

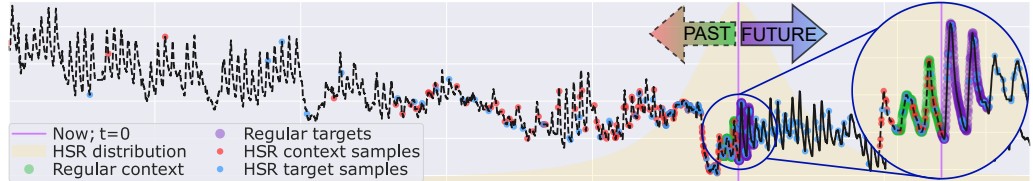

Figure 2: The HSR employed by the DAM, with the distribution in Equation 1 shown in yellow. Regularly sampled context and targets *of the same size as those from the HSR* are shown to demonstrate how the HSR enables a more global perspective while retaining focus close to 'now' ($t = 0$).

Most existing forecasting models were designed to process *regularly sampled data of fixed length*. We argue that this restriction is one of the main reasons reason for poor generalisation in time series forecasting. The DAM can ingest irregular and variable-length data, making it suitable to a broader variety of time series datasets and domains. Consequently, this means that the DAM can be trained easily on a large collection of time series datasets, thus mitigating early overfitting common in time series forecasting. The DAM uses a long-tail distribution over time steps, $x = t/R$, where $R$ is the sample resolution (e.g., hourly). We call this the *history sampling regime* (HSR):

$$p_{\text{hsr}}(x) = \frac{1}{c} \cdot \frac{1}{1 + \frac{x}{\sigma}^2}, \quad (1)$$

where $p_{\text{hsr}}(x)$ has the normalisation constant $c = \sum_{x \in \text{X}}(1 + \frac{x}{\sigma}^2)^{-1}$, and X is the sample support. $\sigma$ is the HSR 'width', where a smaller $\sigma$ biases sampling recent past more than distant past. The HSR is used to sample $x$, from which $(t, v)$ tuples are built, where $v$ is the value at time $t$. The HSR is used for both context data from the past ($x < 0$) and target data (any $x$). The number of points is variable and need not be the same for training and inference. Figure 2 gives an intuitive perspective of the HSR and demonstrates how the same-sized HSR-based context gives access to a more global perspective of the signal than the strictly regular context. This distribution was chosen because it increases the likelihood of sampling the distant past, enabling a global perspective of the signal[1], while ensuring the majority of samples are recent. We list additional advantages in Appendix C.

## 3.3 FORECASTING MECHANISM: BASIS FUNCTION COMPOSITION

The DAM forecasts using basis functions. We selected 437 frequencies from 1 minute ($\frac{1}{1440}$ days) to 10 years ($\pm 52 \cdot 7 \cdot 10$ days) by concatenating even samples in the minute, hour, day, week, and year ranges. These ranges were selected for wide basis function coverage (see Appendix D). The basis function composition that the DAM uses as a forecasting mechanism is:

$$f(t, \boldsymbol{\theta}, \boldsymbol{\nu}) = IQR \left( a \left( \sum_{\nu = \frac{1}{1440}}^{52 \cdot 7 \cdot 10} \theta_{\nu,1} \sin\left(2\pi\nu t\right) + \theta_{\nu,2} \cos\left(2\pi\nu t\right) \right) - b \right) + MED, \quad (2)$$

where $\boldsymbol{\theta}$ is the output vector from the DAM containing basis function coefficients, $\nu \in \boldsymbol{\nu}$ is the frequency. $\theta_{\nu,1}$ and $\theta_{\nu,2}$, are coefficients for sine and cosine functions at frequency $\nu$, which allows the DAM to smoothly capture all sinusoids between $\nu$ and $2\nu$ (Lomb, 1976). Median ($MED$) and inter-quartile range ($IQR$) are computed per-datum for online robust standardisation. $a$ and $b$ are affine adjustments also output from the DAM (Kim et al., 2021). Other methods also leverage basis functions (Oreshkin et al., 2020; Challu et al., 2023; 2022; Triebe et al., 2021), but these typically use some form of implicit basis (i.e., within the model structure) instead of an explicit composition. Our approach has two major advantages: (1) Equation 2 has no fixed horizon and can be assessed for any $t$ (see Section 4.3), and (2) basis functions are naturally interpretable (see Section 5.1).

**Basis function initialisation.** We found empirically that it was advantageous to initialise B-tokens with basis coefficients fit to the context. To this end, we use a linear differential equation solver to find initialisation coefficients, $\boldsymbol{\theta}_0$. Appendix E gives a code listing for this initialisation. Figure 3 shows how $\boldsymbol{\theta}_0$ is capable of representing the past sufficiently when coupled with the HSR, but fails at future extrapolation, meaning this initialisation strategy alone is insufficient for forecasting.

---

[1]We determined empirically that this distribution performed better than Gaussian and Uniform distributions.

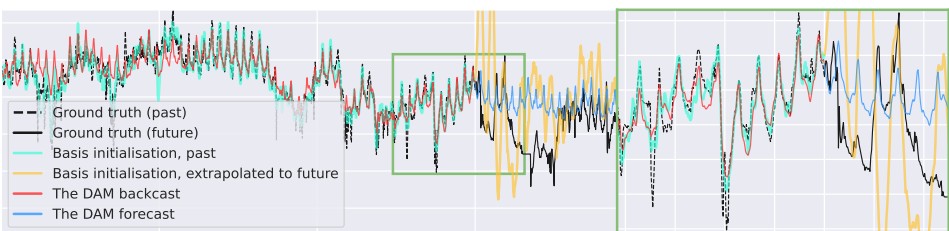

Figure 3: Basis function initialisation versus the DAM, showing past fit and future extrapolation.

## 3.4 TRAINING

We used the Huber loss for training (Huber, 1992), computed over targets sampled from the HSR that include both *past and future*. Thus, the DAM is trained to reconstruct and forecast. The number of context points sampled from the HSR for context and targets was set to 540. $\sigma$ was set to 720 during training. Before aggregation, the loss was re-scaled element-wise using an exponential decay of $2^{-x/360}$ (by 0.5 at time step 360 – empirically determined). We trained the single DAM used in this paper on 25 time series datasets, 10 of which are common benchmark datasets used for evaluation. Training commenced over $1,050,000$ iterations. Appendices H and F give more details.

## 3.5 INFERENCE PROCESS

During inference, the DAM needs only time-value pairs from a given variable, $v$, in order to predict for a given time, $t$, where $x$ are the time steps (effectively indices into the past). Inference involves: (1) using the HSR probability defined over $x$, sample indices using a weighted random choice without replacement, and extract time-value pairs from $v$ at the sampled indices; (2) compute $\theta_0$ from context, where $\theta_0$ is input to the model and **not one of the model parameters**; (3) forward pass to produce $\theta$; and (4) compute basis composition at $t$ or any other query times. Note that the DAM always operates in a univariate fashion (called 'channel independence' by recent works (Nie et al., 2022)) – all multivariate forecasts were generated by forecasting each variable separately.

**HSR tuning.** A significant advantage of using the HSR is that its settings (context size and $\sigma$) can be set *after training* for better forecasting. To this end, we estimated the mean squared error (MSE) per-dataset (on the validation split) for a range of context sizes and $\sigma$ values. Section 4.1 shows our results with and without this tuning while Appendix G has more details on the tuning.

## 4 EXPERIMENTS

We used a total of 33 datasets for training and evaluation. We augment 10 commonly used datasets (following e.g. Wu et al., 2021; Zhou et al., 2021; Liu et al., 2021) that we split into train/valid/test, with another 15 datasets that are additionally used to enhance training (details in Appendix H). The 10 datasets are used to test within-dataset generalisation (Section 4.1) and they are: ETTh1, h2, m1, and m2; ECL; Traffic; Weather; USWeather; Exchange; and Wind. In Section 4.2 we test outwith generalisation on 8 held-out datasets, namely: Illness, Weekdays, UCIPower, Azure, MTemp, MWeb, MWeather, and MMeters. In Section 4.3 we test the DAM on very-long-term forecasting, and in Section 4.4 we demonstrate how it can be used for imputation.

## 4.1 LONG-TERM TIME SERIES FORECASTING

**Results and discussion.** Table 1 gives the multivariate long-term forecasting results for the DAM against 6 SoTA methods (average of 3 seeds). DAM$_{HSR\text{-tuned}}$ denotes when we used optimal HSR values based on validation set performance (Section 3.5; Appendix G). Baselines were trained to specialise on dataset-horizon combinations as designed, meaning that each baseline required 40 unique variants, while we only trained one DAM. This setup provides a best-case scenario for these models' performance and represents the current SoTA gamut in forecasting. **The DAM achieves SoTA performance (1ˢᵗˢ) across 39 of 80 metrics** and is comparable to SoTA on others. The

closest competitor was PatchTST with 28 **1sts**. Normalised MSE and MAE results and results on an additional 8 SoTA methods can be found in Appendix I, with visualisations in Appendix J. We only **trained one DAM for all the results** in Table 1. The DAM was designed for generality, and while it has more data to train on compared to other methods because of this, forecasting across many dataset-horizon combinations is a more challenging task than specialisation.

Table 1: Long-term multivariate forecasting. Mean squared error (MSE) and mean absolute error (MAE) are shown. Context size was set to 720 for the DAM (ToME reduction to 333), and varied per dataset for DAM_HSR-tuned. **Gold**, silver, and **bronze** are 1st, 2nd, and 3rd place per metric-horizon-dataset combination. The DAM's placing is tallied simultaneously for both standard and HSR-tuned.

| | Horizon | DAM MSE | DAM MAE | DAM_HSR-tuned MSE | DAM_HSR-tuned MAE | PatchTST64 MSE | PatchTST64 MAE | DLinear MSE | DLinear MAE | N-HiTS MSE | N-HiTS MAE | Crossformer MSE | Crossformer MAE | Pyraformer MSE | Pyraformer MAE | MICN MSE | MICN MAE |
|---|---|---|---|---|---|---|---|---|---|---|---|---|---|---|---|---|---|
| ETTm1 | 96 | 0.309 | 0.358 | 0.308 | 0.356 | 0.303 | 0.353 | 0.301 | 0.344 | 0.352 | 0.381 | 0.373 | 0.416 | 0.620 | 0.529 | 0.319 | 0.370 |
| | 192 | 0.343 | 0.379 | 0.343 | 0.377 | 0.334 | 0.370 | 0.336 | 0.366 | 0.402 | 0.418 | 0.452 | 0.475 | 0.617 | 0.549 | 0.361 | 0.397 |
| | 336 | 0.351 | 0.384 | 0.351 | 0.382 | 0.364 | 0.390 | 0.370 | 0.387 | 0.443 | 0.447 | 0.685 | 0.628 | 0.749 | 0.630 | 0.405 | 0.431 |
| | 720 | 0.403 | 0.418 | 0.407 | 0.418 | 0.416 | 0.422 | 0.427 | 0.425 | 0.492 | 0.475 | 0.991 | 0.729 | 0.975 | 0.736 | 0.506 | 0.497 |
| ETTm2 | 96 | 0.173 | 0.252 | 0.170 | 0.251 | 0.166 | 0.256 | 0.169 | 0.262 | 0.219 | 0.299 | 0.342 | 0.412 | 0.354 | 0.441 | 0.185 | 0.282 |
| | 192 | 0.222 | 0.288 | 0.220 | 0.287 | 0.222 | 0.295 | 0.228 | 0.304 | 0.322 | 0.362 | 0.670 | 0.600 | 0.640 | 0.604 | 0.275 | 0.349 |
| | 336 | 0.234 | 0.297 | 0.232 | 0.297 | 0.274 | 0.330 | 0.303 | 0.361 | 0.415 | 0.424 | 1.050 | 0.715 | 1.280 | 0.867 | 0.385 | 0.424 |
| | 720 | 0.331 | 0.364 | 0.325 | 0.362 | 0.361 | 0.383 | 0.361 | 0.383 | 0.619 | 0.562 | 3.727 | 1.361 | 3.012 | 1.328 | 0.580 | 0.531 |
| ETTh1 | 96 | 0.373 | 0.404 | 0.367 | 0.401 | 0.372 | 0.401 | 0.379 | 0.401 | 0.427 | 0.431 | 0.397 | 0.422 | 0.721 | 0.641 | 0.408 | 0.426 |
| | 192 | 0.401 | 0.422 | 0.391 | 0.415 | 0.416 | 0.431 | 0.433 | 0.439 | 0.475 | 0.458 | 0.519 | 0.501 | 0.879 | 0.732 | 0.479 | 0.472 |
| | 336 | 0.409 | 0.427 | 0.396 | 0.419 | 0.432 | 0.444 | 0.445 | 0.447 | 0.555 | 0.506 | 0.671 | 0.608 | 1.019 | 0.796 | 0.570 | 0.539 |
| | 720 | 0.438 | 0.462 | 0.421 | 0.443 | 0.458 | 0.474 | 0.497 | 0.508 | 0.592 | 0.546 | 0.812 | 0.688 | 0.999 | 0.795 | 0.807 | 0.671 |
| ETTh2 | 96 | 0.300 | 0.336 | 0.280 | 0.330 | 0.276 | 0.339 | 0.291 | 0.356 | 0.425 | 0.438 | 0.746 | 0.608 | 1.521 | 0.953 | 0.374 | 0.415 |
| | 192 | 0.360 | 0.370 | 0.338 | 0.369 | 0.339 | 0.381 | 0.378 | 0.413 | 0.533 | 0.493 | 1.569 | 0.939 | 5.107 | 1.817 | 0.496 | 0.482 |
| | 336 | 0.369 | 0.375 | 0.346 | 0.376 | 0.364 | 0.403 | 0.451 | 0.463 | 0.572 | 0.523 | 2.360 | 1.240 | 4.652 | 1.824 | 0.620 | 0.552 |
| | 720 | 0.391 | 0.404 | 0.392 | 0.420 | 0.391 | 0.430 | 0.696 | 0.592 | 0.882 | 0.665 | 3.379 | 1.569 | 4.308 | 1.789 | 0.810 | 0.648 |
| ECL | 96 | 0.159 | 0.265 | 0.154 | 0.259 | 0.129 | 0.224 | 0.142 | 0.241 | 0.183 | 0.278 | 0.147 | 0.248 | 0.285 | 0.378 | 0.166 | 0.277 |
| | 192 | 0.176 | 0.280 | 0.171 | 0.274 | 0.148 | 0.242 | 0.156 | 0.254 | 0.189 | 0.286 | 0.162 | 0.262 | 0.298 | 0.393 | 0.177 | 0.287 |
| | 336 | 0.181 | 0.285 | 0.176 | 0.279 | 0.164 | 0.260 | 0.171 | 0.271 | 0.205 | 0.302 | 0.205 | 0.302 | 0.306 | 0.400 | 0.186 | 0.297 |
| | 720 | 0.242 | 0.337 | 0.237 | 0.331 | 0.200 | 0.292 | 0.206 | 0.304 | 0.242 | 0.336 | 0.253 | 0.338 | 0.305 | 0.393 | 0.207 | 0.316 |
| Traffic | 96 | 0.468 | 0.335 | 0.460 | 0.332 | 0.360 | 0.249 | 0.412 | 0.285 | 0.544 | 0.326 | 0.518 | 0.267 | 0.690 | 0.393 | 0.518 | 0.310 |
| | 192 | 0.481 | 0.342 | 0.474 | 0.339 | 0.380 | 0.257 | 0.425 | 0.291 | 0.550 | 0.328 | 0.555 | 0.285 | 0.675 | 0.380 | 0.536 | 0.317 |
| | 336 | 0.486 | 0.344 | 0.479 | 0.341 | 0.392 | 0.264 | 0.438 | 0.300 | 0.571 | 0.334 | 0.579 | 0.299 | 0.673 | 0.377 | 0.548 | 0.322 |
| | 720 | 0.547 | 0.381 | 0.538 | 0.376 | 0.447 | 0.305 | 0.468 | 0.318 | 0.611 | 0.351 | 0.663 | 0.358 | 0.710 | 0.397 | 0.572 | 0.332 |
| Weather | 96 | 0.156 | 0.203 | 0.154 | 0.203 | 0.148 | 0.198 | 0.174 | 0.235 | 0.157 | 0.215 | 0.172 | 0.240 | 0.196 | 0.284 | 0.193 | 0.250 |
| | 192 | 0.193 | 0.238 | 0.191 | 0.238 | 0.193 | 0.241 | 0.216 | 0.274 | 0.189 | 0.252 | 0.223 | 0.294 | 0.231 | 0.315 | 0.234 | 0.291 |
| | 336 | 0.205 | 0.248 | 0.203 | 0.247 | 0.244 | 0.282 | 0.261 | 0.311 | 0.227 | 0.289 | 0.277 | 0.341 | 0.296 | 0.364 | 0.287 | 0.336 |
| | 720 | 0.283 | 0.306 | 0.280 | 0.305 | 0.315 | 0.333 | 0.325 | 0.365 | 0.287 | 0.336 | 0.370 | 0.409 | 0.430 | 0.436 | 0.355 | 0.388 |
| Exchange | 96 | 0.087 | 0.211 | 0.090 | 0.216 | 0.093 | 0.216 | 0.089 | 0.214 | 0.145 | 0.273 | 0.271 | 0.377 | 0.673 | 0.661 | 0.093 | 0.227 |
| | 192 | 0.173 | 0.298 | 0.178 | 0.302 | 0.231 | 0.348 | 0.163 | 0.300 | 0.399 | 0.455 | 0.489 | 0.520 | 0.878 | 0.756 | 0.186 | 0.333 |
| | 336 | 0.204 | 0.325 | 0.208 | 0.327 | 0.352 | 0.435 | 0.317 | 0.426 | 0.560 | 0.558 | 0.960 | 0.761 | 1.108 | 0.863 | 0.320 | 0.443 |
| | 720 | 0.921 | 0.724 | 0.893 | 0.710 | 0.992 | 0.756 | 0.974 | 0.738 | 0.949 | 0.743 | 1.585 | 0.681 | 1.654 | 1.022 | 0.794 | 0.698 |
| Wind | 96 | 0.202 | 0.257 | 0.203 | 0.252 | 0.177 | 0.208 | 0.196 | 0.227 | 0.185 | 0.228 | 0.170 | 0.215 | 0.167 | 0.208 | 0.175 | 0.219 |
| | 192 | 0.213 | 0.264 | 0.217 | 0.261 | 0.191 | 0.220 | 0.215 | 0.241 | 0.199 | 0.240 | 0.188 | 0.227 | 0.185 | 0.220 | 0.192 | 0.230 |
| | 336 | 0.215 | 0.265 | 0.220 | 0.263 | 0.199 | 0.226 | 0.229 | 0.252 | 0.209 | 0.246 | 0.198 | 0.235 | 0.195 | 0.226 | 0.202 | 0.234 |
| | 720 | 0.230 | 0.282 | 0.247 | 0.287 | 0.206 | 0.233 | 0.232 | 0.274 | 0.252 | 0.263 | 0.210 | 0.238 | 0.214 | 0.246 | 0.220 | 0.246 |
| USWeather | 96 | 0.476 | 0.466 | 0.478 | 0.462 | 0.469 | 0.476 | 0.472 | 0.480 | 0.508 | 0.501 | 0.453 | 0.466 | 0.447 | 0.473 | 0.476 | 0.493 |
| | 192 | 0.520 | 0.497 | 0.527 | 0.495 | 0.531 | 0.520 | 0.518 | 0.515 | 0.558 | 0.541 | 0.498 | 0.502 | 0.512 | 0.517 | 0.524 | 0.522 |
| | 336 | 0.529 | 0.503 | 0.537 | 0.502 | 0.558 | 0.536 | 0.549 | 0.536 | 0.576 | 0.553 | 0.526 | 0.527 | 0.541 | 0.540 | 0.546 | 0.534 |
| | 720 | 0.590 | 0.544 | 0.611 | 0.549 | 0.629 | 0.576 | 0.615 | 0.579 | 0.613 | 0.579 | 0.571 | 0.556 | 0.570 | 0.559 | 0.549 | 0.540 |
| Average | | 0.336 | 0.354 | 0.330 | 0.351 | 0.331 | 0.350 | 0.352 | 0.368 | 0.420 | 0.406 | 0.725 | 0.521 | 0.992 | 0.646 | 0.396 | 0.398 |
| 1sts,2nds,3rds | | MSEs:16,17,17, MAEs: 23,20,6 | | | | 14,6,9 | 14,5,13 | 2,11,6 | 3,7,7 | 1,0,2 | 0,0,2 | 2,5,3 | 1,7,6 | 4,2,1 | 3,0,2 | 2,0,3 | 1,2,6 |

**Limitations.** It is common for other methods to use a low epoch count to mitigate overfitting (Wu et al., 2021; Zhou et al., 2022b; 2021), which is a sub-optimal strategy when training large neural networks. While the training cost is *relatively high* for the DAM when compared to other forecasting models, this may indicate that it can learn more complex and useful patterns from the data. Nevertheless, the DAM requires more training than specialised models. Datasets with sharp changes (spikes), such as traffic, electricity, and wind, are challenging for the DAM because representing them with basis functions is difficult. The DAM is univariate, and while modern methods advocate for 'channel independence' to mitigate overfitting (Nie et al., 2022), the DAM does not evidence early overfitting. This suggest cross-variable information could be useful but remains inaccessible.

## 4.2 FORECASTING ON HELD OUT DATASETS

A foundation model should transfer well within scope but outside of its training datasets. Model transfer is typically conducted using either **fine-tuning**—via short training on target datasets—or in **zero-shot** mode, without any additional training. We tested both protocols on 8 datasets held out during training, from the Monash time series repository (Godahewa et al., 2021), the UCI dataset repository (Frank, 2010), and Azure (Shahrad et al., 2020) (Appendix H.3). Table 2 lists transfer

performance of the DAM vs. PatchTST, Dlinear, and NHiTS baselines, including these baselines trained from scratch. These 3 methods are effectively univariate and can therefore be tested on zero-shot transfer.[2] We report zero-shot test performance of baseline model variants trained for Table 1 after selecting the best-performing models on validation sets. Thus, the results for these three baselines are those which are optimal across the 10 training datasets listed in Table 1.

Table 2: Results on held-out datasets averaged over 4 horizons. Zero-shot performance is shown for the DAM vs. 3 SoTA baselines, alongside fine-tuned (the DAM) and standard training (baselines) for comprehensive comparison. Overall best performance is shown in **bold**. * indicates average error (not validation-based selection) because the dataset was too short for the required context.

| | Zero-shot | | | | | | | | Fine-tuned | | Trained from scratch | | | | | |
| | DAM | | PatchTST | | DLinear | | NHiTS | | DAM | | PatchTST64 | | DLinear | | NHiTS | |
| | MSE | MAE | MSE | MAE | MSE | MAE | MSE | MAE | MSE | MAE | MSE | MAE | MSE | MAE | MSE | MAE |
|---|---|---|---|---|---|---|---|---|---|---|---|---|---|---|---|---|
| Illness | 1.964 | 0.920 | 4.266 | 1.497 | 4.173 | 1.460 | 5.195 | 1.570 | 2.019 | 0.925 | **1.884** | **0.906** | 2.357 | 1.094 | 2.338 | 1.041 |
| Weekdays | 0.942 | 0.573 | 1.502 | 0.736 | 1.135 | 0.672 | 1.221 | 0.773 | 0.947 | 0.578 | 0.898 | 0.559 | 0.786 | 0.556 | **0.582** | **0.418** |
| UCIPower | 1.595 | **0.514** | 1.590 | 0.545 | 1.615 | 0.570 | 1.697 | 0.625 | 1.623 | 0.522 | 1.501 | 0.524 | 1.528 | 0.552 | 1.520 | 0.538 |
| Azure | 1.531 | 0.640 | 1.630 | 0.667 | 1.444 | 0.649 | 1.552 | 0.667 | **1.312** | **0.510** | 1.489 | 0.558 | 1.618 | 0.649 | 1.441 | 0.604 |
| MTemp | 0.950 | 0.639 | 1.030* | 0.713* | 0.975 | 0.699 | 1.068 | 0.670 | 0.881 | 0.637 | 1.029 | 0.687 | 1.922 | 1.078 | **0.804** | **0.595** |
| MWeb | 12.633 | **0.586** | 12.656 | 0.700 | 13.321 | 0.732 | 12.847 | 0.678 | 12.512 | 0.603 | 12.831 | 0.632 | 14.559 | 0.927 | 13.920 | 0.726 |
| MWeather | 0.479 | 0.506 | 0.907 | 0.792 | 0.862 | 0.769 | 1.177 | 0.895 | 0.439 | 0.456 | **0.423** | 0.463 | 0.428 | **0.452** | 0.456 | 0.473 |
| MMeters | 0.836 | 0.499 | 0.895 | 0.546 | 0.940 | 0.566 | 1.057 | 0.657 | 0.849 | **0.482** | **0.794** | 0.498 | 0.807 | 0.513 | 0.852 | 0.536 |

**The DAM achieves SoTA zero-shot transfer on 14/16 metrics, and even outperforms baselines that were trained from scratch on some datasets**. The DAM generalises well outside of its training set. In some cases with very short datasets (e.g., MTemp) even baseline methods outperform their variants trained from scratch, showing how brittle performance can be when training on small datasets. Details of the fine-tuning process and standard training in Appendix K.

### 4.3    VERY LONG-TERM FORECASTING

Some scenarios demand very-long-term forecasting, owing to dataset resolution or user requirements. For example, forecasting to one week for the Weather dataset (with a resolution of 10 minutes) requires a horizon of 1008 steps, which is beyond what is typically considered 'long-term' in existing works (i.e., 720 time steps). Since the DAM is agnostic to the horizon, producing very-long-term forecasts requires no additional training; we simply evaluated the same DAM used throughout this paper. Figure 4 shows forecasts over 5000 steps ($\pm$35 days) for the weather dataset, comparing against PatchTST and DLinear trained on this horizon. Evidently, these baseline methods capture the strong daily periodicity and some attempt at a long-term trajectory but fail to produce meaningful or performant very-long-term forecasts. The results in the accompanying table show how the DAM performs well compared to SoTA methods, **even when the latter are trained for these horizons**.

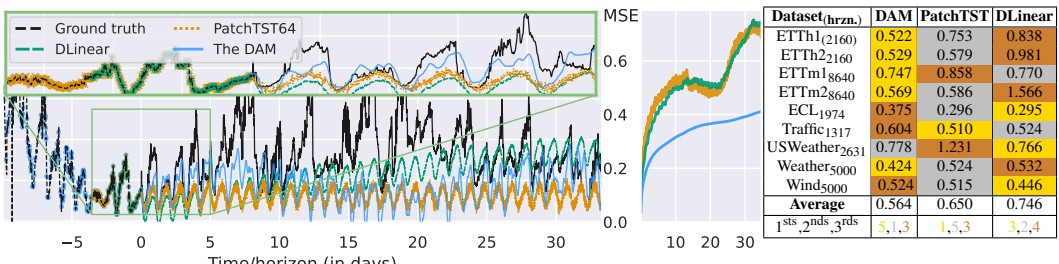

Figure 4: Very long-term forecasting. The 'OT' variable of Weather on the left and MSE versus horizon in the centre. The DAM produces better performing forecasts that also contain interesting multi-scale patterns, compared to baselines. To produce these figures the DAM context was set to 512, matching PatchTST. The inset shows from -512 to 720 steps. MSE vs. very-long horizon on 9 datasets is given in the table. Horizons were set according to 3/4 the length of the validation set.

Figure 4 also demonstrates how a regular context is limited when compared to a context built using the HSR, which enables a more global perspective **at the same cost** (512 samples in this case).

---

[2]They are also the top 3 baseline models in Table1, strengthening the findings of Nie et al. (2022) regarding univariate models being preferred over multivariate models, possibly owing to overfitting.

## 4.4 HELD OUT TASK: IMPUTATION

Time series imputation is important in cases where regularly sampled data is necessary. Upon release, TimesNet (Wu et al., 2023) evidenced SoTA performance on the imputation task. Table 3 shows results using only basis functions with $\theta_0$ coefficients. No training of the backbone is even required in this case because the initialisation coefficients are optimal for past data. Unlike $\theta_0$, the basis coefficients the DAM outputs, $\theta$, are better suited to forecasting and reconstruction than imputation.

Table 3: Average imputation MSEs for 12.5, 25, 37.5, and 50% missing data.

| Models | DAM | | TimesNet | |
|--------|-----|-----|----------|-----|
| Metric | MSE | MAE | MSE | MAE |
| ETTh1 | 0.0971 | 0.1324 | 0.1153 | 0.2336 |
| ETTh2 | 0.0742 | 0.1034 | 0.0673 | 0.1666 |
| ETTm1 | 0.0256 | 0.1000 | 0.0369 | 0.1207 |
| ETTm2 | 0.0185 | 0.0700 | 0.0274 | 0.0968 |
| ECL | 0.0854 | 0.1283 | 0.0946 | 0.2090 |
| Traffic | 0.3078 | 0.1924 | 0.4796 | 0.2695 |
| Weather | 0.0272 | 0.0463 | 0.0302 | 0.0529 |

In Table 3 we masked entire columns (i.e., all variables) per time step as this is a reasonable reflection of what may cause missing data and thus require imputation or reconstruction.[3]

## 5 ANALYSES AND DEMONSTRATIONS

### 5.1 INTERPRETABILITY

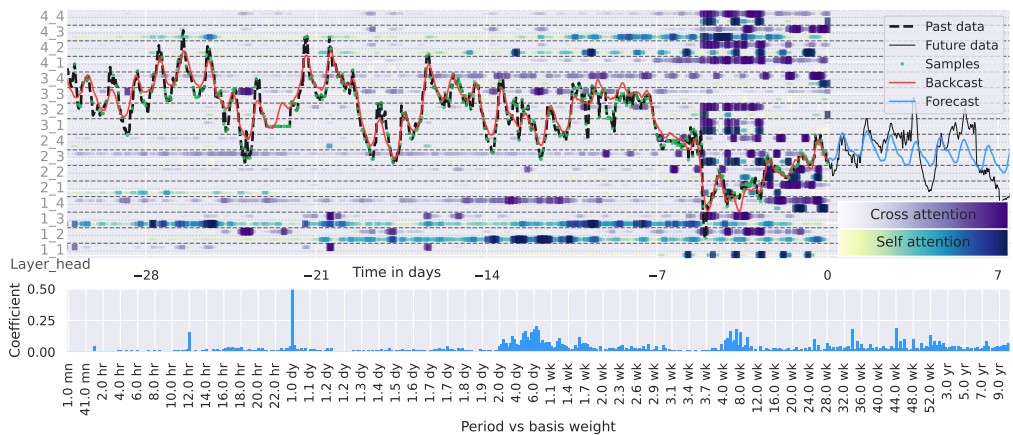

Figure 5: Attention analysis showing HSR samples, backcast, forecast, past and future data (ETTh1 test), and cumulative attentions per TV-token. The degree of attention paid is colour-coded and normalised for each attention head. Basis coefficients per period are also shown.

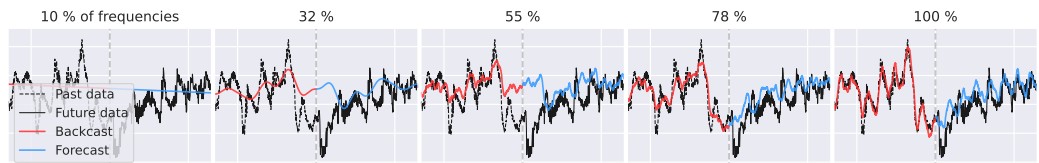

Figure 6: Low-to-high frequency basis composition on the 'OT' variable of ETTm1. From left to right, the percentage labels denote the percentiles of frequencies used to compose the forecast.

Figure 5 demonstrates the DAM's interpretability, showing self- and cross-attention weights, and basis coefficients for ETTh1. The cumulative attention for each layer is shown behind the data and forecast. Cumulative attention is the sum of the attention weights paid to individual TV-tokens. Each attention head is performing a different task in order to build the prediction. High coefficients correspond to the periodicity in this dataset (e.g., daily and weekly). The high coefficients for low frequencies are worth noting since this dataset spans less than 2 years. This evidences that the DAM

---

[3]The results for TimesNet are worse than those published by the authors. This is because their multivariate masking procedure was applied randomly over a 2D tensor instead of for all variables at sampled times.

is able to capture longer term trends (thanks to the HSR) even when only a small part of that trend presents itself in the data. Figure 6 shows how the DAM composes its prediction from low-to-high frequency components. Appendices M and L contain demonstrations on other datasets.

## 5.2 ARCHITECTURE ABLATION STUDY

Table 4 gives the results when ablating components of the DAM architecture (see Figure 1), namely: self-attention, cross-attention, feed-forward (TV), feed-forward (B), feed-forward in the coefficient dimension (B, cross), and ToME reduction. We ablated these components by skipping each one during the forward pass. The results suggest that each component is necessary, but that the feed-forward block acting across basis coefficients ($FF_{B,cross}$) is crucial. $FF_{B,cross}$ is the primary mechanism for updating B-tokens, with the attention components acting to share information and enhance predictive performance, hence the strong reliance on this component.

Table 4: Ablation study showing average results over 4 horizons for the ETT datasets. The first row denotes which component is removed. Performance drops are highlighted in **red gradient**.

|  | Nothing | | Self-Attn | | Cross-Attn | | $FF_{TV}$ | | $FF_B$ | | $FF_{B,cross}$ | | ToME | |
|---|---|---|---|---|---|---|---|---|---|---|---|---|---|---|
|  | MSE | MAE | MSE | MAE | MSE | MAE | MSE | MAE | MSE | MAE | MSE | MAE | MSE | MAE |
| ETTh1 | 0.405 | 0.404 | 0.439 | 0.469 | 0.606 | 0.539 | 0.483 | 0.493 | 0.599 | 0.561 | 31.099 | 2.797 | 0.423 | 0.478 |
| ETTh2 | 0.355 | 0.371 | 0.393 | 0.437 | 0.469 | 0.479 | 0.349 | 0.416 | 0.434 | 0.432 | 13.411 | 1.688 | 0.359 | 0.409 |
| ETTm1 | 0.352 | 0.385 | 0.411 | 0.439 | 0.642 | 0.528 | 1.354 | 0.498 | 0.706 | 0.531 | 51.323 | 3.519 | 0.364 | 0.425 |
| ETTm2 | 0.240 | 0.300 | 0.335 | 0.392 | 0.574 | 0.473 | 0.258 | 0.380 | 0.316 | 0.387 | 18.481 | 2.090 | 0.241 | 0.366 |

## 5.3 FLEXIBLE INFERENCE COST

Figure 7 shows the relationship between performance (measured as MSE) and inference cost (forecast time in milliseconds): MSE reduces approximately exponentially, while cost increases linearly. The DAM's flexibility to context size and forecast horizon make it easier to deploy at scale because it can be tuned according to inference cost-requirements. For instance, low-resource environments or edge devices could use a smaller context size. The *same model can be scaled up or down smoothly without loss of generality.*

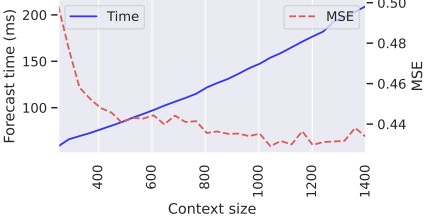

Figure 7: Inference cost versus performance on ETTh1 (minibatch size of 100).

## 6 CONCLUSION

We presented the DAM as a significant step towards a foundation model for universal forecasting. It uses a transformer backbone to ingest randomly sampled history data (from a long-tail distribution), and outputs a continuous function of time via basis function coefficients. The DAM overcomes the practical challenges associated with training a single model to forecast across datasets and domains, where differences in collection processes and data-generating mechanisms result in irregularly sampled data with significantly different patterns, seasonality, stationarity, and continuity. The DAM is flexible regarding data structure, thus enabling the use of a long-tail history sampling regime for an efficient global perspective of the time series signal. **All results in this paper were computed using a single univariate DAM**. This single DAM outperformed existing specialised SoTA methods on 39 of 80 dataset-horizon combinations, with the nearest competitor winning 28 of 80. Results on 8 held-out datasets show that the DAM performs well at zero-shot transfer, **even when compared against baseline models have been trained on the target datasets**. The DAM also excels at very-long-term forecasting, as demonstrated in a practical scenario for weather forecasting, and performs well at imputation. We also demonstrated how the DAM is interpretable via basis composition and attention and cost-flexible during inference. Future work will entail scaling up the model architecture and the training set in order to fully leverage the advantages of a this foundation model.

REPRODUCIBILITY STATEMENT

We included a simplified version of the PyTorch code for the DAM in Appendix B and for initialising basis coefficients in Appendix E. We also provided this as supplementary material for ease of use. Details on model and training hyper-parameters are given in Appendix F. A full working code repository will be released in the near future.

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

## A  ADDITIONAL LITERATURE

**Transformers and attention.**  Transformer-based architectures (Vaswani et al., 2017) have become the standard backbone for building state-of-the-art methods for time series forecasting. PatchTST (Nie et al., 2022) operates by encoding sequential patches of univariate time series data into tokens for the attention mechanism. This design choice leveraged the assumption that tokens require some local semantic information for efficient and performant modelling, resulting in some of the best forecasting performance to date. PatchTST also argued that univariate forecasting was preferred over multivariate forecasting, owing to rapid overfitting in the multivariate regime. Patching enabled a forward-pass speedup (fewer tokens) while also leveraging the advantages of processing sequences via parameterised projections (e.g., similar to what DLinear (Zeng et al., 2023) uses). In terms of forecasting performance, PatchTST is the strongest competitor to the DAM, particularly on datasets with sharp changes. However, PatchTST is limited regarding the structure of input data (e.g., continuity within patches) and forecast horizon (pre-determined), making it less well-suited than the DAM as a foundation model for universal forecasting. For future work, we would like to explore the use of patching for the DAM as this has clear performance benefits; we chose not to do so for this paper because it reduces the overall flexibility of a model to missing or discontinuous data.

Several recent works have made efforts to improve either efficiency or performance of transformers for time series, by way of modifying or extending the standard attention mechanism (Zhou et al., 2021; Wu et al., 2021; Liu et al., 2021; Zhou et al., 2022b). Sparse attention mechanisms, for example, reduce complexity for improved efficiency (Zhou et al., 2021; Kitaev et al., 2020). Informer (Zhou et al., 2021) used a KL-divergence-based ProbSparse self-attention mechanism to effectively reduce the number of comparisons made by the internal matrix multiplication behind attention. The ProbSparse structure biases recent history over distant history, much like the HSR of the DAM, but does so in a pre-determined and constrained fashion.

Zeng et al. (2023) questioned the use of transformers for time series forecasting. They proposed a simple, yet performant, linear method – DLinear. DLinear parameterised the connection between trend-seasonal decomposed context sequences and a vector over the forecast horizon, by way of a simple linear projection from input to output (and summation at all scales of the trend-seasonal decomposition). Their work brought into question whether complex and computationally-costly transformer architectures were at all necessary, given that the performance improvements they offer over the DLinear baseline are often marginal. Evidently, non-linear models (e.g., transformers) offer additional benefits over a linear projection because they can access non-linear temporal dynamics, particularly when reversible instance normalization (Kim et al., 2021) is employed. We argue in this paper that the full capability of transformers is not available in the small data regime followed in earlier works, and that universal forecasting is necessary to unlock the potential of transformers.

**Frequency domain** Modelling directly in the frequency domain affords a global perspective because low-frequency components describe long-term trends (Wu et al., 2021; Zhou et al., 2022b;a). Fedformer (Zhou et al., 2022b) used a 'frequency-enhanced' block and corresponding attention mechanism. This enhanced block uses a low-rank approximation to encourage learning long-range seasonality. Autoformer (Wu et al., 2021) used an autocorrelation-based attention mechanism to model multi-scale trends. The autocorrelation mechanism works by computing attention scores via autocorrelation and aggregates similar sub-series weighted according to these scores. Any method that uses a decomposition process requiring continuity (e.g., FFT) cannot work with irregular time series data (Rubanova et al., 2019). Furthermore, FFT and autocorrelation are both sensitive to noise because they are exact solutions. The frequency decomposition procedure for the DAM uses a least-squares approximation (Lomb, 1976), meaning that it can work with irregular randomly sampled histories (i.e., the HSR) and is robust to noisy data.

**Explicit multi-scale methods.** TimesNet (Wu et al., 2023) was designed to operate at multiple resolutions by way of reshaping 1D time series into 2D tensors such that the convolutional kernels it employs are able to span a multitude of periods in the data. This approach enables learning intra- and inter-period variations over a given input sequence. The importance of cross-periodic relationships is highlighted by the strong performance of TimesNet. Internally, TimesNet selects the top-k (with k=5 for long term forecasting) periods within the data using an FFT decomposition. MICN (Wang et al., 2022) uses multi-scale convolutions for short-term patterns, and isometric convolutions to capture longer-term pattners.. Internally, MICN uses a Seasonal Prediction Block to predict seasonal information and a Trend-cyclical Prediction Block to predict trend-cyclical information.

NBEATS (Oreshkin et al., 2020) used multiple blocks of dense connections between inputs and outputs to directly forecast over a pre-determined horizon. Following the 'classical' forecasting regime, three separate blocks were used to model trend, seasonality and residuals. Each block predicts basis expansion coefficients, both forward and backwards in time, to model the trend and seasonality, while residuals are modeled by a regular weighted connection. NBEATS produces explainable forecasts but the model scales poorly to long-sequences. N-HiTS (Challu et al., 2023) uses multi-rate data sampling and hierarchical interpolation to extend NBEATS, improving both performance and efficiency. This explicit use of multi-scale processing evidently improves performance because it forces a model to find patterns that extrapolate. The DAM models explicitly at multiple scales, but does not require a complex architecture or pre-processing of input data.

**Continuous functions.** Applying neural ordinary differential equations (ODEs) to irregularly sampled time series (Rubanova et al., 2019; Kidger et al., 2020; Morrill et al., 2021; De Brouwer et al., 2019) is a promising approach that may, in the future, be applicable to universal forecasting. Although the Neural ODE approach also models a continuous function of time, applying it to the universal forecasting task is still to be accomplished. Recent work has also shown how neural implicit representations (NIRs) (Sitzmann et al., 2020) can be learned for time series (Naour et al., 2023). The NIR approach is similar to the DAM basis composition but involves a non-linear neural network as output and is challenging to learn. We tested using a small neural network as an additional predictor (projecting its weights from its own B-token) for residuals, but found that the performance gain over basis functions was negligible. This is likely because (1) the basis functions are sufficient to capture the underlying temporal dynamics (see Figure 3) and (2) are easier to learn.

## B  DAM MODEL CODE

```python
import torch
import torch.nn as nn

class D3A2M(nn.Module):
    """
    The DAM wrapper to build the embedders, backbone, and collapsors.
    The forward method estimates the basis coefficients and the forecast
    method assess those for any given times.
    Args:
        d_model: latent width of model; default=256
        d_ff: width of hidden layer for each feed-forward block;
            default=256
```

```python
        n_layers: number of DAM layers; default=4 or 12
        n_tome: number of TV-tokens after iterative ToME; default=250
        n_heads: number of MHSA and cross attention heads; default=4
        dropout: default=0
    """
    def __init__(self, d_model, d_ff, n_layers, n_heads, n_tome,
                 dropout=0, base_frequency=86400):
        super(D3A2M, self).__init__()
        # The DAM's base frequency is 1 day (86400 seconds)
        self.register_buffer('time_scaling',
                             torch.Tensor([86400/base_frequency]))
        # Basis frequencies, shape [1, 1, 437]
        self.register_buffer('frequencies',
            torch.concatenate((1440/torch.arange(5, 61, 10),
                               24/(torch.arange(1, 48, 0.5)),
                               1/torch.arange(2, 28, 0.5),
                               1/torch.arange(28, 52*7, 7),
                               1/torch.arange(52*7, 52*7*10+1, 26*7))
                              ).unsqueeze(0).unsqueeze(0))
        # Embeddings for TV-tokens, B-tokens, and affine
        self.temporal_embedding = nn.Linear(self.frequencies.size(-1),
                                            d_model)
        self.value_embedding = nn.Linear(1, d_model)
        self.btoken_period_embedder = nn.Linear(2, d_model)
        self.btoken_coeffs_embedder = nn.Linear(2, d_model)
        self.affine_embedding = nn.Linear(50, d_model)
        # Backbone model
        self.backbone = TransformerBackbone(d_model=d_model,
                                            d_freq=437,
                                            d_ff=d_ff,
                                            n_layers=n_layers,
                                            n_tome=n_tome,
                                            n_heads=n_heads,
                                            dropout=dropout)
        # Collapse B-tokens into 2 coeffs each
        self.basis_collapsor = nn.Linear(d_model, 2)
        # Affine collapse into offset and scale
        self.affine_collapser = nn.Linear(d_model, 2)

    def forecast(self, times):
        """
        The actual forecasting method (after the DAM forward pass)
        args:
            times: forecast times (past or future) in seconds
        """
        pred = 0  # Set this up for the forecast
        times = times.unsqueeze(-1)/self.time_scaling  #
        cos_coeff = self.basis_coeff[:,:,0].unsqueeze(1)
        sin_coeff = self.basis_coeff[:,:,1].unsqueeze(1)
        sins = sin_coeff * torch.sin(2*torch.pi*self.frequencies*times)
        coss = cos_coeff * torch.cos(2*torch.pi*self.frequencies*times)
        pred += sins.sum(-1)
        pred += coss.sum(-1)
        # Reverse the robust standardisation and apply affine params:
        med = self.normalisation_params[:,:,0]
        iqr = self.normalisation_params[:,:,1]
        affine_offset = self.normalisation_params[:,:,2]
        affine_scale = self.normalisation_params[:,:,3]
        pred = iqr * ((pred - affine_offset)/affine_scale) + med
        return pred

    def forward(self, times, values):
        """
        The DAM wrapper expects times in seconds, where 0 is 'now',
        the future is positive time and the past is negative time.
```

```python
        Values are always univariate.
        args:
            times: time in seconds; shape=[M, C]
            values: values at times; shape=[M, C]
        M: minibatch size
        C: context size (e.g., 540)
        """
        times = times/self.time_scaling  # Convert time to
        with torch.no_grad():
            # Robust standardisation, unlike RevInv
            med = torch.median(values, axis=1, keepdim=True)[0]
            iqr = torch.quantile(values, q=0.75, axis=1, keepdim=True)-\
                torch.quantile(values, q=0.25, axis=1, keepdim=True)
            iqr[iqr<1e-6] = 1e-6
            values = (values - med)/iqr
            sin_coeffs, cos_coeffs = init_theta(times,
                                                values,
                                                self.frequencies)
        # Embed times for TV-tokens at the scales of the frequencies
        temporal_embedding = self.temporal_embedding(
            torch.sin(2*torch.pi*self.frequencies * times.unsqueeze(-1)))
        value_embedding = self.value_embedding(values.unsqueeze(-1))
        tv_tokens = value_embedding + temporal_embedding
        # Embed frequencies as periods similarly
        periods_embed = self.btoken_period_embedder(
            torch.concatenate((
                torch.sin(2*torch.pi/self.frequencies).transpose(1,2),
                torch.cos(2*torch.pi/self.frequencies).transpose(1,2)),
                -1))
        # arcsinh damps absolutely high coeffs
        b_tokens = self.btoken_coeffs_embedder(
            torch.arcsinh(0.1*\
                torch.stack((cos_coeffs, sin_coeffs), -1))/0.1)\
                + periods_embed
        # Quantiles for affine token
        quantiles = torch.quantile(values,
                        q=torch.linspace(0., 1, 50+2)[1:-1], dim=1)
        affine_token = self.affine_embedding(quantiles.T.unsqueeze(1))
        # Backbone
        b_tokens_out, affine_token_out = self.backbone(tv_tokens,
                                                       b_tokens,
                                                       affine_token)

        # Collapse into usable function space
        affine_collapsed = self.affine_collapser(affine_token_out)
        # Set coeffs and normalisation params
        # as attributes for forecasting step (hereafterr)
        self.normalisation_params = torch.concatenate(
                                (med.unsqueeze(-1),
                                 iqr.unsqueeze(-1),
                                 affine_collapsed), -1)
        self.basis_coeffs = self.basis_collapsor(b_tokens_out)

class TransformerBackbone(nn.Module):
    r"""
    Backbone for the DAM, taking in embedded TV-tokens, B-tokens,
    and affine token.
    Args:
        d_model: width of model; default=256
        d_freq: number of frequencies and B-tokens; default=437
        d_ff: width of hidden layer for each feed-forward block;
                default=256
        n_layers: number of DAM layers; default=4 or 12
        n_tome: number of TV-tokens after iterative ToME; default=250
        n_heads: number of MHSA and cross attention heads; default=4
        dropout: default=0
```

```python
    """
    def __init__(self,
                 d_model,
                 d_freq,
                 d_ff,
                 n_layers,
                 n_tome,
                 n_heads,
                 dropout=0,
                 ):
        super(TransformerBackbone, self).__init__()
        self.n_layers = n_layers
        self.n_tome = n_tome
        self.encoder_layers = nn.ModuleList([
            DAMLayer(d_model, d_freq, d_ff, n_heads, dropout) \
                                        for _ in range(n_layers)
        ])
    def forward(self, tv_tokens, p_tokens, affine_token):
        r"""
        Logic in this wrapper includes making sure ToME reduces by
        the right amount with each layer.
        Args:
        """
        n_tv_tokens = tv_tokens.size(1)
        # Reduce TV-tokens iteratively reduce at each layer using ToME
        r = int(round((n_tv_tokens-self.n_tome)/self.n_layers))
        for li in range(self.n_layers):
            if li == self.n_layers - 1:
                r = (p_tokens.size(1)) - self.n_tome  # rounding
            tv_tokens, p_tokens, affine_token = \
                self.encoder_layers[li](tv_tokens,
                                        p_tokens,
                                        affine_token,
                                        r)
        return p_tokens, affine_token

class DAMLayer(nn.Module):
    def __init__(self,
                 d_model,
                 d_freq,
                 d_ff,
                 n_heads,
                 dropout=0,
                 ):
        super(DAMLayer, self).__init__()
        self.mhsa_tv = nn.MultiheadAttention(d_model, n_heads, dropout,
                                             batch_first=True,
                                             add_zero_attn=True)
        self.cross_attention = nn.MultiheadAttention(d_model, n_heads,
                                             dropout, batch_first=True,
                                             add_zero_attn=True)
        self.feed_forward_tv = nn.Sequential(nn.Linear(d_model, d_ff),
                                             nn.GELU(),
                                             nn.Linear(d_ff, d_model),
                                             nn.Dropout(dropout))
        self.feed_forward_b = nn.Sequential(nn.Linear(d_model, d_ff),
                                             nn.GELU(),
                                             nn.Linear(d_ff, d_model),
                                             nn.Dropout(dropout))
        self.feed_forward_b_cross = nn.Sequential(
                                             nn.Linear(d_freq, d_freq*2),
                                             nn.GELU(),
                                             nn.Linear(d_freq*2, d_freq),
                                             nn.Dropout(dropout))
        self.feed_forward_aff = nn.Sequential(nn.Linear(d_model, d_ff),
```

```python
                                          nn.GELU(),
                                          nn.Linear(d_ff, d_model),
                                          nn.Dropout(dropout))
        self.layernorm1 = nn.LayerNorm(d_model)
        self.layernorm2 = nn.LayerNorm(d_model)
        self.layernorm3 = nn.LayerNorm(d_model)
        self.layernorm4 = nn.LayerNorm(d_model)
        self.layernorm5 = nn.LayerNorm(d_model)
        self.layernorm6 = nn.LayerNorm(d_model)
        self.layernorm7 = nn.LayerNorm(d_model)

    def forward(self, tv_tokens, b_tokens, affine_token, r):
        r"""
        Concatenate TV-tokens and affine token (in additional_p_tokens)
        to compute attention, but split this up again for ToME
        Args:
            tv_tokens: Time-value tokens; shape=[M, Ntv, D]
            b_tokens: B-tokens for basis coeffs; shape=[M, Nb, D]
            affine_token: for affine adjustment; shape=[M, 1, D]
            r: ToME reduction amount; scalar.
        M: minibatch size
        D: model dimension
        """
        tokens = torch.concatenate((affine_token, tv_tokens), 1)
        attn_output_tv, _  = self.mhsa_tv(tokens,
                                          tokens,
                                          tokens)
        attn_output_affine = attn_output_tv[:,:affine_token.size(1)]
        attn_output_tv = attn_output_tv[:,affine_token.size(1):]
        # Bipartite soft matching from ToME
        # https://github.com/facebookresearch/ToMe
        merge_method, _ = bipartite_soft_matching(attn_output_tv, r)
        tv_tokens = merge_method(tv_tokens)  # Merged
        attn_output_tv = merge_method(attn_output_tv)  # Merged
        tv_tokens = self.layernorm1(tv_tokens + attn_output_tv)
        tv_tokens = self.layernorm2(
                        self.feed_forward_tv(tv_tokens)+tv_tokens)  # FF
        affine_token = self.layernorm3(affine_token + attn_output_affine)
        affine_token = self.layernorm4(
                        self.feed_forward_aff(affine_token)+\
                                affine_token)
        # Transfer information from TV-toekns into B-tokens
        kv_tokens = torch.concatenate((affine_token, tv_tokens), 1)
        cross_attn, _ = self.cross_attention(b_tokens,
                                             kv_tokens,
                                             kv_tokens)
        b_tokens = self.layernorm5(b_tokens + cross_attn)
        b_tokens = self.layernorm6(
                    self.feed_forward_b(b_tokens) + b_tokens)  # FF
        # Process ACROSS B-tokens
        b_tokens_T = b_tokens.transpose(1,2)
        b_tokens = self.layernorm7(
            self.feed_forward_b_cross(b_tokens_T).transpose(1, 2)\
                                + b_tokens)
        return tv_tokens, b_tokens, affine_token

if __name__=='__main__':
    dam = D3A2M(256, 256, 4, 4, 250)
    # Dummy data and times would be sampled from
    # A long tail distribution and are not necessarily
    # sequential.
    dummy_values = torch.zeros((32, 540)).float()  # shape: [M, C]
    # -10 until before 'now':
    dummy_times = torch.zeros((32, 540)).random_(-864000, -1).float()
    # Estimate the basis coeffs
```

```
dam(dummy_times, dummy_values)
# Times can be past or future
dummy_forecast_times = torch.zeros((32, 540)).random_(-864000,
        864000).float()  # anywhen
forecast = dam.forecast(dummy_forecast_times)
assert forecast.shape == dummy_forecast_times.shape
```

Listing 1: Working PyTorch (Paszke et al., 2019) code for the DAM architecture.

## C  ADDITIONAL HSR ADVANTAGES

1. Data from far in the past is available with no additional cost because intermediate data can be ignored.

2. A HSR can be chosen such that the recent history has a higher likelihood of being sampled, thus weighing the importance of the recent past higher than the distant past. This is intuitively similar to the way humans perceive time.

3. Longer periods and trends are made available at no additional cost – earlier research Wu et al. (2023); Challu et al. (2023) capitalise on multi-periodicity in time series data, but these approaches are limited by the short-range perspective offered by a fixed-length sequence.

4. The HSR is not fixed during inference and can be use-case specific – altering the shape or type of distribution does not change the model structure and does not necessarily require retraining.

5. Missing data can be dealt with elegantly because those regions can simply be ignored in the HSR.

6. Taking multiple samples for a single forecast enables a notion of sample uncertainty.

7. The non-i.i.d. nature of time series datasets is alleviated by the randomness of the HSR.

### C.1  THE IMPACT OF THE HSR DURING INFERENCE

Varying the HSR involves changing either the number of context points or the width of the distribution from which they are sampled (see Section 3.2). Figure 8 shows the effect that context size and $\sigma$ have on the shape of the distribution defined by the HSR and the consequential forecasts. A small $\sigma$ (e.g., 100) results in samples most similar to fixed-length contexts because we sample without replacement.

## D  BASIS CHOICE

We pre-selected the basis function frequencies to span a range of resolutions that are common in time series datasets. The periods (for readability) we selected are:

- Minutes: [1, 6, 11, 16, 21, 25, 31, 36, 41, 45, 50, 55]

- Hours: [1.0, 1.2, 1.5, 1.7, 2.0, 2.2, 2.5, 2.7, 3.0, 3.2, 3.5, 3.7, 4.0, 4.2, 4.5, 4.8, 5.0, 5.2, 5.5, 5.8, 6.0, 6.3, 6.5, 6.7, 7.0, 7.3, 7.5, 7.8, 8.0, 8.2, 8.5, 8.7, 9.0, 9.2, 9.5, 9.7, 10.0, 10.3, 10.5, 10.7, 11.0, 11.2, 11.5, 11.8, 12.0, 12.2, 12.5, 12.7, 13.0, 13.2, 13.5, 13.7, 14.0, 14.2, 14.5, 14.8, 15.0, 15.3, 15.5, 15.7, 16.0, 16.2, 16.5, 16.8, 17.0, 17.2, 17.5, 17.8, 18.0, 18.3, 18.5, 18.8, 19.0, 19.3, 19.5, 19.7, 20.0, 20.2, 20.5, 20.8, 21.0, 21.2, 21.5, 21.8, 22.0, 22.3, 22.5, 22.7, 23.0, 23.3, 23.5, 23.7]

- Days: [1.00, 1.01, 1.02, 1.03, 1.04, 1.05, 1.06, 1.07, 1.08, 1.09, 1.10, 1.11, 1.12, 1.14, 1.15, 1.16, 1.17, 1.18, 1.19, 1.20, 1.21, 1.22, 1.23, 1.24, 1.25, 1.26, 1.27, 1.28, 1.29, 1.30, 1.31, 1.32, 1.33, 1.34, 1.35, 1.36, 1.37, 1.39, 1.40, 1.41, 1.42, 1.43, 1.44, 1.45, 1.46, 1.47, 1.48, 1.49, 1.50, 1.51, 1.52, 1.53, 1.54, 1.55, 1.56, 1.57, 1.58, 1.59, 1.60, 1.61, 1.62, 1.64, 1.65, 1.66, 1.67, 1.68, 1.69, 1.70, 1.71, 1.72, 1.73, 1.74, 1.75, 1.76, 1.77, 1.78, 1.79, 1.80, 1.81, 1.82, 1.83, 1.84, 1.85, 1.86, 1.87, 1.89, 1.90, 1.91, 1.92, 1.93, 1.94, 1.95, 1.96, 1.97, 1.98, 1.99, 2.00, 2.25, 2.50, 2.75, 3.00, 3.25, 3.50, 3.75, 4.00, 4.25, 4.50, 4.75, 5.00, 5.25, 5.50, 5.75, 6.00, 6.25, 6.50, 6.75, 7.00]

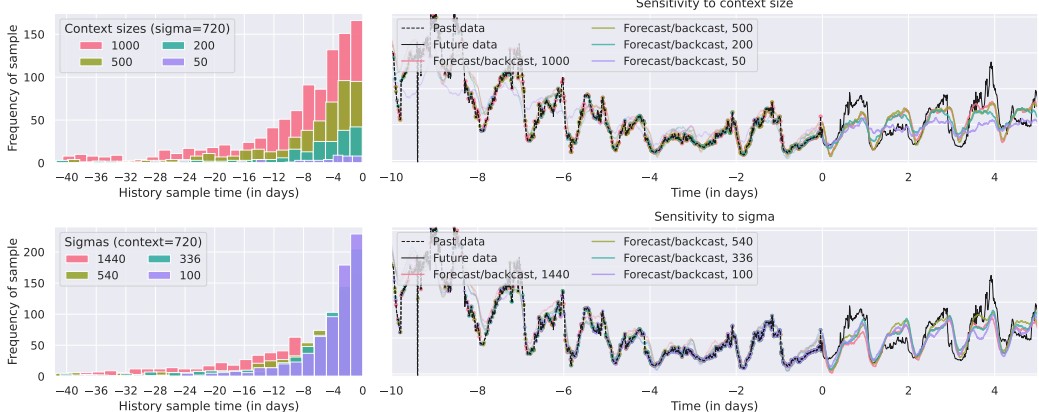

Figure 8: The effect of varying the history sampling regime (HSR). The top row demonstrates variation on the context size (i.e., the number of samples) on the forecast. The bottom row demonstrates variation on sigma (i.e., the width of the distribution from which samples are drawn). Note that sampling is done *without replacement*. The forecasts are shown for 720 sequence points, and 1440 points for the backcast.

- Weeks: [1.04, 1.07, 1.11, 1.14, 1.18, 1.21, 1.25, 1.29, 1.32, 1.36, 1.39, 1.43, 1.46, 1.50, 1.54, 1.57, 1.61, 1.64, 1.68, 1.71, 1.75, 1.79, 1.82, 1.86, 1.89, 1.93, 1.96, 2.00, 2.04, 2.07, 2.11, 2.14, 2.18, 2.21, 2.25, 2.29, 2.32, 2.36, 2.39, 2.43, 2.46, 2.50, 2.54, 2.57, 2.61, 2.64, 2.68, 2.71, 2.75, 2.79, 2.82, 2.86, 2.89, 2.93, 2.96, 3.00, 3.04, 3.07, 3.11, 3.14, 3.18, 3.21, 3.25, 3.29, 3.32, 3.36, 3.39, 3.43, 3.46, 3.50, 3.54, 3.57, 3.61, 3.64, 3.68, 3.71, 3.75, 3.79, 3.82, 3.86, 3.89, 3.93, 3.96, 4.00, 4.50, 5.00, 5.50, 6.00, 6.50, 7.00, 7.50, 8.00, 8.50, 9.00, 9.50, 10.00, 10.50, 11.00, 11.50, 12.00, 12.50, 13.00, 13.50, 14.00, 14.50, 15.00, 15.50, 16.00, 16.50, 17.00, 17.50, 18.00, 18.50, 19.00, 19.50, 20.00, 20.50, 21.00, 21.50, 22.00, 22.50, 23.00, 23.50, 24.00, 24.50, 25.00, 25.50, 26.00, 26.50, 27.00, 27.50, 28.00, 28.50, 29.00, 29.50, 30.00, 30.50, 31.00, 31.50, 32.00, 32.50, 33.00, 33.50, 34.00, 34.50, 35.00, 35.50, 36.00, 36.50, 37.00, 37.50, 38.00, 38.50, 39.00, 39.50, 40.00, 40.50, 41.00, 41.50, 42.00, 42.50, 43.00, 43.50, 44.00, 44.50, 45.00, 45.50, 46.00, 46.50, 47.00, 47.50, 48.00, 48.50, 49.00, 49.50, 50.00, 50.50, 51.00, 51.50, 52.00]

- Years: [1.25, 1.50, 1.75, 2.00, 2.25, 2.50, 2.75, 3.00, 3.25, 3.50, 3.75, 4.00, 4.25, 4.50, 4.75, 5.00, 5.25, 5.50, 5.75, 6.00, 6.25, 6.50, 6.75, 7.00, 7.25, 7.50, 7.75, 8.00, 8.25, 8.50, 8.75, 9.00, 9.25, 9.50, 9.75, 10.00]

We found during initial experimentation that the DAM was relatively robust to the choice of frequencies and the number thereof. We choose the above 437 frequencies because: (1) fewer B-tokens enables faster learning, (2) inspection of the initial basis weight fits, $\theta_0$, for all 25 training datasets revealed sufficient reconstruction of the signal (see Figure 3 and Appendix E), and (3) these frequencies cover ranges common in time series datasets because their sources (e.g., weather or traffic) are aligned with human time elements (minutes, hours,, etc.).

# E    BASIS COEFFICIENT INITIALISATION

Listing 2 is a PyTorch Paszke et al. (2019) listing of the code used to initialise the basis coefficients, $\theta_0$, for processing by the DAM.

# F    HYPER-PARAMETERS

Section 3.1 details the DAM architecture and Appendix B is a full PyTorch code listing of the model architecture. We included both for completeness and refer to nomenclature from these below.

```python
import torch
def init_theta(t, v, nu, lambda=1):
    # t has shape [B, L, 1]; times
    # v has shape [B, L, 1]; values
    # nu has shape [1, 1, 437]; pre-selected frequencies
    # lambda is for regularisation
    X = torch.concatenate((
            torch.sin(2*torch.pi*nu*t),  # broadcast to shape [B, L, 437]
            torch.cos(2*torch.pi*nu*t)),
            -1)  # shape [B, L, 437]
    I = torch.eye(X.size(-1))  # shape [437, 437]
    I[0,0] = 0
    reg = torch.repeat_interleave(
            lambda*I.unsqueeze(0),
            repeats=t.size(0), dim=0)
    theta = torch.linalg.solve(
            torch.bmm(X.transpose(1,2), X) + reg,
            torch.bmm(X.transpose(1,2), values).sum(-1))
    return theta  # shape [B, 437*2]
```

Listing 2: Pytorch (Paszke et al., 2019) code for $\theta_0$ initialisation.

**Model hyper parameters.** The following model hyper-parameters were used for the DAM in this paper:

- Model width, $d_{\text{model}}$, of 256.
- Feed-forward internal width, $d_{ff}$, of 256.
- 4 layers.
- 4 MHSA and cross-attention heads.
- A ToME reduction target of 250 TV-tokens.
- Dropout of 0.1.
- Time units of 1 day, such that $\delta t = 1$ denotes one day from 'now' and $\delta t = -1$ is one day into the past.

**Training hyper parameters.** The following training hyper-parameters were used for the DAM in this paper:

- Minibatch size of 32.
- Data sampling with replacement, where the probability of sampling a given univariate is proportional to how 'useful' it is. We define a proxy for utility here to be a combination of the standard deviation over an average day (if the sample resolution is sufficient, otherwise an average week) and the average standard deviation of all daily points, taking the average day as the vector mean. The former ensures that we do not over-sample very noisy data, while the latter captures intrinsic variation from the commonly periodic data. This choice was determined experimentally to be optimal when compared to sampling regimes balanced according to dataset length or the number of variables. Within any chosen univariate, samples are taken uniformly.
- Two phase learning: (a) 1,000,000 iterations with 10,000 warmup steps followed by cosine annealing from $1^{-3}$ to $1^{-14}$; and (b) an additional 50,000 iterations with 2,000 warmup steps followed by cosine annealing from $1^{-3}$ to 0.
- Gradient clipping to the 90[th] percentile of the latest 1000 gradients of all model weights.
- HSR context size and $\sigma$ of 540 and 720, respectively.
- 540 target points (also sampled from the HSR) over which to compute the loss.
- Random seed of 42 (the answer).

## G   HSR TUNING

Figure 9 shows 4 heatmaps for MSE estimates on the validation sets of the 10 datasets we tested on in Section 4.1. The differences in optimal HSR parameters across datasets is both insightful regarding the datasets themselves, and the differences amongst them, and how the DAM is adaptable to these differences. Note that no training was undertaken to optimise these values and the same model can be used with any combination of context size and $\sigma$. In each case we retained a ToME reduction ratio as per training (see Appendix F).

The optimal HSR settings for each dataset are:

- ETTm1: context size of 900, $\sigma = 100$, ToME reduction to 417.
- ETTm2: context size of 1200, $\sigma = 100$, ToME reduction to 556.
- ETTh1: context size of 1300, $\sigma = 100$, ToME reduction to 602.
- ETTh2: context size of 400, $\sigma = 100$, ToME reduction to 185.
- ECL: context size of 900, $\sigma = 100$, ToME reduction to 417 .
- Traffic: context size of 1300, $\sigma = 300$, ToME reduction to 602.
- Weather: context size of 1400, $\sigma = 200$, ToME reduction to 648.
- Exchange: context size of 1100, $\sigma = 100$, ToME reduction to 509.
- Wind: context size of 900, $\sigma = 100$, ToME reduction to 417.
- USWeather: context size of 1200, $\sigma = 100$, ToME reduction to 556.

## H   DATASETS' DETAILS

### H.1   DESCRIPTIONS AND SUMMARIES

| Dataset | Res. | Horizons | Train/valid/test | Num | Domain |
|---|---|---|---|---|---|
| ETTm1, ETTm2 | 15 mins | [96,192,336,720] | [34465, 11521, 11521] | 7 | Electricity |
| ETTh1, ETTh2 | 1 hour | [96,192,336,720] | [8545, 2881, 2881] | 7 | Electricity |
| ECL | 1 hour | [96,192,336,720] | [18317, 2633, 5261] | 321 | Electricity |
| Traffic | 1 hour | [96,192,336,720] | [12185, 1757, 3509] | 862 | Traffic |
| Weather | 10 mins | [96,192,336,720] | [36792, 5271, 10540] | 21 | Weather |
| Exchange | 1 day | [96,192,336,720] | [5120, 665, 1422] | 8 | Finance |
| Wind | 1 hour | [96,192,336,720] | [183982, 26299, 52594] | 28 | Weather |
| USWeather | 1 hour | [96,192,336,720] | [24544, 23229, 46455] | 12 | Weather |

Table 5: Dataset details. 'Res.' is the dataset resolution, or sampling rate. The listed horizons are those that are commonly tested on. 'Num' lists the number of variables (i.e., columns) in each dataset.

Table 5 contains information about the 10 public datasets used for both training and evaluation. Appendix H.2 details an additional 15 datasets used only for training. The ETT datasets[4] track electrical transformer statistics (e.g., load capacity and oil temperature). ECL [5] tracks electricity consumption for 321 clients, from 2012 to 2014, and is recorded in kilowatts. Traffic [6] tracks road occumpancy rates for 862 sensors on San Francisco Bay area freeways. It is a collection of 48 months (2015-2016) worth of hourly data from the California Department of Transportation. Exchange [7] tracks daily exchange rates of 8 currencies (Australian Dollar, Pound Sterling, Canadian Dollar, Swiss Franc, Chinese Yuan, Japanese Yen, New Zealand Dollar, and Singapore Dollar) versus the US Dollar. It spans 26 years, from 1990 to 2016.

---

[4] https://github.com/zhouhaoyi/ETDataset
[5] https://archive.ics.uci.edu/ml/datasets/ElectricityLoadDiagrams20112014
[6] http://pems.dot.ca.gov
[7] https://github.com/laiguokun/multivariate-time-series-data

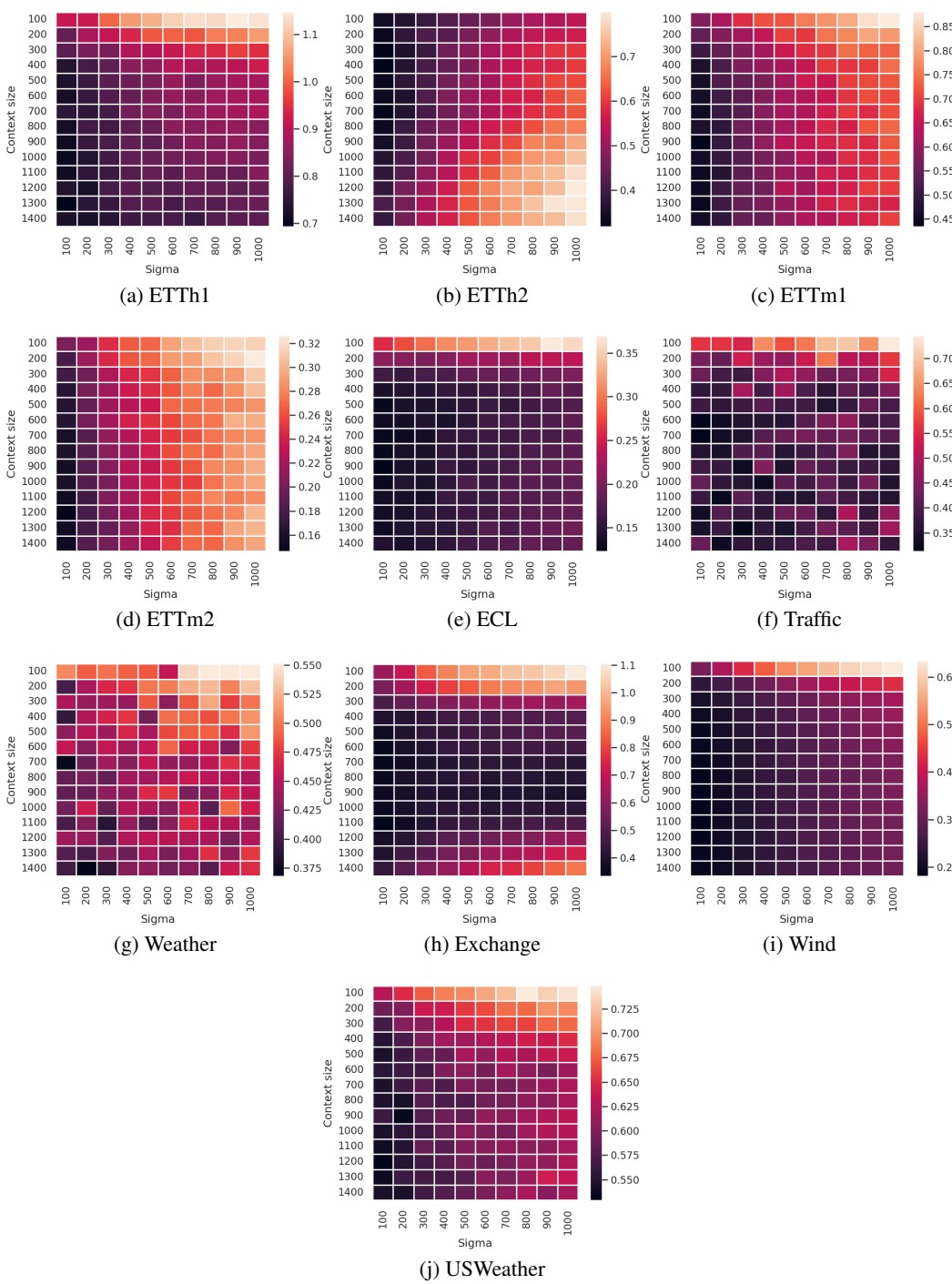

Figure 9: HSR parameter search for 10 commonly used benchmark datasets. The heatmaps show the MSE values for the corresponding context size and $\sigma$ combination

Weather [8] is the widely used Germany weather dataset, while USWeather[9] is rarely test on, most notably for Informer (Zhou et al., 2021). Wind [10] contains hourly power generation estimates 28 area's energy potential from 1986 to 2015 as a percentage of the power plants maximum output; it is also rarely tested on, most notably used for PyraformerLiu et al. (2021).

These datasets were all split according to prior work (Wu et al., 2021; Zhou et al., 2021; Liu et al., 2021).

## H.2 Additional datasets for training

We sourced an additional 15 datasets for training, aside from the 10 commonly used benchmarking datasets discussed above. Since the DAM is a universal forecaster that is not constrained to a single dataset or characteristic (e.g., resolution), it should benefit from additional training data. Many of the datasets listed below contain missing data, making it challenging for models that expect regularly sampled data. For instance, stock prices are not recorded on the weekend, resulting in persistent missing data. We did not evaluate the DAM on these additional datasets because there are typically no existing comparison points in the literature.

We used four sources for additional data, namely the Monash time series repository (Godahewa et al., 2021),[11] the UCI machine learning repository (Kelly et al., 2007),[12] the Huawei public and private cloud data release (Joosen et al., 2023),[13] and Kaggle[14] for stock market data. Table6 contains information about the additional 15 publicly datasets used to augment training in this paper.

| Dataset | Res. | Train/valid/test | Num | Missing | Domain |
|---|---|---|---|---|---|
| Monash Australian electricity | 30 mins | [139363, 46454, 46454] | 6 | Yes | Electricity |
| Monash Oikolab Weather | 1 hour | [6034, 2011, 2011] | 8 | No | Weather |
| Monash Solar | 10 mins | [31356, 10512, 10512] | 137 | No | Weather |
| Monash Sunspot | 1 day | [44354, 14784, 14784] | 1 | Yes | Weather |
| UCI Beijing air quality | 1 hour | [21038, 7012, 7012] | 132 | Yes | Air quality |
| UCI PM data | 1 hour | [31550, 10517, 10517] | 41 | Yes | Air quality |
| UCI Air Quality | 1 hour | [5612, 1872, 1872] | 13 | Yes | Air quality |
| UCI Tetouan City power | 10 mins | [31449, 10483, 10483] | 8 | Yes | Electricity |
| UCI Metro Traffic Volume | 1 hour | [28922, 9641, 9641] | 1 | Yes | Traffic |
| StocksForbes2000 | 1 day | [8013, 2671 , 2671] | 146 | Yes | Finance |
| StocksSP500 | 1 day | [8013, 2671 , 2671] | 154 | Yes | Finance |
| StocksNASDAQ | 1 day | [8013, 2671 , 2671] | 111 | Yes | Finance |
| StocksNYSE | 1 day | [8013, 2671 , 2671] | 206 | Yes | Finance |
| HuaweiPubM | 1 min | [22463, 7489, 7489] | 14 | No | Cloud |
| HuaweiPvtM | 1 min | [121823, 40608, 40609] | 35 | Yes | Cloud |

Table 6: Additional training dataset details. 'Res.' is the dataset resolution, or sampling rate. The 'Missing' column denotes whether the dataset contains any missing data.

The Monash time series repository (Godahewa et al., 2021) contains 30 datasets in total, from which we selected the following 4 based on their lengths: Australian electricity, Oikolab Weather, Solar, and Sunspot. Australian electricity demand is a dataset curated by the Monash team. The Oikolab[15] weather dataset is a public weather dataset extracted by the Monash team from a domain-specific platform. The Solar dataset contains simulated power output of photovoltaic power plants;[16] it is also used by other works (Lai et al., 2018; Liu et al., 2022). Sunspot was sourced from the sunspot index and long-term solar observations platform[17].

---

[8]https://www.bgc-jena.mpg.de/wetter/
[9]https://www.ncei.noaa.gov/data/local-climatological-data/
[10]https://www.kaggle.com/datasets/sohier/30-years-of-european-wind-generation
[11]https://forecastingdata.org/
[12]https://archive.ics.uci.edu/datasets
[13]https://github.com/sir-lab/data-release
[14]https://www.kaggle.com/datasets/paultimothymooney/stock-market-data
[15]https://oikolab.com/
[16]https://www.nrel.gov/grid/solar-power-data.html
[17]https://www.sidc.be/SILSO/newdataset

We included an additional 5 datasets from the UCI dataset repository: PM Data, Air Quality, Beijing Air Quality, Tetouan City power, Metro Traffic Volume. PM Data (Chen, 2017b) records hourly data for PM2.5 (particles that are 2.5 microns or less in diameter) and meteorological variables for 5 cities, namely Beijing, Shanghai, Guangzhou, Chengdu and Shenyang. Air Quality (Vito, 2016) contains the hourly responses of a gas multisensor device deployed on the field in an Italian city. Beijing Air Quality (Chen, 2017a) contains PM2.5 data of the US Embassy in Beijing along with meteorological data from Beijing Capital International Airport. Tetouan City power (Salam & El Hibaoui, 2023) includes the power consumption of three different distribution networks of Tetouan city in Morocco. Metro Traffic Volume (Hogue, 2019) includes the hourly traffic volume westbound on the I-94 in Minneapolis-St Paul, USA.

The stocks data on Kaggle contains data for Forbes2000, NASDAQ, NYSE, and the SP500. We filtered the stocks, keeping only those that had at least 10000 days of data (ignoring missing weekend data, however).

Huawei recently released 2 cloud resource datasets for their private and public cloud services (Joosen et al., 2023), spanning over 7 months at per-minute and per-second sample resolutions. The HuaweiPvtM dataset is derived from Huawei's internal workloads and contains detailed per-minute (aggregated from per-second) statistics for 200 functions running across multiple Huawei cloud data centers. We selected the function request data, keeping only those functions whose $10^{th}$ percentile was greater than 30 requests (35 functions). We applied this filtering because infrequently requested functions do not present interesting data for training. The HuaweiPubM dataset is from Huawei's public function-as-a-service platform, containing per-minute data. Following a similar justification, we employed the same filtering procedure, but kept only those functions whose $10^{th}$ percentile was greater than 100 requests (14 functions).

## H.3 HELD-OUT DATASETS

| Dataset | Res. | Train/valid/test | Num | Missing | Domain | Horizons | Time unit scaling for the DAM held-out inference |
|---|---|---|---|---|---|---|---|
| Illness | 1 week | [676, 97, 193] | 7 | No | Medicine | 1 year | |
| Monash London Smart Meter (MMeters) | 30 mins | [26880, 2840, 8680] | 100 | No | Electricity | 24,36,48,60 | 1 day |
| Monash Weather (MWeather) | 1 day | [7579, 1083, 2166] | 679 | No | Weather | 96,192,336,720 | 1 year |
| Monash Web traffic (MWeb) | 1 day | [562, 81, 160] | 500 | No | Web | 24,36,48,60 | 1 week |
| Monash Temperature Rain (MTemp) | 1 day | [507, 73, 146] | 1266 | Yes | Weather | 24,36,48,60 | 12 hours |
| Azure 2019 | 1 minute | [12096, 4032, 4032] | 159 | No | Cloud | 96,192,336,720 | 10 mins |
| UCI Power | 1 hour | [24211, 3459, 6918] | 6 | Yes | Electricity | 96,192,336,720 | 1 day |
| Wikipedia weekdays | 1 day | [1789, 597, 1194] | 7 | No | Web | 96,192,336,540 | 1 week |

Table 7: Additional training dataset details. 'Res.' is the dataset resolution, or sampling rate. The 'Missing' column denotes whether the dataset contains any missing data. For comparison to baseline methods we filled all missing values with zeros prior to dataset normalisation. Time unit scaling (i.e., multiplier on time for better frequency coverage) was determined empirically by inspection on the respective validation sets.

Table 7 lists details of the 8 datasets we used for held-out evaluations in Section 4.2. Illness [18] captures weekly patient data from Centers for Disease Control and Prevention in the United States, between 2002 and 2021. The values are the ratio of patients in certain age groups versus the total number of patients. Columns were filtered and splits determined according to earlier work (Wu et al., 2021; Zhou et al., 2021; Liu et al., 2021).

[18]https://gis.cdc.gov/grasp/fluview/fluportaldashboard.html

We selected four datasets from the Monash time series forecasting repository, namely: (1) London Smart Meters (MMeters) (Kaggle, 2023), (2) Weather (MWeather) (Sparks et al., 2017), Web traffic (MWeb) (Google, 2017), and Temperature Rain (MTemp) (Governmen, 2017). Note that MWeather is distinct from Oikolab Weather.

For MMeters we selected the top 100 longest variables (of 5560) and, owing to slight differences in start and end collection times, truncated each variable to the latest and earliest start and end times, respectively. This yielded a half-hourly dataset spanning from 10 October 2011 to 16 February 2014.

For MWeather we selected 679 of 3010 variables of the same length (10828). For MTemp we extracted the minimum, mean, and maximum temperatures from the 422 weather stations in the original time dataset. For MWeb we selected the 500 longest (considering missing data) of 145063 available time series, and filled remaining missing data with zeros (for training of baseline methods).

Similarly to the Huawei datasets, we processed the Azure 2019 function trace (Shahrad et al., 2020) by selecting the top 159 function requests by removing any functions whose 10th percentile was more than 70 requests. For the UCI household power consumption dataset (UCIPower) (Hebrail & Berard, 2012), we aggregated the minutely samples to per-hour and filled missing values with zeros. We constructed Wikipedia weekdays using the Wikipedia pageviews[19] tool, extracting page views for all 7 weekdays (Monday to Sunday) for 2983 days.

## H.4 MULTIVARIATE CORRELATIONS

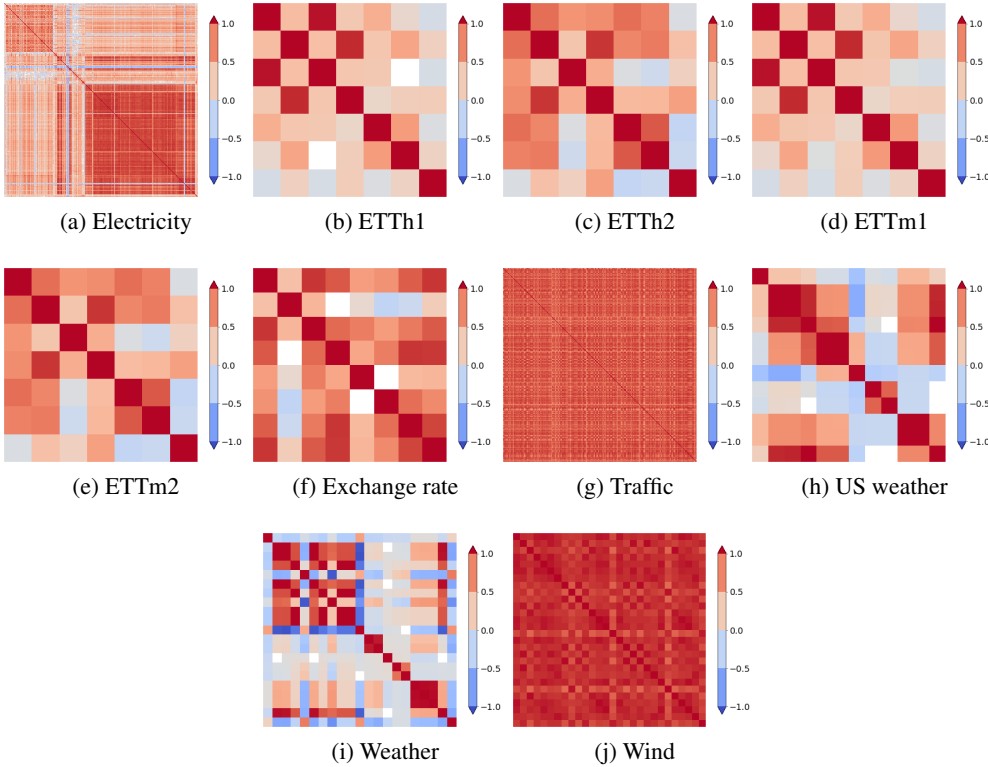

|  |  |  |  |
|---|---|---|---|
| (a) Electricity | (b) ETTh1 | (c) ETTh2 | (d) ETTm1 |
| (e) ETTm2 | (f) Exchange rate | (g) Traffic | (h) US weather |
|  | (i) Weather | (j) Wind |  |

Figure 10: Spearman correlations across variables for all datasets (the train split only), showing how some datasets have more correlation across variables than others. The DAM is a univariate model, but future work may incorporate a multivariate component.

We were interested in the correlation across variables for the commonly used time series datasets. To this end we computed the Spearman correlations between variables in each benchmark dataset, as shown in Figure 10. Coefficient values are only colored when $p < 0.05$, with the cell remaining

---

[19]https://pageviews.wmcloud.org/

blank (null) otherwise. We undertook this analysis to understand what potential impact a multivariate component may have on the DAM. Future work will entail making full use of cross-variable information.

Even though there is significant correlation across variables for some datasets, this does not necessarily mean that a multivariate perspective necessarily improves forecasting. In other words, a correlation between variables **at a shared time** does not directly translate into additional information about the future state of any of those variables. Nie et al. (2022) argued for a univariate approach and showed how multivariate models overfit rapidly, but also suggested that a simplified inclusion of 'some' cross variable information may be beneficial.

## I  ADDITIONAL LONG-TERM FORECASTING RESULTS AND DETAILS

For all the comparison methods in this paper we adopted hyper-parameter settings as described in the corresponding papers. Where relevant we set input/context size windows to be as large as possible (constrained to settings described by the authors), but always ensured that we selected the best performing settings such that the results in this paper represent the best case scenario for comparison methods. We used the official implementation for PatchTST[20], Dlinear[21] and TimesNet. For other methods we used the TimesNet repository as it contains many implementations.[22] We give normalised MSE and MAE in Table 1 in I.1 and results on additional SoTA methods in Section I.2.

### I.1  NORMALISED MSE AND MAE

Table 8: Normalised metrics corresponding to Table 1. For NMSE we normalised by the l2-norm of the ground truth variable, $|v|_2$, and for NMAE we normalised by the l1-norm thereof, $|v|_1$.

| Dataset | Horizon | NMSE | NMAE | NMSE | NMAE | NMSE | NMAE | NMSE | NMAE | NMSE | NMAE | NMSE | NMAE | NMSE | NMAE | NMSE | NMAE |
|---|---|---|---|---|---|---|---|---|---|---|---|---|---|---|---|---|---|
| ETTm1 | 96 | 0.279 | 0.453 | 0.278 | 0.450 | 0.274 | 0.446 | 0.272 | 0.435 | 0.318 | 0.482 | 0.337 | 0.526 | 0.560 | 0.669 | 0.288 | 0.468 |
| ETTm1 | 192 | 0.310 | 0.479 | 0.310 | 0.477 | 0.302 | 0.468 | 0.304 | 0.463 | 0.363 | 0.528 | 0.409 | 0.600 | 0.558 | 0.694 | 0.326 | 0.502 |
| ETTm1 | 336 | 0.317 | 0.485 | 0.317 | 0.483 | 0.329 | 0.493 | 0.334 | 0.489 | 0.400 | 0.565 | 0.619 | 0.794 | 0.677 | 0.796 | 0.366 | 0.545 |
| ETTm1 | 720 | 0.364 | 0.528 | 0.368 | 0.528 | 0.376 | 0.533 | 0.386 | 0.537 | 0.445 | 0.600 | 0.896 | 0.921 | 0.881 | 0.930 | 0.457 | 0.628 |
| ETTm2 | 96 | 0.055 | 0.186 | 0.054 | 0.185 | 0.053 | 0.189 | 0.054 | 0.193 | 0.070 | 0.220 | 0.109 | 0.303 | 0.113 | 0.325 | 0.059 | 0.208 |
| ETTm2 | 192 | 0.071 | 0.212 | 0.070 | 0.211 | 0.071 | 0.217 | 0.073 | 0.224 | 0.103 | 0.267 | 0.214 | 0.442 | 0.205 | 0.445 | 0.088 | 0.257 |
| ETTm2 | 336 | 0.075 | 0.219 | 0.074 | 0.219 | 0.088 | 0.243 | 0.097 | 0.266 | 0.133 | 0.312 | 0.336 | 0.527 | 0.410 | 0.638 | 0.123 | 0.312 |
| ETTm2 | 720 | 0.106 | 0.268 | 0.104 | 0.267 | 0.116 | 0.282 | 0.128 | 0.302 | 0.198 | 0.414 | 1.193 | 1.002 | 0.964 | 0.978 | 0.186 | 0.391 |
| ETTh1 | 96 | 0.336 | 0.509 | 0.330 | 0.505 | 0.335 | 0.505 | 0.341 | 0.505 | 0.385 | 0.543 | 0.358 | 0.531 | 0.649 | 0.807 | 0.367 | 0.536 |
| ETTh1 | 192 | 0.361 | 0.531 | 0.352 | 0.522 | 0.375 | 0.543 | 0.390 | 0.553 | 0.428 | 0.577 | 0.467 | 0.631 | 0.792 | 0.921 | 0.431 | 0.594 |
| ETTh1 | 336 | 0.368 | 0.537 | 0.357 | 0.527 | 0.389 | 0.559 | 0.401 | 0.563 | 0.500 | 0.637 | 0.604 | 0.765 | 0.918 | 1.002 | 0.513 | 0.678 |
| ETTh1 | 720 | 0.394 | 0.582 | 0.379 | 0.558 | 0.412 | 0.597 | 0.448 | 0.639 | 0.533 | 0.687 | 0.731 | 0.866 | 0.900 | 1.001 | 0.727 | 0.845 |
| ETTh2 | 96 | 0.096 | 0.247 | 0.089 | 0.243 | 0.088 | 0.249 | 0.093 | 0.262 | 0.135 | 0.322 | 0.238 | 0.447 | 0.485 | 0.700 | 0.119 | 0.305 |
| ETTh2 | 192 | 0.115 | 0.272 | 0.108 | 0.271 | 0.108 | 0.280 | 0.120 | 0.305 | 0.170 | 0.362 | 0.500 | 0.690 | 1.627 | 1.335 | 0.158 | 0.354 |
| ETTh2 | 336 | 0.118 | 0.276 | 0.110 | 0.276 | 0.116 | 0.296 | 0.144 | 0.340 | 0.182 | 0.384 | 0.752 | 0.911 | 1.482 | 1.340 | 0.198 | 0.406 |
| ETTh2 | 720 | 0.125 | 0.297 | 0.125 | 0.309 | 0.125 | 0.316 | 0.222 | 0.435 | 0.281 | 0.489 | 1.077 | 1.153 | 1.373 | 1.315 | 0.258 | 0.476 |
| ECL | 96 | 0.157 | 0.320 | 0.152 | 0.312 | 0.127 | 0.270 | 0.140 | 0.291 | 0.181 | 0.335 | 0.145 | 0.299 | 0.281 | 0.456 | 0.164 | 0.334 |
| ECL | 192 | 0.174 | 0.338 | 0.169 | 0.330 | 0.146 | 0.292 | 0.154 | 0.306 | 0.186 | 0.345 | 0.160 | 0.316 | 0.294 | 0.474 | 0.175 | 0.346 |
| ECL | 336 | 0.179 | 0.344 | 0.174 | 0.336 | 0.162 | 0.313 | 0.169 | 0.327 | 0.202 | 0.364 | 0.202 | 0.364 | 0.302 | 0.482 | 0.184 | 0.358 |
| ECL | 720 | 0.239 | 0.406 | 0.234 | 0.399 | 0.197 | 0.352 | 0.203 | 0.367 | 0.239 | 0.405 | 0.250 | 0.408 | 0.301 | 0.474 | 0.204 | 0.381 |
| Traffic | 96 | 0.322 | 0.420 | 0.317 | 0.416 | 0.248 | 0.312 | 0.284 | 0.358 | 0.374 | 0.409 | 0.357 | 0.335 | 0.475 | 0.493 | 0.357 | 0.389 |
| Traffic | 192 | 0.331 | 0.429 | 0.326 | 0.425 | 0.262 | 0.322 | 0.293 | 0.365 | 0.379 | 0.411 | 0.382 | 0.358 | 0.465 | 0.477 | 0.369 | 0.398 |
| Traffic | 336 | 0.334 | 0.432 | 0.330 | 0.428 | 0.270 | 0.331 | 0.301 | 0.376 | 0.393 | 0.419 | 0.398 | 0.375 | 0.463 | 0.473 | 0.377 | 0.404 |
| Traffic | 720 | 0.376 | 0.478 | 0.370 | 0.472 | 0.308 | 0.383 | 0.322 | 0.399 | 0.421 | 0.440 | 0.456 | 0.449 | 0.489 | 0.498 | 0.394 | 0.416 |
| Weather | 96 | 0.245 | 0.334 | 0.242 | 0.334 | 0.233 | 0.326 | 0.273 | 0.387 | 0.247 | 0.354 | 0.270 | 0.395 | 0.308 | 0.467 | 0.303 | 0.411 |
| Weather | 192 | 0.303 | 0.392 | 0.300 | 0.392 | 0.303 | 0.397 | 0.339 | 0.451 | 0.297 | 0.415 | 0.350 | 0.484 | 0.363 | 0.518 | 0.368 | 0.479 |
| Weather | 336 | 0.322 | 0.408 | 0.319 | 0.406 | 0.383 | 0.464 | 0.410 | 0.512 | 0.357 | 0.476 | 0.435 | 0.561 | 0.465 | 0.599 | 0.451 | 0.553 |
| Weather | 720 | 0.445 | 0.504 | 0.440 | 0.502 | 0.495 | 0.548 | 0.511 | 0.601 | 0.451 | 0.553 | 0.582 | 0.673 | 0.676 | 0.717 | 0.558 | 0.638 |
| Exchange | 96 | 0.024 | 0.136 | 0.025 | 0.139 | 0.026 | 0.139 | 0.025 | 0.138 | 0.041 | 0.176 | 0.076 | 0.243 | 0.189 | 0.426 | 0.026 | 0.146 |
| Exchange | 192 | 0.049 | 0.192 | 0.050 | 0.195 | 0.065 | 0.224 | 0.046 | 0.193 | 0.112 | 0.293 | 0.137 | 0.335 | 0.246 | 0.487 | 0.052 | 0.214 |
| Exchange | 336 | 0.057 | 0.209 | 0.058 | 0.211 | 0.099 | 0.280 | 0.089 | 0.274 | 0.157 | 0.359 | 0.269 | 0.490 | 0.311 | 0.556 | 0.090 | 0.285 |
| Exchange | 720 | 0.258 | 0.466 | 0.251 | 0.457 | 0.278 | 0.487 | 0.273 | 0.475 | 0.266 | 0.479 | 0.445 | 0.439 | 0.464 | 0.658 | 0.223 | 0.450 |
| Wind | 96 | 0.203 | 0.328 | 0.204 | 0.321 | 0.178 | 0.265 | 0.197 | 0.289 | 0.186 | 0.291 | 0.171 | 0.274 | 0.168 | 0.265 | 0.176 | 0.279 |
| Wind | 192 | 0.214 | 0.336 | 0.218 | 0.333 | 0.192 | 0.280 | 0.216 | 0.307 | 0.200 | 0.306 | 0.189 | 0.289 | 0.186 | 0.280 | 0.193 | 0.293 |
| Wind | 336 | 0.216 | 0.338 | 0.221 | 0.335 | 0.200 | 0.288 | 0.230 | 0.321 | 0.210 | 0.314 | 0.199 | 0.300 | 0.196 | 0.288 | 0.203 | 0.298 |
| Wind | 720 | 0.231 | 0.359 | 0.248 | 0.366 | 0.207 | 0.297 | 0.253 | 0.349 | 0.233 | 0.335 | 0.211 | 0.303 | 0.215 | 0.314 | 0.221 | 0.314 |
| USWeather | 96 | 0.503 | 0.603 | 0.505 | 0.598 | 0.495 | 0.616 | 0.499 | 0.621 | 0.537 | 0.648 | 0.478 | 0.603 | 0.472 | 0.612 | 0.503 | 0.638 |
| USWeather | 192 | 0.549 | 0.643 | 0.557 | 0.640 | 0.561 | 0.673 | 0.547 | 0.666 | 0.589 | 0.700 | 0.526 | 0.649 | 0.541 | 0.669 | 0.553 | 0.675 |
| USWeather | 336 | 0.559 | 0.651 | 0.567 | 0.649 | 0.589 | 0.695 | 0.580 | 0.693 | 0.608 | 0.715 | 0.556 | 0.682 | 0.571 | 0.698 | 0.577 | 0.691 |
| USWeather | 720 | 0.623 | 0.704 | 0.645 | 0.710 | 0.664 | 0.745 | 0.650 | 0.749 | 0.647 | 0.749 | 0.603 | 0.719 | 0.602 | 0.723 | 0.580 | 0.698 |

---

[20] https://github.com/yuqinie98/PatchTST

[21] https://github.com/cure-lab/LTSF-Linear

[22] TimesNet:https://github.com/thuml/Time-Series-Library

## I.2 RESULTS ON ADDITIONAL SoTA METHODS

Table 9: Additional long term forecasting results. Results are always computed as the average of 3 seeded runs. Owing to time and resource constraints, methods with an asterisk* are from single runs (not an average of 3 seeded runs). These will be updated in due course.

| | | FiLM | | TimesNet | | Autoformer | | FEDformer | | Informer | | LightTS | | ETSformer* | |
|---|---|---|---|---|---|---|---|---|---|---|---|---|---|---|---|
| | | MSE | MAE | MSE | MAE | MSE | MAE | MSE | MAE | MSE | MAE | MSE | MAE | MSE | MAE |
| ETTm1 | 96 | 0.309 | 0.351 | 0.341 | 0.376 | 0.371 | 0.413 | 0.474 | 0.465 | 0.624 | 0.560 | 0.395 | 0.409 | 0.528 | 0.496 |
| | 192 | 0.341 | 0.369 | 0.398 | 0.410 | 0.429 | 0.444 | 0.531 | 0.493 | 0.781 | 0.643 | 0.423 | 0.433 | 0.565 | 0.537 |
| | 336 | 0.370 | 0.386 | 0.418 | 0.422 | 0.453 | 0.461 | 0.608 | 0.527 | 0.769 | 0.805 | 0.458 | 0.460 | 0.657 | 0.602 |
| | 720 | 0.419 | 0.413 | 0.481 | 0.457 | 0.538 | 0.501 | 0.604 | 0.529 | 1.032 | 0.773 | 0.546 | 0.517 | 0.800 | 0.695 |
| ETTm2 | 96 | 0.177 | 0.265 | 0.185 | 0.266 | 0.193 | 0.282 | 0.249 | 0.330 | 0.391 | 0.479 | 0.232 | 0.328 | 0.350 | 0.409 |
| | 192 | 0.224 | 0.296 | 0.255 | 0.308 | 0.264 | 0.325 | 0.281 | 0.338 | 0.705 | 0.644 | 0.401 | 0.446 | 0.421 | 0.468 |
| | 336 | 0.275 | 0.331 | 0.320 | 0.348 | 0.325 | 0.364 | 0.339 | 0.373 | 1.454 | 0.922 | 0.481 | 0.492 | 1.421 | 0.993 |
| | 720 | 0.359 | 0.388 | 0.431 | 0.410 | 0.431 | 0.427 | 0.456 | 0.440 | 4.291 | 1.529 | 0.739 | 0.621 | 4.563 | 1.821 |
| ETTh1 | 96 | 0.402 | 0.416 | 0.394 | 0.414 | 0.376 | 0.418 | 0.433 | 0.447 | 0.914 | 0.744 | 0.442 | 0.447 | 0.555 | 0.536 |
| | 192 | 0.441 | 0.441 | 0.455 | 0.452 | 0.420 | 0.444 | 0.503 | 0.488 | 1.007 | 0.785 | 0.496 | 0.479 | 0.686 | 0.619 |
| | 336 | 0.473 | 0.462 | 0.484 | 0.463 | 0.464 | 0.469 | 0.533 | 0.505 | 1.038 | 0.791 | 0.552 | 0.511 | 0.870 | 0.730 |
| | 720 | 0.495 | 0.496 | 0.514 | 0.493 | 0.508 | 0.503 | 0.537 | 0.522 | 1.196 | 0.876 | 0.615 | 0.570 | 1.088 | 0.851 |
| ETTh2 | 96 | 0.315 | 0.360 | 0.339 | 0.376 | 0.348 | 0.391 | 0.384 | 0.415 | 2.949 | 1.361 | 0.410 | 0.447 | 0.382 | 0.429 |
| | 192 | 0.384 | 0.406 | 0.402 | 0.412 | 0.432 | 0.441 | 0.448 | 0.452 | 6.521 | 2.115 | 0.558 | 0.526 | 0.543 | 0.523 |
| | 336 | 0.409 | 0.431 | 0.460 | 0.452 | 0.472 | 0.476 | 0.486 | 0.488 | 5.249 | 1.920 | 0.665 | 0.577 | 0.624 | 0.563 |
| | 720 | 0.428 | 0.448 | 0.456 | 0.463 | 0.476 | 0.488 | 0.590 | 0.540 | 3.936 | 1.683 | 0.992 | 0.712 | 0.632 | 0.573 |
| ECL | 96 | 0.154 | 0.246 | 0.168 | 0.271 | 0.196 | 0.310 | 0.207 | 0.320 | 0.336 | 0.417 | 0.214 | 0.317 | 0.250 | 0.353 |
| | 192 | 0.167 | 0.259 | 0.185 | 0.286 | 0.204 | 0.318 | 0.229 | 0.335 | 0.359 | 0.439 | 0.224 | 0.327 | 0.270 | 0.365 |
| | 336 | 0.189 | 0.284 | 0.202 | 0.303 | 0.227 | 0.340 | 0.244 | 0.348 | 0.358 | 0.438 | 0.244 | 0.347 | 0.285 | 0.377 |
| | 720 | 0.249 | 0.341 | 0.223 | 0.318 | 0.275 | 0.375 | 0.296 | 0.382 | 0.388 | 0.452 | 0.281 | 0.375 | 0.311 | 0.395 |
| Traffic | 96 | 0.411 | 0.284 | 0.593 | 0.319 | 0.582 | 0.365 | 0.664 | 0.413 | 0.730 | 0.414 | 0.667 | 0.422 | 0.970 | 0.583 |
| | 192 | 0.407 | 0.284 | 0.617 | 0.327 | 0.608 | 0.380 | 0.634 | 0.397 | 0.759 | 0.439 | 0.661 | 0.426 | 1.020 | 0.593 |
| | 336 | 0.426 | 0.298 | 0.631 | 0.336 | 0.620 | 0.387 | 0.630 | 0.394 | 0.848 | 0.479 | 0.682 | 0.436 | 1.043 | 0.599 |
| | 720 | 0.526 | 0.372 | 0.665 | 0.352 | 0.639 | 0.393 | 0.645 | 0.399 | 0.978 | 0.547 | 0.738 | 0.457 | 1.088 | 0.614 |
| Weather | 96 | 0.185 | 0.235 | 0.171 | 0.220 | 0.223 | 0.306 | 0.308 | 0.361 | 0.385 | 0.437 | 0.173 | 0.233 | 0.222 | 0.307 |
| | 192 | 0.225 | 0.265 | 0.230 | 0.270 | 0.275 | 0.341 | 0.324 | 0.377 | 0.458 | 0.468 | 0.216 | 0.276 | 0.280 | 0.365 |
| | 336 | 0.268 | 0.297 | 0.283 | 0.304 | 0.338 | 0.383 | 0.370 | 0.399 | 0.560 | 0.522 | 0.266 | 0.315 | 0.359 | 0.425 |
| | 720 | 0.330 | 0.341 | 0.358 | 0.354 | 0.410 | 0.418 | 0.429 | 0.435 | 1.052 | 0.756 | 0.330 | 0.363 | 0.494 | 0.513 |
| Exchange | 96 | 0.213 | 0.345 | 0.112 | 0.240 | 0.162 | 0.292 | 0.152 | 0.282 | 0.913 | 0.765 | 0.132 | 0.217 | 0.096 | 0.220 |
| | 192 | 0.320 | 0.424 | 0.220 | 0.341 | 0.280 | 0.384 | 0.280 | 0.388 | 1.153 | 0.859 | 0.440 | 0.391 | 0.195 | 0.319 |
| | 336 | 0.512 | 0.539 | 0.385 | 0.456 | 0.446 | 0.492 | 0.494 | 0.524 | 1.593 | 1.017 | 0.506 | 0.541 | 0.373 | 0.449 |
| | 720 | 1.144 | 0.801 | 0.978 | 0.752 | 1.178 | 0.834 | 1.119 | 0.822 | 2.668 | 1.341 | 0.967 | 0.754 | 0.779 | 0.668 |
| Wind | 96 | 0.199 | 0.222 | 0.201 | 0.230 | 0.176 | 0.241 | 0.201 | 0.262 | 0.181 | 0.224 | 0.172 | 0.217 | 0.198 | 0.262 |
| | 192 | 0.219 | 0.236 | 0.221 | 0.242 | 0.193 | 0.253 | 0.214 | 0.269 | 0.196 | 0.236 | 0.189 | 0.229 | 0.208 | 0.269 |
| | 336 | 0.234 | 0.251 | 0.231 | 0.246 | 0.206 | 0.266 | 0.246 | 0.292 | 0.204 | 0.244 | 0.200 | 0.237 | 0.215 | 0.277 |
| | 720 | 0.264 | 0.289 | 0.244 | 0.253 | 0.228 | 0.292 | 0.256 | 0.306 | 0.206 | 0.246 | 0.220 | 0.253 | 0.228 | 0.297 |
| USWeather | 96 | 0.480 | 0.482 | 0.493 | 0.483 | 0.504 | 0.502 | 0.556 | 0.540 | 0.498 | 0.506 | 0.487 | 0.492 | 0.487 | 0.501 |
| | 192 | 0.528 | 0.516 | 0.552 | 0.520 | 0.568 | 0.542 | 0.630 | 0.581 | 0.562 | 0.550 | 0.533 | 0.526 | 0.529 | 0.534 |
| | 336 | 0.561 | 0.537 | 0.586 | 0.543 | 0.602 | 0.560 | 0.612 | 0.571 | 0.606 | 0.577 | 0.550 | 0.541 | 0.552 | 0.549 |
| | 720 | 0.647 | 0.589 | 0.611 | 0.559 | 0.632 | 0.581 | 0.655 | 0.595 | 0.596 | 0.575 | 0.587 | 0.566 | 0.589 | 0.573 |

## J    FORECASTING VISUALISATIONS

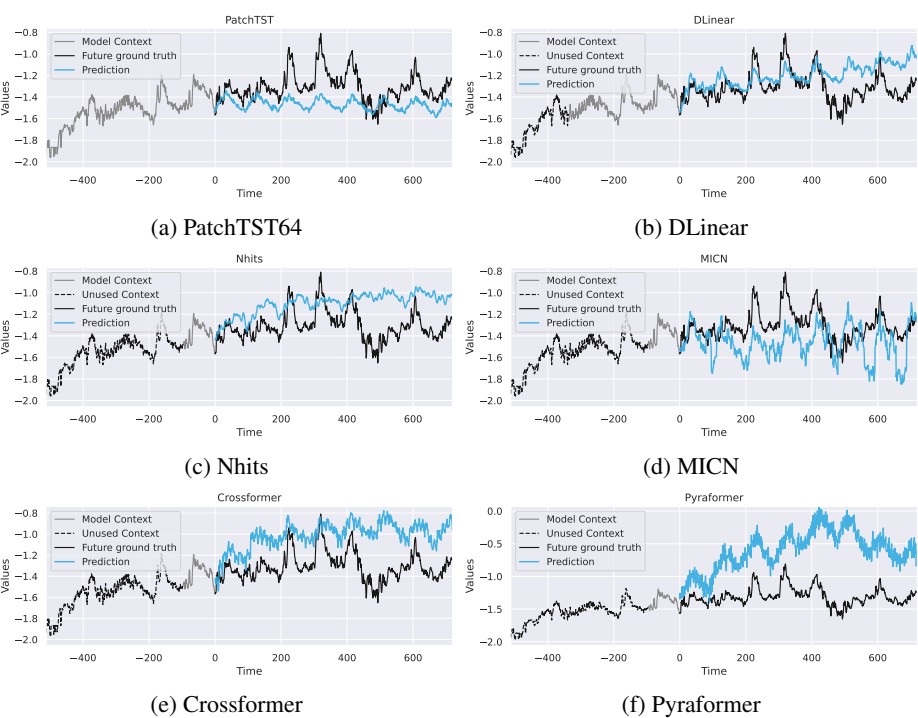

Figure 11: Forecasting demonstrations on ETTm1 (test set; forecasts commence exactly halfway through the entire set) for competitor methods.

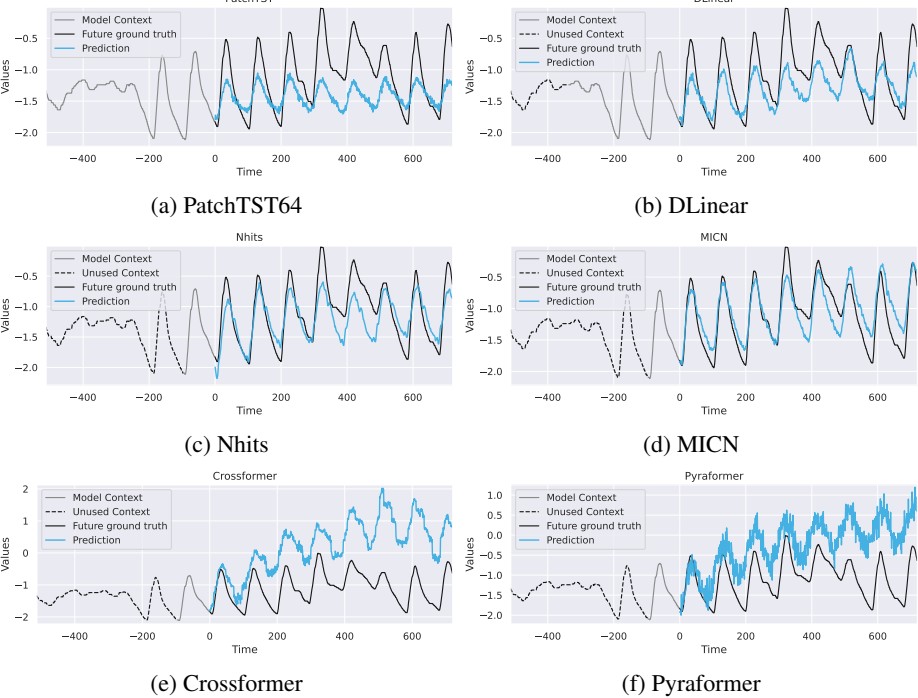

Figure 12: Forecasting demonstrations on ETTm2 (test set; forecasts commence exactly halfway through the entire set) for competitor methods.

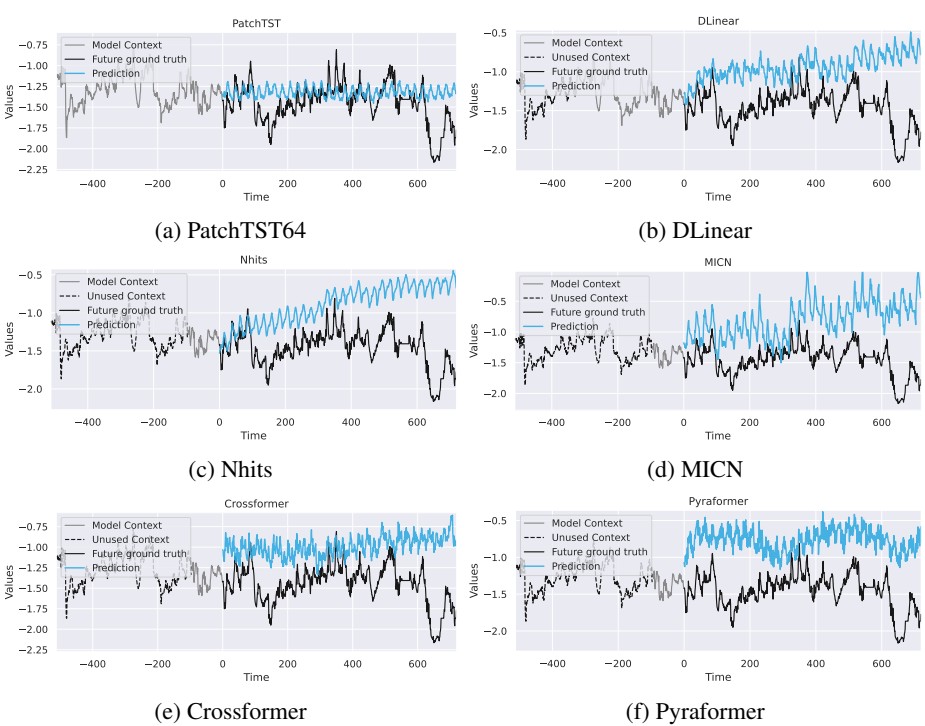

Figure 13: Forecasting demonstrations on ETTh1 (test set; forecasts commence exactly halfway through the entire set) for competitor methods.

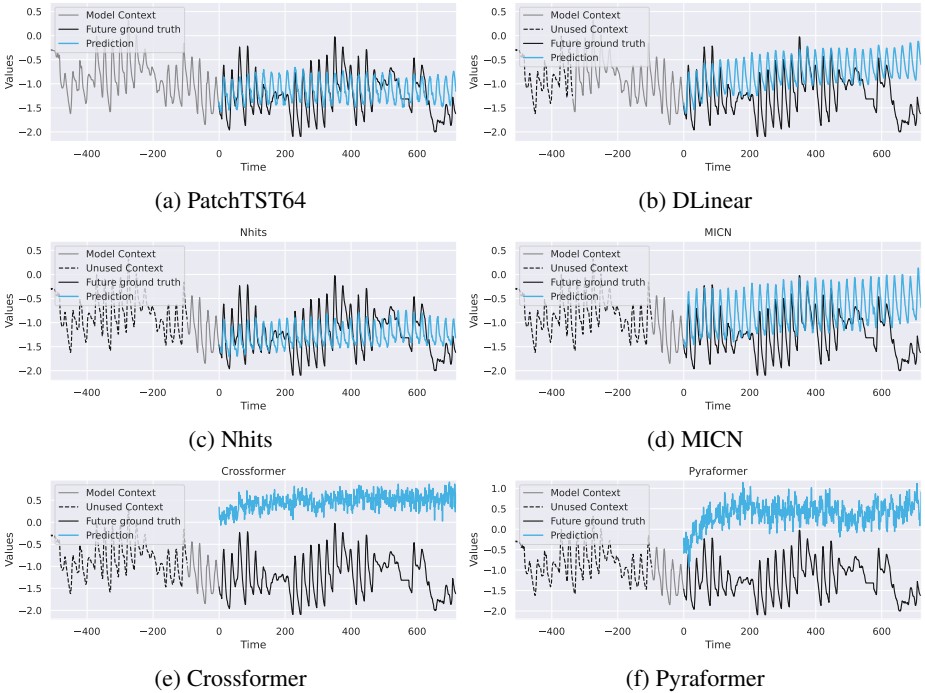

Figure 14: Forecasting demonstrations on ETTh2 (test set; forecasts commence exactly halfway through the entire set) for competitor methods.

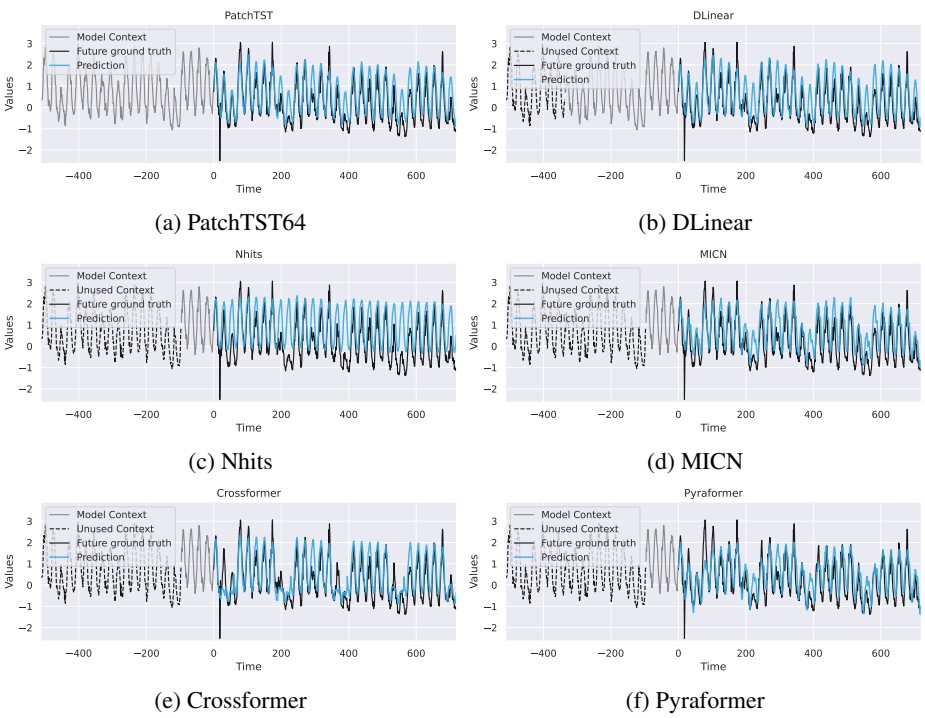

Figure 15: Forecasting demonstrations on ECL (test set; forecasts commence exactly halfway through the entire set) for competitor methods.

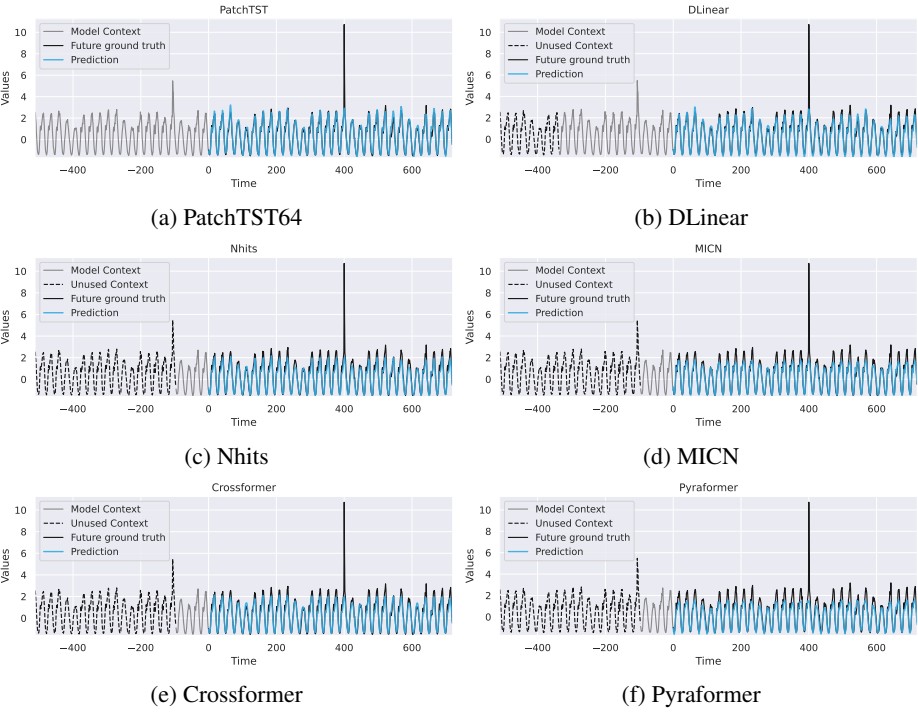

Figure 16: Forecasting demonstrations on Traffic (test set; forecasts commence exactly halfway through the entire set) for competitor methods.

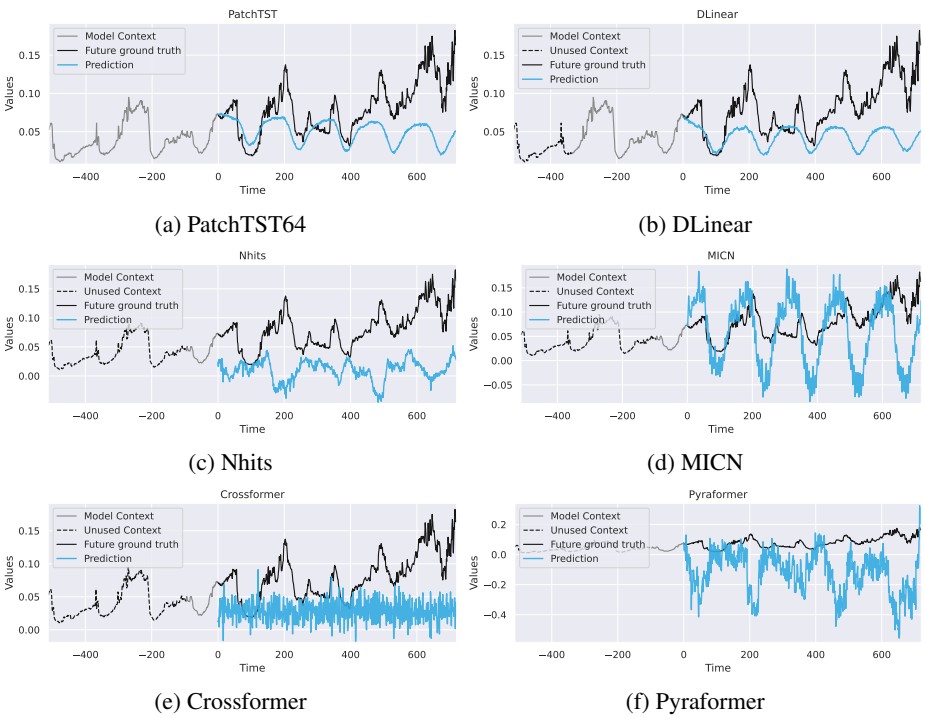

Figure 17: Forecasting demonstrations on Weather (test set; forecasts commence exactly halfway through the entire set) for competitor methods.

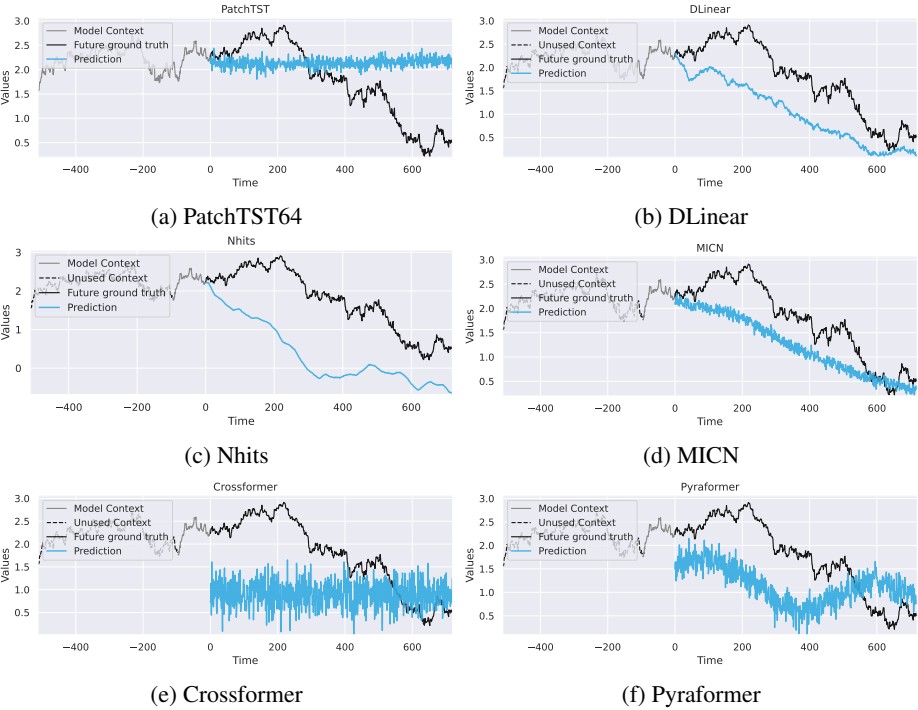

Figure 18: Forecasting demonstrations on Exchange (test set; forecasts commence exactly halfway through the entire set) for competitor methods.

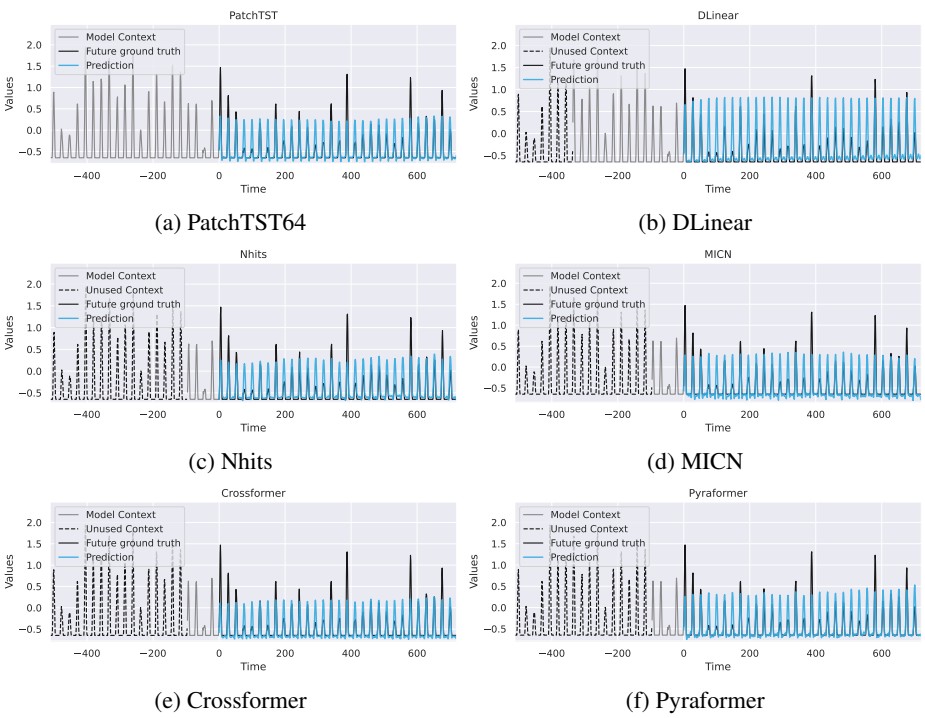

Figure 19: Forecasting demonstrations on Wind (test set; forecasts commence exactly halfway through the entire set) for competitor methods.

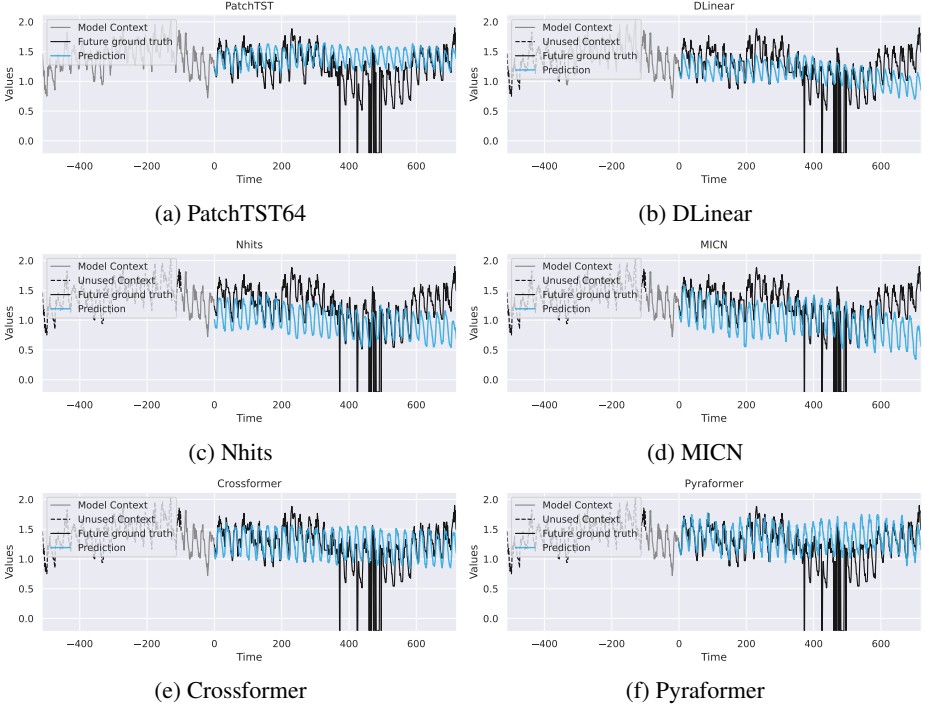

Figure 20: Forecasting demonstrations on USWeather (test set; forecasts commence exactly halfway through the entire set) for competitor methods.

# K    ADDITIONAL DETAILS FOR HELD-OUT EXPERIMENTS

This appendix details the experimental setup for results in Section 4.2. Table 7 contains three main sections: '**zero-shot**', '**fine-tuned**', and '**standard training**'.

**Baselines**    For standard training of baseline methods we used settings from their respective papers, paying heed to which settings (for short versus long-term forecasting; e.g., Illness versus UCIPower). For baseline methods in the zero-shot setting, we reported the average performance across all model variants trained for Table 1.

**DAM zero-shot**    For the DAM in zero-shot mode we first scaled time units by inspection on the validation set for some datasets in order to enable the DAM to a better basis function coverage – details of these settings in Table 7. Such a visual inspection process is realistic compared to a full validation search over time units when, during deployment, very little data exists to determine what scales are reasonable. We used the same HSR settings as in Table 1 for long-term datasets and reduced context size to 250, ToME reduction to 200, and $\sigma$ to 40 for short-term datasets (Illness, MWeb, and MTemp).

**DAM fine-tuned**    For fine-tuning the DAM we implemented short training runs for each target dataset. Owing to difference in dataset sizes and complexity, we and conducted a small hyper-parameter search, using validation, over training iterations and initial learning rates (decayed with a cosine annealing scheduler to $1^{-14}$) to yield the following settings:

1. **Illness**: a starting learning rate of $1^{-6}$ and 10000 iterations.
2. **Weekdays**: a starting learning rate of $1^{-5}$ and 10000 iterations.
3. **UCIPower**: a starting learning rate of $1^{-5}$ and 10000 iterations.
4. **Azure**: a starting learning rate of $1^{-5}$ and 10000 iterations.
5. **MTemp**: a starting learning rate of $1^{-6}$ and 10000 iterations.
6. **MWeb**: a starting learning rate of $1^{-5}$ and 10000 iterations.
7. **MWeather**: a starting learning rate of $1^{-3}$ and 30000 iterations.
8. **MMeters**: a starting learning rate of $1^{-3}$ and 30000 iterations.

Note that these training runs are 2 orders of magnitude shorter than training the DAM from scratch. On a NVIDIA A40 GPU they take between 20 minutes and 1.5 hours.

## L    FURTHER BASIS COMPOSITION DEMONSTRATIONS

Figure 21 is an expanded version of Figure 6 and Figure 22 is another version on ETTh2.

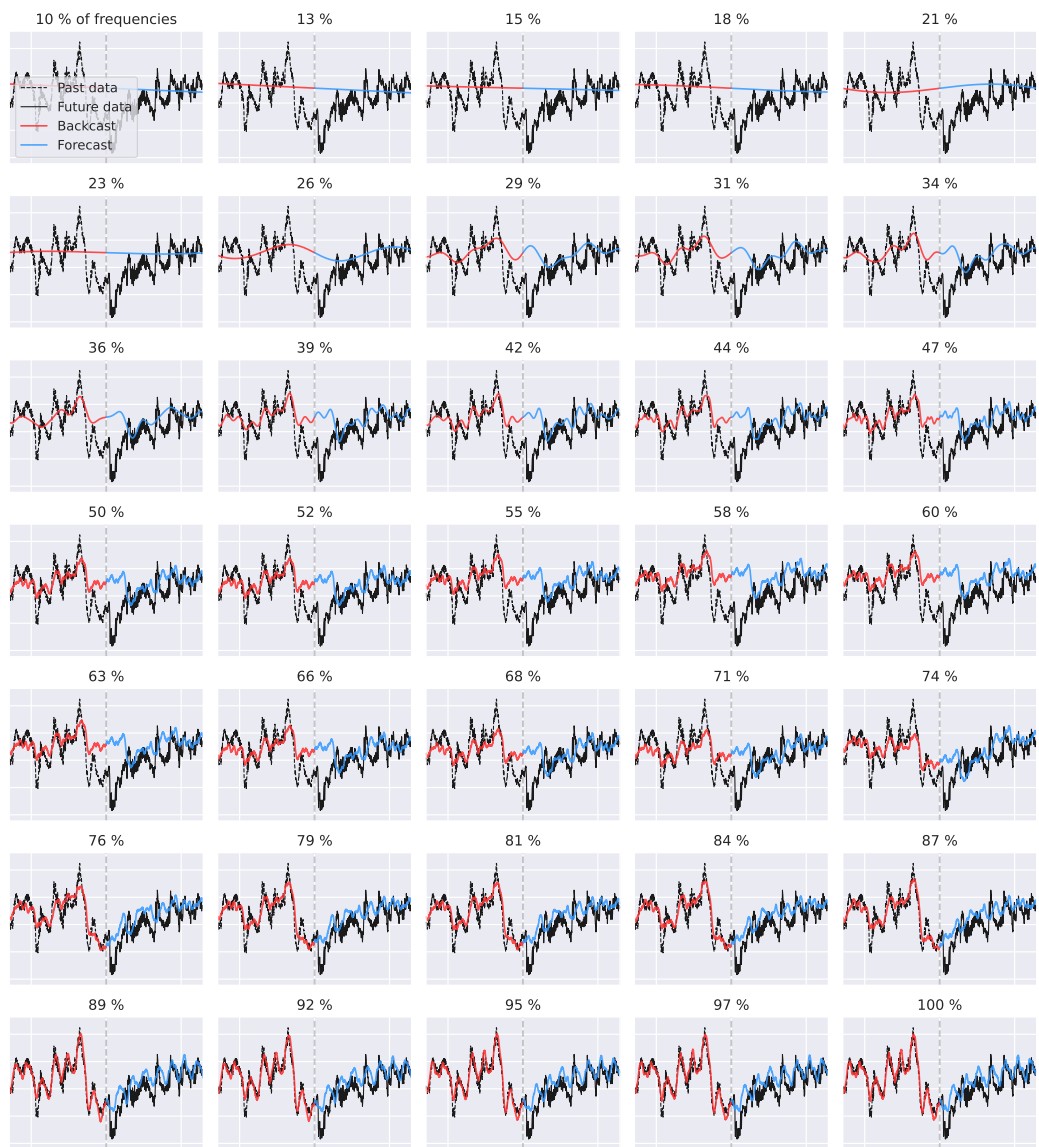

Figure 21: Dataset: ETTm1; basis function composition from low to high frequency components. From upper left to lower right, the percentage labels denote the percentiles of frequencies used to compose the forecast (e.g., 10% denotes using all frequencies that constitute the first 10 percentile).

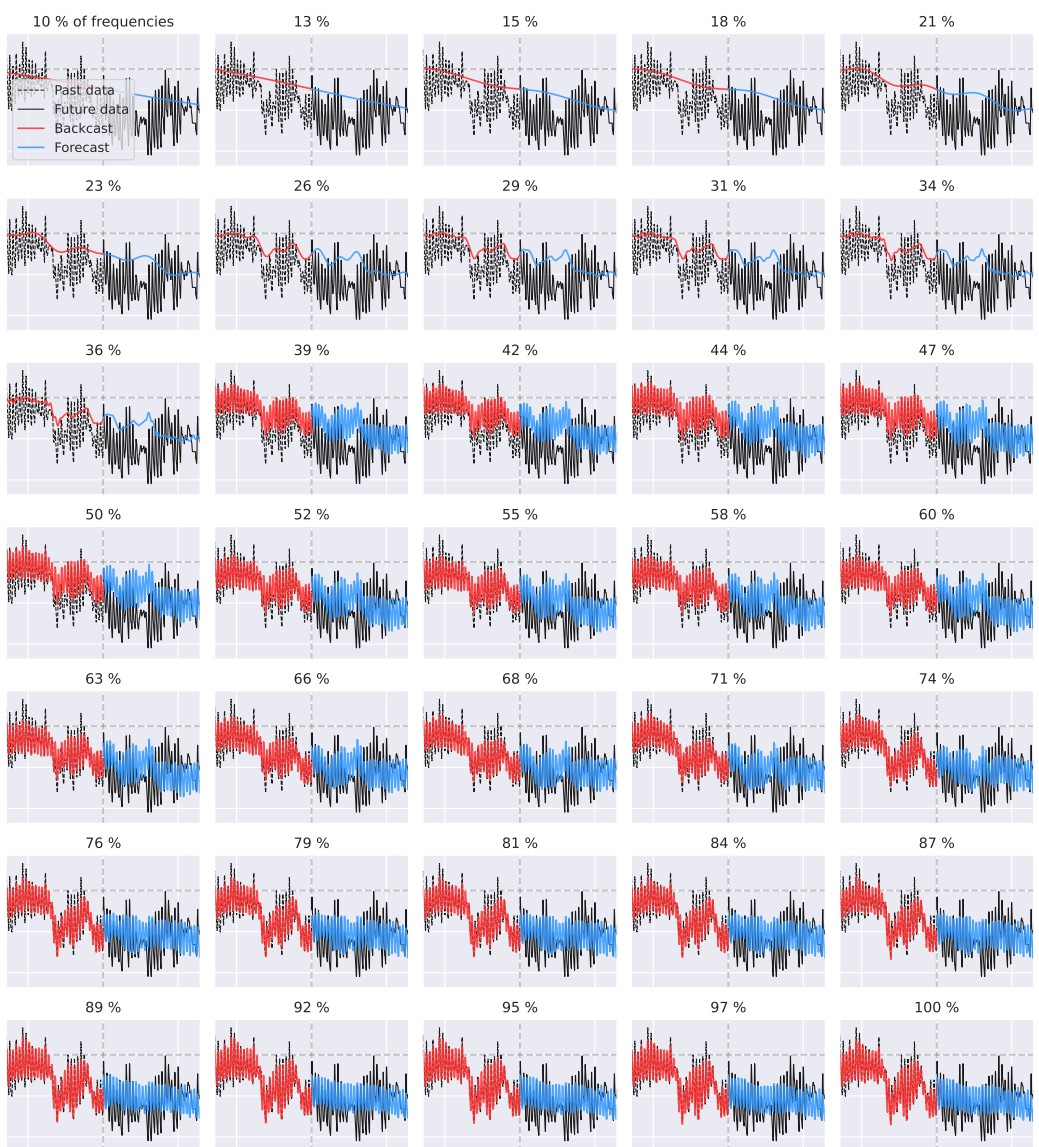

Figure 22: Dataset: ETTh2; basis function composition from low to high frequency components. From upper left to lower right, the percentage labels denote the percentiles of frequencies used to compose the forecast (e.g., 10% denotes using all frequencies that constitute the first 10 percentile).

## M    ADDITIONAL INTERPRETABILITY DEMONSTRATIONS

Figures 23 and 24 are extensions of Figure 5 but on different datasets. They show the DAM's attention and resultant basis coefficients for a variety of datasets. The supplementary video entitled 'DAM_interpretability_demo1' demonstrates the same interpretability of Figure 5 but as a video as time progresses, and shows how some heads in the attention mechanism focus on certain patterns, while other heads are dedicated to certain points in time. This is not a property that is built into the DAM, but rather one that emerges from learning.

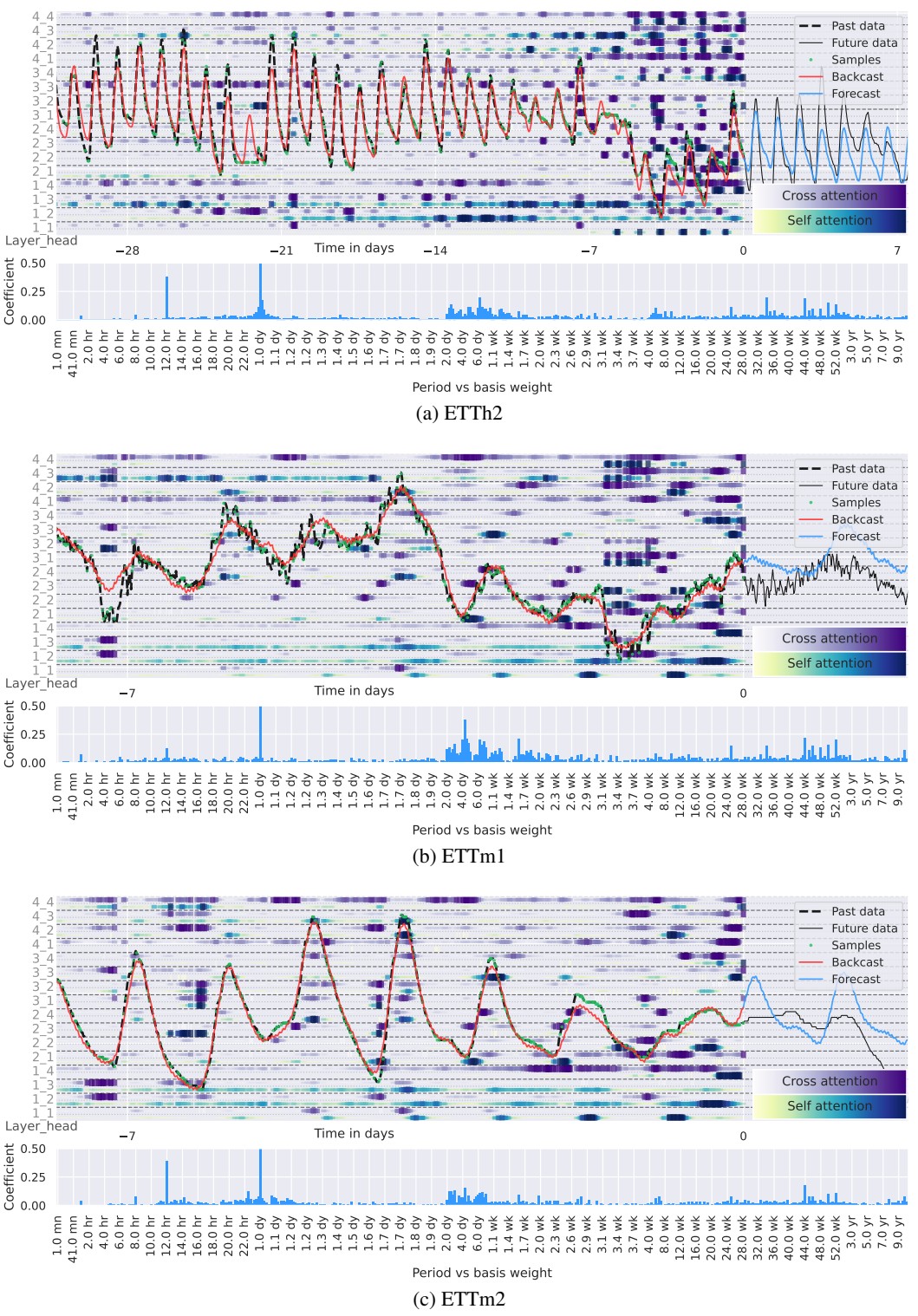

Figure 23: Extensions to Figure 5 with other datasets.

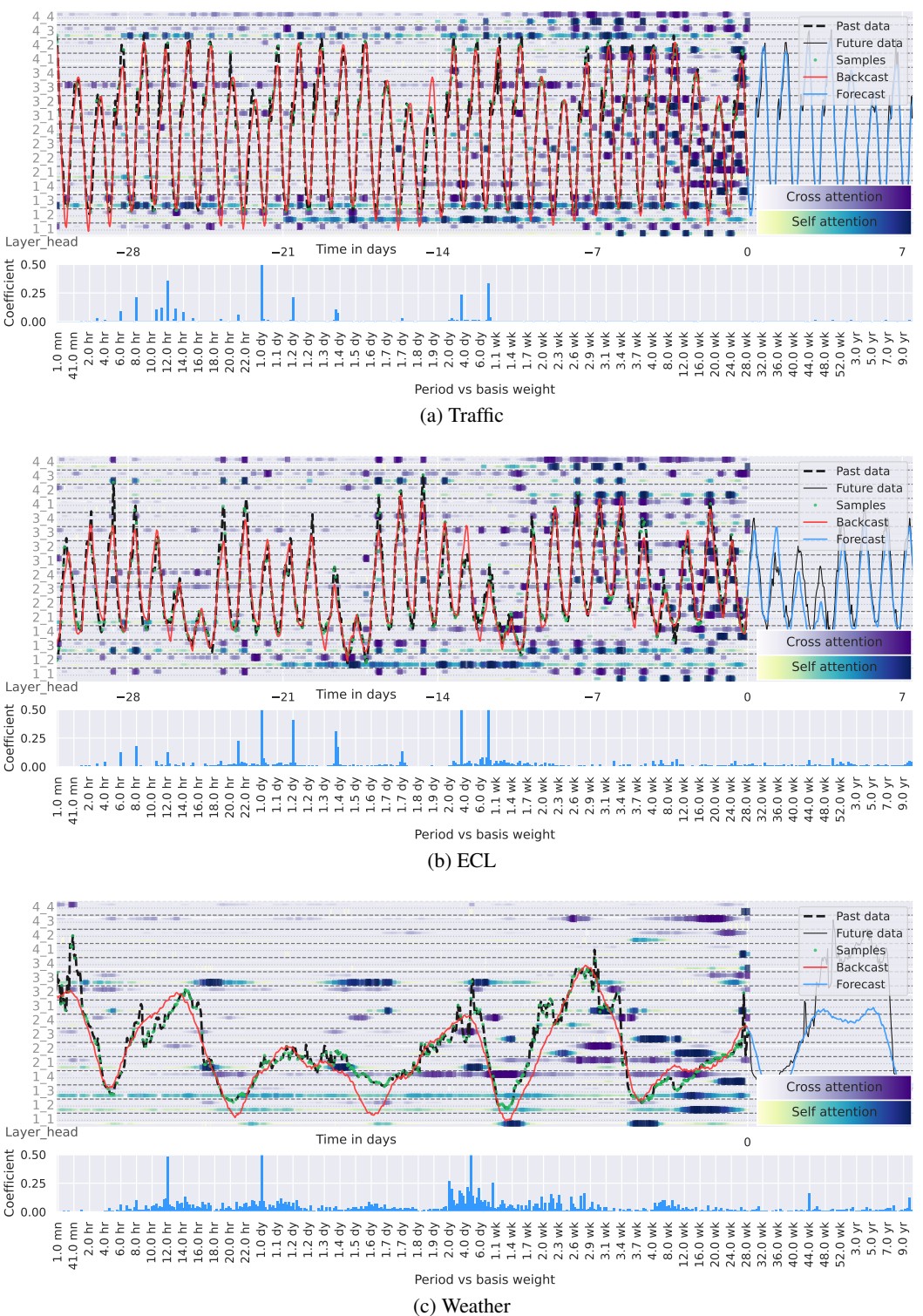

Figure 24: Extensions to Figure 5 with other datasets.

## N  SCALING LAWS OF FOUNDATION MODELS

The term foundation model was coined by researchers in the field as "any model that is trained on broad data (generally using self-supervision at scale) that can be adapted (e.g., fine-tuned) to a wide range of downstream tasks." (Time, 2023) Scaling laws (Kaplan et al., 2020; Frantar et al., 2023) are a property (rather than a requirement) of almost all larger models, not particularly just foundation models—and indeed do not necessarily imply the required generalisation—though they are helpful in understanding how the future development might unfold.

Owing to time and resource constraints, we trained several variants of the DAM on ETTh1 (using a shorter training run of 100000 iterations) in order to build a perspective on how the DAM scales with increased model size. Figure 25 shows the test performance of ETTh1 for different model sizes and also indicates the DAM trained for this paper. While not conclusive, these results show that the DAM scales well with increased model size. Further, training across multiple datasets and for longer has a strong positive impact, as evidenced by the superior performance of the DAM used throughout this paper.

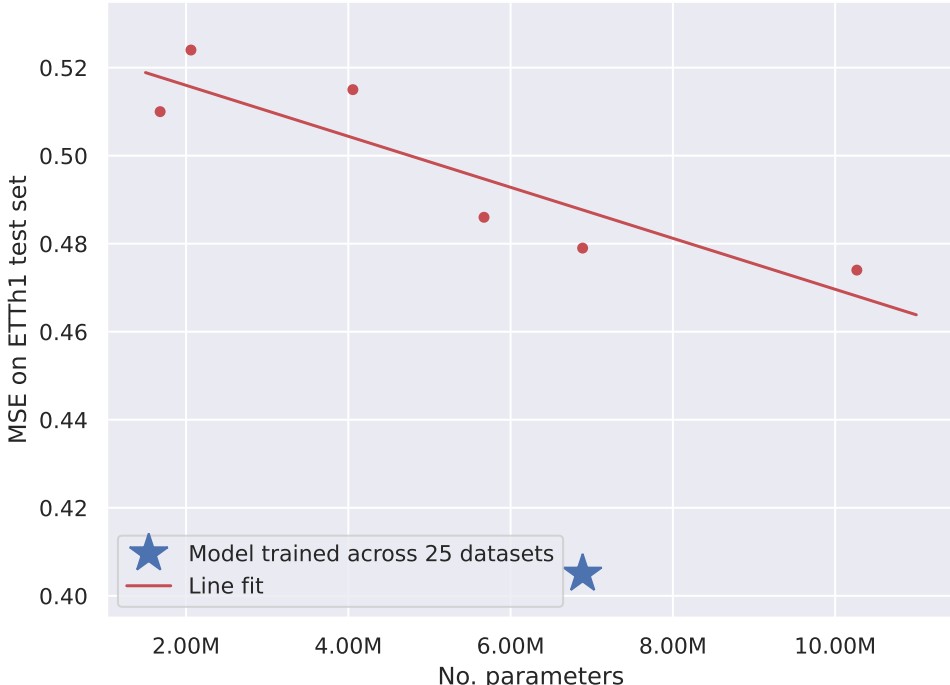

Figure 25: The DAM scaling laws. We trained several variations of the DAM, by scaling $D_{\mathrm{model}}$ and the number of layers, on ETTh1 to develop this plot. Also shown is the performance of the single DAM used throughout this paper (i.e., across 25 datasets), evidencing that it benefits from training across diverse datasets.

What defines a foundation model is not that is is *trained across* multiple datasets, but that it performs well when *tested across* diverse tasks (and hence datasets) that are not of the exact form it is trained on – something that we have done extensively in Section 4. All foundation models have a scope (language, vision tasks, etc.) and the scope of the DAM is univariate time series forecasting.

