# OpenReview forum: "DAM: Towards a Foundation Model for Forecasting"
_ICLR.cc/2024/Conference — ICLR 2024 poster_

### Official Review · Reviewer_PhNp · 2023-11-01

**Soundness:** 3 good
**Presentation:** 3 good
**Contribution:** 4 excellent
**Rating:** 8
**Confidence:** 5

**Summary:**

The author proposes a foundation model for time series forecasting called DAM (deep data-dependent approximate analytical model).
DAM takes input time series data and samples (time, value) pairs using HSR (history sampling regime) technique. HSR is not a regular and fixed-length sampling method unlike other time series forecasting techniques. Instead, it defines a probability distribution that is closer to the top-head section as the sampling points are closer to the prediction target point, and closer to the long-tail section as they are farther away. The author claims that this method is a major difference from other time series forecasting models and prevents overfitting and improves generalization performance. The sampled (time, value) pairs and the coefficient values initialized by the linear solver are used in the transformer to determine the coefficient values of 219 basis functions. The final predicted time-series is a combination of the basis functions.
The author evaluated 12 benchmark datasets used at training and 2 datasets not used at training (for validation of generalization performance), and compared them with 6 existing sota models trained specifically for each dataset, showing better or similar performance. In particular, DAM showed excellent performance in long-term forecasting. Additionally, DAM can adjust the trade-off between inference cost and performance by changing the context size, and also provides interpretabilty features based on basis function composition and attention map analysis.

**Strengths:**

1. Addressing foundation model for time-series forecasting model is a significant and timely task (and it is a very difficult problem).

2. The irregular and non-fixed size sampling technique is interesting.

3. Various and numerous experiments were conducted to verify the ability of DAM.

**Weaknesses:**

1. To be a foundation model, it is necessary to verify that the computational amount and performance follow the scaling law, but there is no related content. Even if the computational amount and performance do not follow the scaling law in time series forecasting, the authors should have explained why they had to use 10 datasets to train DAM (not single dataset or many more), and what was the bottleneck for improving the foundation model performance.

2. The paper too much depends on appendix. The content provided in the main paper should be a complete argument, but it is difficult to follow without looking at the appendix. For example, in 4.1 Results and discussion, the authors need to summarize the content of Table 1 and argue the authors’ claim, but it is omitted. Instead, the content of the appendix is summarized. To be a regular paper, it seems that the structure of the paper needs to be more organized.

3. One of the most important abilities of a time series forecasting model is peak time prediction, but as the authors pointed out in Limitations, DAM does not seem to predict well. This limits the role of DAM in real-world problems.

**Questions:**

1. According to the description of Table 1, DAM performed multivariate forecasting. I guess that when creating (time, value) pairs, multiple variables were included in the value part. Is that right? If so, I think it should be more clarified in the paper.

2. At Figure 6, it seems that the inference time can be reduced from 200ms to below 100ms depending on the context size. However, the inference time and cost seem to be very small compared to LLM (~seconds), so it seems possible to reduce costs by increasing the computing resources not expensively at around 200ms. I don’t fully understand what the authors want to deliver in 5.1 FLEXIBLE INFERENCE COST. Could you explain more about it? (sorry)

---

> ### Author Response · Authors · 2023-11-13
> **Response 1/2**
>
> Thank you for your insightful review. Please note that we updated the paper shortly after the rebuttal period commenced, including the following: an improved model that uses 437 basis functions that is trained on 25 datasets (an additional 15 training datasets that we sourced and collected after submission), improved performance across all tasks, and improved figures. Please see the comment summarising these and other initial changes. That said, we will be updating the paper before the end of the rebuttal period, accounting for any of your concerns and suggestions.
>
> ## Regarding the DAM as a foundational model
> The process of collecting, curating, and training on many time series datasets is time intensive and challenging. While there are many very short time series datasets readily available, these are less useful for training when the task is long-term forecasting, hence we do not include short datasets. More data means that we can train for longer without overfitting, and typically generalise better on test set performance and in the zero-shot setting. However, longer training is also costly and time-intensive. As we mentioned in the conclusion: future work will entail scaling up the DAM both in terms of model size and training corpus. As it is, the DAM is the only forecasting model that is designed for training across many time series datasets. We had to rethink the way time series were ingested and predicted in order to design the DAM; and that is our primary contribution. The generality, flexibility, high-performance, and utility of the DAM is what makes it a foundational model.
>
> ### The nature of time series data
> There are several challenges to overcome for training a time series foundational model, namely:
> 1. Collection and curation of time series data.
> 2. The non-iid nature of time series datasets (i.e., temporally close data points are highly dependent).
> 3. Owing to this non-iid property, balancing amongst data points, variables, and datasets is non-trivial and remains an open question. For example, an extremely long time series dataset that is highly repetitive (i.e., a daily repeating signal) is less 'useful' for training than a shorter time series that contains many diverse patterns that are 'useful' to learn on. Thus, the optimal training regime is complex and not yet known. Perhaps this is the performance bottleneck.
> 4. Scaling the model up results in longer training runs.
>
> That said, the DAM is a foundational model because **it can be trained across many datasets** by design. Hopefully the inclusion of the additional 15 time series datasets, and consequent improved performance, is good evidence for this. We are open to any analysis suggestions you may have, however. Please let us know if there are any additional experiments you would like to see.
>
> ## Many appendices
> As you rightly mentioned, addressing a foundational model for time series forecasting is very challenging. Hence, we required to include longer appendices. . Another reason for the large number of appendices is that we endeavoured to be as open and clear as possible with this work. For example, including the PyTorch code of the model directly in an Appendix makes it far easier for other researchers to reuse and replicate our work. Most of the appendices are not strictly necessary, but rather a consequence of us trying open our research, since we believe that the DAM is an actual important contribution to the field of time series forecasting because of what it enables.
>
> In the current update of the paper we improved the results discussion such that it relies less on the appendices.

---

> > ### Comment · Reviewer_PhNp · 2023-11-15
> >
> > From your further experiments (437 basis functions, additional 15 train datasets), it seems that you also assume the scaling laws in the time-series domain or at least have belief that more data will increase the performance of DAM (please correct me if not, but in this case, please explain why did you increase the scale of DAM to improve performance)
> > How can we further increase performance of a foundation model of the time-series domain similar to the computer vision domain (https://arxiv.org/abs/2103.00020) and the natural language domain (https://arxiv.org/abs/2001.08361)?
> > Is it enough for us to increase the scale of the computation?
> > I think a foundation model paper should give a direction about it because that paper should be the foundation of the future time-series research as well.
> >
> > I also want to hear the author's opinion about the necessity conditions for a model to be considered a foundation model.
> > As this paper (https://fsi.stanford.edu/publication/opportunities-and-risks-foundation-models) pointed out, could you find any emergence property from DAM?
> > You said 'the DAM is a foundational model because it can be trained across many datasets by design'. Yes, it is important. However, many machine learning-based forecasting models are already using a cross-learning strategy that trains multiple time-series dataset at once (Section 4.3, https://www.sciencedirect.com/science/article/pii/S0169207021001874). What distinguishes the DAM from existing methods (or the multi-task learning method) that makes it a foundational model?
> >
> > For the rest of your answer, I will get back to you soon!
> >
> > Thanks!

---

> > > ### Author Response · Authors · 2023-11-15
> > > **Why is the DAM foundational?**
> > >
> > > Thank you very much for your reply. Please note that we are close to submitting another revision that takes all changes into account, notably improved experiments and performance on held-out data. We will show that the DAM **performs on par or better on held out datasets, even when SoTA models are trained on those datasets**.
> > >
> > > ## Scaling laws (versus Vision and NLP)
> > >
> > > We do believe that performance increases with more data and model scaling, but we cannot make any formal claims on this belief owing to the lack of evidence. Particularly, even though we have scaled up the collective training dataset for the DAM to 25 datasets (over 44 million data samples), this is still far less than data available and used in both vision and NLP domains. The reason for this is complex, but related to the following:
> > > 1. A single 'sample' is a 1D observation of a univariate, while a sample in vision and NLP is high dimensional.
> > > 2. Time series data is non-i.i.d., meaning that a given sample is directly dependent on the values of preceding samples (and across variables within a dataset). This is simply the nature of time series data, but effectively reduces the the size of training data by an appreciable factor.
> > > 3. Periodicity in time series further exacerbates this issue. For example, one might have trillions of per-minute samples for a given variable (e.g., in the cloud domain) but this variable might have very strong daily periodicity, meaning that the total information content is low.
> > >
> > > That said, even with our large training corpus and updated model, we have yet to leverage the advantages of power scaling, but believe that the DAM remains foundational because it is the only existing model that is poised to benefit from this.
> > >
> > > ## Training
> > > As a reference point, consider that existing methods typically train for approximately 10 epochs and must use early-stopping. For a dataset of length 10000 and a minibatch of 32, that results in 312 iterations of training. Conversely, we trained for over 1 million iterations. The more data we can collect (as future work), the longer we will be able to train for, and the larger the model can be. We mentioned at the end of the paper that future work will involve scaling up dataset and model size. We acknowledge that this is an important direction, but must reserve it for future work owing to resource constraints.
> > >
> > > ## Why is the DAM foundational?
> > > Crucially, we believe the DAM is foundational because because of 3 key components:
> > > 1. Its ability to consume irregularly sampled data from arbitrarily far into the past.
> > > 2. Its ability to forecast to any time (past or future).
> > > 3. Evident high performance and transfer.
> > >
> > > Consider that one can train any forecasting method on sufficiently 'well-behaved' data (e.g., two datasets from the same resolution, regularly sampled, without missing data), but this need not translate into improved performance. Particularly regarding resolution, existing methods are often disadvantaged **by design**: the patterns that they must learn to extract from per-minute versus per-day data are vastly different, resulting in the need to solve two very different problems simultaneously. Owing to our TV-token formulation, the DAM learns patterns at both scales without interference, and benefits (as evidenced by our results) from this.
> > >
> > > ### Emerging properties
> > > We do find emergent properties in the DAM. For example, it finds high basis coefficients for very long periods for ETTh1 (currently Figure 9) **even when this dataset is shorter than those periods**. This observation could inform future collection procedures. Compare this to the same analysis on traffic (currently Figure 25.a.): long periods are less useful.
> > >
> > > Another emergent property that we did not highlight in the paper (space is limited) is **sample uncertainty**: we can take multiple samples from the HSR, forecast for each, and measure the uncertainty in the forecast. This is not designed into the DAM, but is a useful property of the HSR approach.
> > >
> > > ## Foundational for the future
> > > Training variants of a model design on a single dataset-horizon combination is, in our opinion, outdated and was the central reason for us creating the DAM. We firmly believe that our approach is the future of time series forecasting because of its elegant design, enabling future researchers to ask interesting questions, such as:
> > > 1. How do we balance sampling data points given the unique properties of time series?
> > > 2. How do we schedule training?
> > > 3. What loss (or scaling) is optimal for variable horizons?
> > > 4. What is the best HSR strategy?
> > > 5. How do we generalise cross-variable (i.e., multivariate) information such that the DAM can use it?
> > > 6. Does the DAM adhere to scaling laws similar to Vision and NLP domains?
> > >
> > > We give options for some of the above in our (many) appendices. Please let us know if you would like us to include a summary of these future directions and/or emergent properties in the paper, and feel free to ask any more questions.

---

> ### Author Response · Authors · 2023-11-13
> **Response 2/2**
>
> ## Peak prediction
> One of the contributions of this paper is that we compare 14 SoTA baseline methods (with 8 of in the appendix, owing to space constraints and their relatively lower performance compared to other baselines) across 10 benchmark datasets. It is rare in this field to see comparisons across this many datasets. We intentionally did not include weak or outdated methods (e.g., LSTMs) because of how strong the DAM's performance is, and we wanted to posit it amongst the absolute current SoTA. This is one reason for the DAM to show weaker peak time predictions in comparison to some of the selected methods, as we are only comparing against the best models in the SoTA. The updated model (see above) has improved performance on 'peak' data.
>
> ## The DAM is univariate
> Please see the separate comment entitled 'A note on inference and the univariate nature of the DAM', where we explain precisely how inference is accomplished, and how the DAM is a fully univariate model. To predict a given variable the DAM only consumes context **from that variable**. We will clarify this by adding a section in an incoming update. Future work will involve extending the DAM by adding a generalised multivariate component.
>
> ## Flexible inference cost
> The DAM is far smaller than common LLM models. What we are trying to show in this section is that the DAM is flexible, and that a single model can incur low or high computational cost, depending on user choice. The ability to balance performance and inference cost, without having to train additional models, can be extremely useful in practical scenarios. For example, in cloud resource allocation, 200ms might be too high a cost. Although we agree that 200ms is already inexpensive, this might not hold for much larger versions of the DAM (future work).

---

> ### Author Response · Authors · 2023-11-20
> **Rebuttal changes**
>
> Good day.
>
> Please note that we have updated the paper as per your commentary and requests. There is a summary entitled "Rebuttal Summary - After addressing reviewers' comments and questions".
>
> Of particular note for your consideration:
> 1. Appendix N - a small scale test of the DAM's scaling with model size (for a perspective of 'scaling laws'). We put this experiment together based our discussions with you. We find that the DAM does indeed scale well with increased model size, but also that training longer and across all 25 training datasets evidences a strong benefit.
> 2. A reworked discussion on the the limitations of the DAM, explaining how a future version could take cross-variable (i.e., multivariate) information into account. This *might* be the current limiting factor to the DAM, but there are many avenues requiring exploration in the future.
> 3. We have included an ablation study of all the components of the DAM (now Section 5.2). This indicates which components of the DAM are important, and therefore should be focussed on for future work. We hope that this gives a better insight into what the 'bottlenecks' are for improving performance.
>
> Please let us know if you have any more questions or requests that we can address before the end of the rebuttal period.
>
> Thank you again for your review.

---

> > ### Comment · Reviewer_PhNp · 2023-11-21
> > **DAM as a Foundation Model**
> >
> > Thanks for your updating.
> > Regarding the foundational properties of DAM. I think there are still some aspects that are unclear.
> >
> > 1) How many basis functions do we need? By increasing # of basis functions, how much can we improve its performance across various tasks? The same questions can be applied to the number of datasets (How many dataset do we need? How much improvement can we expect?)
> >
> > 2) For the tasks that DAM underperformed compared to competitive models, why did DAM fail on those task? Is there any domain requirement for DAM to be a foundation model? Or was it due to the lack of scale (data, basis function, parameters, etc..) and is it expected to be solved by DAM by addressing the scale issue?
> >
> > Thanks!

---

> > > ### Author Response · Authors · 2023-11-21
> > > **Addressing unclear aspects of foundational properties**
> > >
> > > Thank you for your reply. Your questions are pertinent and we will provide as much information as possible, here.
> > >
> > > Going from 219 basis functions (previous version) to 437 (current version), and 10 to 25 datasets, improved results across the board. However, we did not see *more relative improvement* across e.g., traffic than ETT datasets, suggesting that the reason for **performance imbalance** lies elsewhere.
> > >
> > > Your focussed questions about model scalability are pertinent, but we believe that the reason for the DAM's relatively worse performance on some datasets lies elsewhere. We think that the true cause lies with the **complexities of training regime and dataset balancing when considering multi-time-series-dataset training**, as explained below.
> > >
> > > # What we think is the bottleneck: the training regime.
> > > We believe that the actual bottleneck to the DAM's performance is *the training regime*. Specifically, how to balance data sampling in this setup (i.e., training on many datasets) is a *challenging open question*. Since the topic of foundation models for time series is new, no existing research has yet asked the fundamental question of: **"how do we train using many diverse time series datasets simultaneously?"**
> > >
> > > We found that trivial sampling (e.g., sampling every data point individually) was sub-optimal. We believe that this is because of a combination of the following properties of times series:
> > > 1. Within a given univariate, samples are non-iid. Dependency across data points is both temporal (e.g., datum at $i+1$ is strongly dependent on datum at $i$) and periodic (e.g., dependence one day/week/hour in the past; datasets like traffic are more strongly periodic than ETT - see Figures 24 and 25).
> > > 2. The length of a univariate is not a good proxy for sampling frequency. A dataset can have many time steps while containing highly periodic and repetitive data, while a short dataset may contain more diversity. Therefore, the **effective utility** gained by training thereon is not necessarily related to dataset length.
> > > 3. Noise level is also variable. A short but noisy dataset may exhibit less repetition but also have low **effective utility**.
> > > 4. Dataset resolution impacts all of the above. For example, a high-resolution dataset (e.g., per-second Huawei datasets; see Appendix H.2.) may have higher or lower effectively utility, depending on the dataset. Note that one of the strong advantages of the DAM is that even for a high-resolution dataset, the HSR can still capture long-term trends (per-second data contains 86400 time steps per day, which is far beyond what regular contexts can see).
> > > 4. Every multivariate dataset may exhibit very different utility across each variable.
> > >
> > > ### A strategy and future work
> > > We explored a number of strategies and converged on one that approximates **effectively utility** of a univariate (see Appendix F for some details). Future work (e.g., a paper like "How to train your DAM...") must and will entail exploring data sampling and/or loss balancing strategies that enable **good performance across all training datasets**. While we do not have sufficient evidence for this owing to resource constraints and scope limitation of this paper, the results in Table 1 suggest that the current sampling strategy (Appendix F) may be under-sampling from datasets that are highly repetitive (such as electricity, traffic, and wind). **Exploring this phenomenon is outside the scope of this paper**.
> > > ___
> > > Solving the multi-dataset time series balancing problem is clearly highly non-trivial. We cannot solve every problem related to foundation models with time series in a single paper, but will endeavour to bring to light and attempt to solve this challenge in future work.
> > > ___
> > > ### A note on Figure 3
> > > Figure 3 shows how the $\mathbf{\theta}_0$ initialisation is capable of representing the signal quite well using 437 basis functions; before training we ensured that these basis functions could represent each dataset sufficiently well. This is further evidence that it is *not the representational capacity* of the DAM that is the bottleneck, but rather the inherently challenging task of balancing training across many time series datasets.
> > >
> > > ### A note on the DAM's underperformance on some datasets
> > > We do not think that the DAM *failed* on the datasets on which it did not win. Please bear in mind that we compare against the absolute SoTA at the time of writing and give each of these methods their best chance possible (e.g., training 120 variants of each method across the 10 datasets, 4 horizons, and 3 seeded runs). Considering existing literature, the DAM outperforms many other methods on these datasets, but we endeavoured to give in the main text only the top results available. Further, analysing the  forecasts on traffic and electricity (Figures 24 (a) and (b)) shows that the DAM is *not necessarily failing*.

---

> > > > ### Comment · Reviewer_PhNp · 2023-11-22
> > > >
> > > > Thanks for your reply.
> > > >
> > > > I think your opinion is sound.
> > > > However, I still think that DAM is just one of the potential candidate methods to become a foundation model but not enough at this time.
> > > > Your explanation of the emerging properties of DAM is weak. It should be defined as functionality of DAM, e.g., DAM can have function 'A' even though we didn't intentionally train it to do that. For me, finding high basis coefficients for very long periods for ETTh1 is just similar to what an ensemble model does, i.e., giving high coefficient values for weak learners even though their performances are not good.
> > > > Additionally, the strategy for further performance improvement is unclear at this time.
> > > > To me, the subject of this paper seems more closely related to generalization ability of the model, which should be different from a foundation model.
> > > > For readers, I presume most will expect foundation model like GPT, i.e., large-scale training, and surprising emerging abilities of the model. However, as you pointed out, there was not much discussion about foundation model in the context of time-series model. To define a foundation model in the time-series domain, we need more discussion from various experts. I hope this paper can become a stepping stone.
> > > >
> > > > I can raise my score if the authors remove 'foundation' term in the paper and replace it to other word that represent 'generalization'

---

> ### Author Response · Authors · 2023-11-22
> **Toward a foundation model?**
>
> Thank you for your comment. Your perspective suggests that you think that the DAM takes a significant step toward being a foundation model, but that we have not yet provided sufficient evidence to convince you *that it is already foundational*.  Would it suffice if we used the term **"towards"** when referring to the DAM in the context of a foundation model?
>
> That said, we propose to change the following parts of the paper:
> 1. The title, from "DAM: A Foundation Model for Forecasting" to "**DAM: Towards A Foundation Model for Time Series Forecasting**". This also better captures the scope of the work. *Please note: changing the title and abstract will not reflect in this openreview setting, but rather only on the paper itself. This will need to be accounted for for the camera ready version, and we will do that.*
> 2. In the introduction, from "We present the deep data-dependant approximate analytical model (DAM) as a foundation
> model for universal forecasting" to "**We present the deep data-dependant approximate analytical model (DAM) as a significant step toward a foundation model for universal forecasting**".
> 3. Also in the introduction, from "To the best of our knowledge, the DAM is the first foundation model for universal time series forecasting" to "**To the best of our knowledge, the DAM is the first model for universal time series forecasting that can be trained simultaneously across many diverse datasets, of different resolutions, and for various horizons, such that it generalises well both within and outwith the training set**".
> 4. In Section 4.2, from "This is strong evidence that the DAM is a foundation model." to "**The DAM generalises well outside of its training set**".
> 5. In the conclusion, from "We presented the DAM – a foundation model for universal forecasting" to "**We presented the DAM as a significant step toward a foundation model for universal forecasting.**".
>
> ___
> **Will that be sufficient for you to raise your score?** If so, we would very much appreciate if you did that before the end of the rebuttal period (a few hours from now). That said, please also consider all of the significant updates we have made to the paper when considering what to change your score to. Specifically, as a step "toward" a foundation model for time series forecasting, we believe that our work on the DAM overcomes many significant practical hurdles in the time series domain (e.g., cross-resolution training, variable horizons, interpretability, etc.), and, as such, deserves serious consideration.
>
> We will update the paper as soon as the changes are made and after we receive your confirmation.

---

> > ### Comment · Reviewer_PhNp · 2023-11-22
> >
> > I changed my score

---

### Official Review · Reviewer_sH5F · 2023-11-03

**Soundness:** 3 good
**Presentation:** 3 good
**Contribution:** 2 fair
**Rating:** 6
**Confidence:** 4

**Summary:**

The paper proposes a foundation model for universal time-series forecasting. Building such a model is challenging because the sample resolution, periodicity, and prediction task are different. To address these challenges, the paper proposes to take randomly sampled histories and output coefficients of basis functions. The designed basis function enables forecasting to non-fixed horizons.
Experiments show that the single model trained on 10 datasets can outperform or match existing SOTA dataset-specific models. It also performs well on held-out datasets and has good interpretability.

**Strengths:**

Building a foundation model for time series forecasting is challenging because different datasets have different patterns and resolutions. This paper proposes reasonable solutions to address these challenges. The experimental results also show that the proposed method is promising. Additionally, it has good interpretability and is robust to missing and irregularly sampled data.

**Weaknesses:**

Training such a foundational model requires high computational cost. The biggest advantage of the foundation model is its ability for zero-shot forecasting. However, the paper only reports the performance on two held-out datasets which is not sufficient. This restricts the practical use of the model in the real-world application.

**Questions:**

How does the training cost of the proposed method compare with the baselines?

---

> ### Author Response · Authors · 2023-11-12
> **Initial response**
>
> Thank you for your insightful review. Please note that the current updated version of the paper has a number of improvements, including using more basis functions (437 vs 219), better results, improved figures, and more training datasets (an additional 15).
>
> ## Regarding zero-shot experiments
> Based on these reviews we have identified that the main weakness of our paper is the limited analysis on the zero-shot setting. You are correct that the main advantage of a foundational model for forecasting is its applicability to unseen datasets. There is another very important advantage, namely that training on many datasets simultaneously improves performance on those datasets, when compared to training on each individually. We postulate that this is owing to a combination of factors, including but not limited to:
> 1. Mitigating the early overfitting that is common when training time series models. Many SoTA methods train for 1-10 epochs, which is only 3125 iterations using a minibatch size of 32 on a training dataset of length 10000. This is 336x shorter than the training of the DAM. We did not witness overfitting when training. See the section below entitled 'Computational cost' for a discussion on the importance of this.
> 2. Learning from patterns in one dataset being useful for generalising to another dataset. For example, weather data from Germany will likely contain patterns that are useful to learn from when generalising to weather data from the US.
>
> That said, we agree that more demonstrations on the zero-shot setting would be pertinent. To this end, we are in the process of analysing zero-shot performance across the following 8 datasets (6 new): Illness, wiki-weekdays, Monash (including ‘temperature rain’, London smart meters, weather, web traffic),  UCI 'individual household electric power consumption' (https://archive.ics.uci.edu/dataset/235/individual+household+electric+power+consumption), and the Azure 2019 function trace.
>
> ### How to assess zero-shot transfer?
> Another reviewer (wavr) suggested that when comparing the DAM to other methods in the zero-shot setting, those comparison methods should be trained on said datasets (i.e., not zero-shot transfer). Given how poorly PatchTST and DLinear transferred to new datasets (Table 2), it is understandable to have a closer comparison. However, the issue is that the DAM is the only method we know of that is designed as a universal forecaster that is dataset agnostic: there are no fully fair comparison methods. We are happy to do what is suggested (even though this advantages other methods quite strongly), but would like to know your opinion regarding how to perform this analysis? What would you, ideally, like to in such an experiment? We are open to discussion and suggestions during this rebuttal period, and will endeavour to make ready an updated paper, with more zero-shot analysis, by the end of the rebuttal period.
>
>
> ## Computational cost
> As mentioned above and in the paper itself, the DAM takes much longer to train. At first this seems to be a disadvantage, but we argue that it signifies a positive shift toward a better match between deep models and time series. Consider other fields, such as NLP or vision: training runs are typically far longer than several thousand iterations. This is simply because the datasets available are far larger than time series datasets. That is one of the primary motivating factors for our development of the DAM; we had to rethink the way time series were processed and predicted in order to design a modelling approach that could extend to any time series dataset. Only then could we actually apply this model to a collection of datasets and move toward training runs comparable to other fields.
>
> Perhaps a pertinent comparison to make is that of training cost of the DAM against the cumulative training cost of other methods. Assume an average training run of 3125 iterations (see above) per dataset. Assume also that we would have to train baseline methods on each horizon-dataset combination. For $25+8=33$ datasets (considering the 25 datasets we used for training and additional 8 for zero-shot transfer) we would need to train $4*33=132$ variants of a baseline model. This is close to the scale of training we used for the DAM, but with the important caveat that baseline methods cannot be trained for longer owing to overfitting, while the DAM benefits from longer training runs by design. Further, these 132 variants do not necessarily transfer well to other datasets.

---

> ### Author Response · Authors · 2023-11-20
> **Rebuttal changes**
>
> Good day.
>
> Please note that we have updated the paper as per your commentary and requests. There is a summary entitled "Rebuttal Summary - After addressing reviewers' comments and questions".
>
> Of particular note for your consideration:
> 1. A considerable extension to the held-out experiments, including an additional 6 datasets (now 8 in total) tested in zero-shot mode, and compared against 3 SoTA methods in zero-shot mode **and when trained on the target datasets**. We also included performance of the DAM when fine-tuned on target datasets for comprehensive comparisons. The DAM beats SoTA methods at zero-shot transfer across 14/16 metrics and **even outperforms SoTA methods trained on the target datasets** in some cases.
> 2. Improved interpretability demonstrations, including a video version submitted as supplementary material.
>
> Please let us know if you have any more questions or requests that we can address before the end of the rebuttal period.
>
> Thank you again for your review.

---

### Official Review · Reviewer_wavr · 2023-11-04

**Soundness:** 3 good
**Presentation:** 2 fair
**Contribution:** 3 good
**Rating:** 6
**Confidence:** 4

**Summary:**

The paper proposes deep data-dependant approximate analytical model (DAM) as a "foundational model" for time series forecasting. DAM uses a long tail distribution to sample from the history of the time series. These irregularly-sampled time-value pairs are fed into a transformer-based model which outputs basis coefficients. The basis coefficients are then used in a basis function composition to generate forecasts. The authors trained a single DAM model across multiple datasets and show that this model is competitive against dataset-specific baselines on long-term forecasting benchmarks. Small-scale analyses have also been conducted on very long-term forecasting and imputation tasks.

**Strengths:**

- The combinations of ideas presented in this work involving history regime sampling, basis composition-based forecasting, and training a single model across multiple datasets are original and interesting.
- The model provides the flexibility to forecast arbitrarily far into the future which is an attractive property. While autoregressive models can already do that, they are generally slow.
- The paper is well-written in general. Some discussion and visualizations can be improved (see weaknesses).

**Weaknesses:**

- The main weakness of this paper is that it overclaims and underdelivers. In its current state, the study is not strong enough to claim the title of a "foundational model".
    - The authors mention that they use datasets from diverse domains. However, out of the 12 datasets studied, 6 come from a single domain. The distribution of sampling frequencies of these datasets are also not diverse with 6 hourly datasets and a limited representation of other frequencies (and some popular frequencies completely missing).
    - Another aspect that could have justified the term "foundational" is a diversity of tasks. However, the paper mostly focuses on the long-term forecasting tasks with limited discussion of other tasks. Importantly, the practically relevant task of short-term forecasting (e.g., Monash time series forecasting archive) gets very less attention.
    - The claim _Most existing forecasting models were designed to process regularly sampled data of fixed length. We argue that this restriction is the central reason for poor generalisation in time series forecasting_ has not been justified convincingly.
- The visualizations are poorly done and confusing for a serious academic paper. Please consider using cleaner figures. It is unclear how exactly inference on a new dataset is performed. It would improve the clarity of the paper if a specific paragraph on inference is added. Please see specific questions in the questions section.
- The results on the long-term forecasting benchmarks, while reasonable, are not impressive for a "foundational model" that has been trained on a larger corpus of datasets.
- The very-long-term forecasting task is of limited practical significance. Despite that, the discussion requires improvement, e.g., by conducting experiments on more datasets and training the baseline models with the "correct" forecast horizon to put the results in a proper context.
- The zero-shot analysis (Sec. 4.3) has only been conducted on two datasets. Moreover, since prior works such as PatchTST and NHITS do not claim to be foundational models, a proper comparison would be with baselines trained specifically on these held-out datasets. DAM would most likely be worse in that case but it would be a better gauge for the zero-shot performance.

**Questions:**

- How exactly is inference performed on a dataset? Is basis function initialization also required during inference?
- How costly is context selection during inference, in general?
- Can you clarify what is meant by "No training of the backbone is even required in this case because the initialisation coefficients are optimal for past data"? Is model training not needed for imputation?
- You mention "The DAM produces relatively high basis coefficients for ETTh2 in the year range, suggesting that it captures the long-term trend." but the ETTh2 dataset only has data over two years (from 2016 to 2018) while relatively high positive values are seen for 3, 5, 7, 9 yr basis components. How do you explain this phenomenon?

---

> ### Author Response · Authors · 2023-11-12
> **Initial response 1/2: A foundational model**
>
> Thank you for your informative review. Since the submission deadline we have worked to improve the paper and have updated it accordingly. A summary of the changes is provided in another comment. Pertinent to your review, and in summary, please note that the new version of the paper uses a model with more basis coefficients (437 vs 219) for improved performance across all datasets, making it a stronger competitor as a foundational model; has improved figures and visualisations to aid in clarity of understanding; and includes an additional 15 **training** datasets from various sources. These additional training datasets cover an additional 2 resolutions (30 minutes and 1 minute), and additional domains, namely air quality, stocks, sunspot index, and cloud resources.
>
> We have broken the rest of our response into two parts (this and another), covering the main weakness you mentioned (this) and other weaknesses and questions (in the next comment).
>
> ## Addressing the main weakness
>
> Our model now uses 25 datasets for training, with 8 held out (including 6 we will add to the paper by the end of the rebuttal deadline). We note that while we could have trained on much larger datasets, many available time-series data have very low signal-to-noise ratios. In our work, we have carefully selected our data to increase the signal in the training and evaluations. In addition, we have used open datasets to improve reproducibility. To the best of our knowledge, we are among very few papers that select datasets from very different domains.
>
> That said, we claim the title of a foundational model because of the novel design of the DAM, not simply because it was trained on many datasets from diverse domains. We believe that  the term 'foundational model' does not necessarily only apply to models trained on massive datasets, at huge cost,  but  that it  is the model itself that **is foundational by design** if it can be used across disciplines. Otherwise,  that claim will necessarily be reserved for big companies with high resources. This is particularly true for time series forecasting because designing a foundational model in this field requires more than just scaling up an existing model to more data. Instead, it required a rethinking of the way data was sampled and ingested to yield a flexible, efficient, and global context (i.e., the HSR and TV-tokens), and a non-fixed horizon forecast (i.e., the basis composition) for multiple use-cases.
>
> ### Dataset-horizon combinations as individual tasks
> In some sense, each individual dataset-horizon combination (in Tables 1 and 2) is a separate 'task' when considering how these are traditionally **solved by training variants of a single model design**. Please consider that the DAM is currently the only SoTA forecasting model that is, *by design*, agnostic to resolution, missing data, forecast horizon, and data-domain. That is the core novelty of this paper and we believe it is hence a foundational model. Please note that the updated version of the paper includes an additional 15 datasets for training and posits the DAM far more clearly. One of our requirements for using a dataset is that it should be sufficiently long. We are open to suggestions for sourcing additional datasets (aside from the 25 we use for training and the new datasets we used for the held out tests).

---

> > ### Comment · Reviewer_wavr · 2023-11-17
> >
> > Thank you for your response and the revision. The updated results look interesting. The figures are much better than the previous (although still dense). Looking forward to the zero shot results.
> >
> > > That said, we claim the title of a foundational model because of the novel design of the DAM, not simply because it was trained on many datasets from diverse domains. We believe that the term 'foundational model' does not necessarily only apply to models trained on massive datasets, at huge cost, but that it is the model itself that is foundational by design if it can be used across disciplines. Otherwise, that claim will necessarily be reserved for big companies with high resources.
> >
> > I never said that a foundation model is only when one trains large models on massive datasets at a _huge cost_. I was commenting on respresentation of the different domains in the training set. I also highlighted the aspect of applicability to a diversity of tasks.
> >
> > > This is particularly true for time series forecasting because designing a foundational model in this field requires more than just scaling up an existing model to more data.
> >
> > This statement hasn't been proven in the literature yet.

---

> > > ### Author Response · Authors · 2023-11-17
> > >
> > > Thank you very much for engaging with us during this review process. We have been working hard to make updates and improvements to the paper and **should** submit another comprehensive update to the paper within 2-3 days and are hoping for additional engagement therethrough. Your comments and perspective have been extremely useful. Thank you.
> > >
> > > ### Regarding foundation models
> > > Adding the additional 15 datasets helps to cover more domains. We also test the DAM on 3 short-term datasets (Illness, Web traffic from Monash, and Temperature Rain from Monash) for held-out transfer and will show the results in an incoming update to the paper (we are waiting on inference to complete for baseline methods on held-out datasets).
> > >
> > > In some sense every dataset-horizon combination can be viewed as a unique task, particularly considering that existing models are trained to specialised on particular combinations. Coupled with very-long-term forecasting, held-out datasets, and imputation, we believe is representativeof variety of diverse tasks. That being said, we are happy to run other experiments that you would suggest that could be considered as unique tasks. At this point we would like to point out that the incoming update to the paper involves a comprehensive update to both the held-out and very-long-term tasks, and provides more comprehensive evidence of the DAM's broad utility.
> > >
> > >
> > > ### Regarding `more than just scaling up'
> > > We agree that there is not literature coverage of the limitations of existing time series models, including those limitations involved in naïve scale-up. While some of the results in our (incoming) held-out experiments provide evidence toward this statement, we hope to prove this claim in future work. Thank you for pointing that out and we will aim to include a short discussion in our final version, at least as part of the appendix if you would agree to this.

---

> ### Author Response · Authors · 2023-11-12
> **Initial response 2/2: addressing remaining weajnesses and questions**
>
> ## Regarding weaknesses
> 1. We have replaced all visualisations with new and improved digital versions.
> 2. Regarding inference, we have made an additional comment, titled ‘ A note on inference and the univariate nature of the DAM’ to address this. The DAM is always univariate, inference **does** involve fitting $\mathbf{\theta}_0$ to the context data. This is because $\mathbf{\theta}_0$ is not part of the model itself, but is instead an input to the model. We will ensure that there is an ‘inference’ section in an upcoming updated version of the paper.
> 3. Regarding very-long-term forecasting: the notion of ‘long’ is dependent on the resolution of the data. For example, 6000 time steps is roughly 41 days (as in Figure 5) for weather, but is 250 days for an hourly dataset and 16 years for a daily dataset. The practical significance for very-long-term forecasting is therefore reliant on dataset resolution. Testing the same sets of horizons amongst datasets of different resolution is for comparison against existing literature. We chose to demonstrate the very-long-term forecasting for weather because (1) longer term forecasting (e.g., 1008 steps=1 week) is useful for weather, and (2) the high resolution of the weather dataset (10 mins) requires longer term forecasting for practical use. We will clarify this in an incoming update to the paper. Regarding an experiment that uses the 'correct' horizon (+- 6000 steps in this case): it is practically very challenging to train conventional models with this horizon because of the way that they forecast (via a vector the length of the horizon). The DAM, on the other hand, does not require assessing all points between `now' and the distant future, making it elegant for the very-long term case. That all said, we can train PatchTST and DLinear using a very-long horizon of 6000 steps and update the paper by the end of the rebuttal period, should you require that. Please let us know if you still think this is necessary.
> 4. Thank you for pointing this weakness out; it is a pertinent deduction. We will ensure that this analysis is sufficiently extended. Specifically, we be analysing zero-shot performance across the following 8 datasets: Illness, wiki-weekdays, Monash (including ‘temperature rain’, London smart meters, weather, web traffic),  UCI 'individual household electric power consumption', and the Azure 2019 function trace. We will keep the results for illness and wiki-weekdays (as in the current version of the paper), but also include results for PatchTST and DLinear when trained on these datasets. Since this is not strictly a fair comparison (i.e., zero-shot DAM against standard baslines), we will endeavour to make that clear in the revised version of the paper. It is likely that we will be removing Figure 1 in favour of additional space for this extension.
>
> ## Regarding questions
> 1. See 1, above.
>
> 2. In the aforementioned comment we show how context selection is as simple as sampling indices (`np.random.choice(np.arange(len(v)), p=phsr, replace=False)`) and is thus extremely lightweight. Your question actually opens up an interesting discussion about how one might alter $p_{hsr}$ according to some intrinsic statistics about the context (which would be more costly) but currently the $p_{hsr}$ is static and independent of the query time series.
>
> 3. Hopefully our clarifications above explained the role of the initialisation coefficients, $\mathbf{\theta}_0$ , and how these are fit to context data (even when data is missing; hence being relevant for imputation). Figure 4 shows the basis initialisation for the past, and how well this fits to the context data. It is *this* that is used for imputation. So, effectively, you are correct: model training is not needed for imputation. Fitting the initial basis coefficients is perhaps an additional contribution of our paper, but there are other more important contributions we chose to focus on.
>
> 4. The high basis coefficients outside of the effective ‘visibility’ of the data is interesting and worth considering. One explanation might be that trends of that scale exist in the data even though the collection procedure could not have captured a full cycle of these trends, and that the DAM is capable of approximating those trends to improve forecasting. There are a number of potential alternative explanations, but it should be noted that this seems like a common behaviour across most datasets (Appendix L).
>
> We believe that the combination of the already updated paper, additional writing and clarifications to be made, and additional zero-shot experiments cover the weaknesses and questions you raised. Please kindly let us know if there is anything else you would like seen done in order to cover your concerns or reticence.

---

> > ### Author Response · Authors · 2023-11-13
> > **Regarding the claim 'Most existing forecasting models were designed to process regularly sampled data of fixed length...'**
> >
> > We will remove this claim based on your comment. We will replace it with the observation that it is extremely challenging to apply existing forecasting models across a range of datasets with different properties because they are not designed to function across resolutions (for example), and that the result is training on very small datasets, which in turn limits generalisation.
> >
> > The justification of this claim is that the DAM achieves SoTA forecasting results when compared to specialised methods.

---

> ### Author Response · Authors · 2023-11-20
> **Rebuttal changes**
>
> Good day.
>
> Please note that we have updated the paper as per your commentary and requests. There is a summary entitled "Rebuttal Summary - After addressing reviewers' comments and questions".
>
> Of particular note for your consideration:
> 1. More diverse training datasets and extensive evaluations on held-out datasets, with improved performance throughout the paper.
> 2. Improved visualisations.
> 3. Description of inference process (Section 3.5).
> 4. A discussion on the high basis coefficients for ETTh2 (Section 5.1).
>
> Please let us know if you have any more questions or requests that we can address before the end of the rebuttal period.
>
> Thank you again for your review.

---

> ### Author Response · Authors · 2023-11-23
> **TOWARDS foundation**
>
> Good day,
>
> Please take note we have heeded your and another reviewer's comments (PhNP) and decided to adjust the title and claim of our paper such that it is "towards foundational".
>
> This and many of the changes made throughout this rebuttal period have been in responsive to your thought-provoking review. Thank you.
>
> **Do you believe that these changes go even a little way towards addressing some of your concerns? An answer from you, prior to the closing of the rebuttal period in a few hours, would be extremely appreciated.** Moreover, if we have satisfactorily addressed your concerns, could you also consider changing your score? We firmly believe that the DAM is good work and deserves attention.
>
> Many thanks once more for all of your hard work in reviewing this paper.

---

> > ### Comment · Reviewer_wavr · 2023-11-23
> >
> > Thank you for all the changes to the paper. The new zero shot comparisons, in particular, look very appealing. I am raising my score to 6. I hope the authors will be able to release their pretrained model and code in the interest of the community at large.

---

### Official Review · Reviewer_yXbk · 2023-11-04

**Soundness:** 3 good
**Presentation:** 3 good
**Contribution:** 3 good
**Rating:** 8
**Confidence:** 4

**Summary:**

This is an interesting paper proposing to solve a very general time series problem. This paper aims at flexible (long or short horizon) time series forecasting with long or short context input for any general time series. Moreover the context could also be irregularly sampled. The paper additionally proposes a strategy for efficient long context inputs.

The model is therefore a general time series forecaster and trained on many time series for generalization to any held out datasets. As such it could be considered a foundational model for time series forecasting.

Experimental results are promising, especially on held out datasets. One of the main advantages of a foundational model is to be general enough to be able to predict on held out datasets. While the results are most positive on held out datasets, the model performs at par with the baselines on the training datasets. To summarize, I believe this to be an interesting contribution to the time series literature, and I am willing to increase my score if the concerns below are addressed.

Update: Increased the score

**Strengths:**

- The problem being solved in the paper is one of the most interesting problems in the time series community i.e. general time series forecasting for flexibly sampled time series.
- The proposed solution (described as a foundational model) involving attention is a befitting solution to this problem by virtue of its ability to adapt to flexible input sizes.
- Experimental results are promising on both training and held out datasets.

**Weaknesses:**

- The model architecture is quite complex and the rationale behind such a choice is not fully explained. One important question that is raised is the usefulness of each of the components in the model. An ablation study may be performed to measure the importances of each of the components proposed in the paper.
- Simple linear forecasting models are not considered as baselines. For comparison between different models a normalized score is preferred rather than a absolute score. Normalization helps interpret the improvements easier since they are scaled. As such, the experimental section need improvement.

**Questions:**

- When using a trained model for forecasting a single time series, how does inference look like in such a simple setting? Do attention models work well in such scenarios? In other words, do attention models produce better forecasts when a larger context in provided? The larger context could be in the form of multiple time series or a larger history.

---

> ### Author Response · Authors · 2023-11-10
> **Initial response to review (prior to making adjustments to paper, if necessary)**
>
> Thank you for your review. Please note that we have uploaded a revised version of the paper and have made a comment describing the changes thus far. In summary, we improved the model by adding more basis functions (437 instead of 219), trained it on 25 datasets (instead of 10), and improved the figures and results in the paper. In the coming days we will be uploading more updated versions of the paper that addresses concerns for all reviewers. That said, we would like to address some of your concerns prior to this, and clarify some points before making the necessary changes to the paper.
>
> Regarding the weaknesses you mentioned:
>  - The model architecture is actually far closer to the standard transformer architecture (from NLP) than many transformer methods in the literature for forecasting. This is mentioned in the 'Model structure' paragraph in Section 3.1., but we have updated the architecture figure into a better digital format. We chose to be explicit about every component instead of hiding details in collective 'blocks' as is often the case for modern architectures.
> That said, we can add an additional paragraph in Section 3.1 to address the intended purposes of MHSA, Cross attention, and the feed forward (cross) block that acts across the basis dimension. The MHSA acts to share information across TV-tokens, and thus build a latent context for the model to adjust basis coefficients. The cross attention acts to make use of this latent context to update basis coefficients for prediction. The cross feed-forward block explicitly models patterns across basis coefficients in order to share information in the frequency domain. After seeing the new Figure 2, please confirm whether you think this will be necessary. ~~Running ablation studies is, unfortunately, not feasible within the rebuttal period because the DAM is expensive to train. Should this be an absolute necessity for you to increase your score, please let us know so that we can figure out how to do this in the time available.~~EDIT: we have performed an ablation study and discussed this in the next comment.
>
>  - DLinear is a linear forecasting model and a well-known strong baseline. Are you suggesting a less capable linear model should be included for comparison? We endeavoured to include results form SoTA methods to highlight the strength of the DAM, testing 14 methods (including Table 6 in Appendix I), instead of testing it against poorer performing baselines. However, we can certainly test a simpler linear method if you insist, and replace a method in Table 1 with these results.
>  - Could you please clarify what you mean by a normalized score? We have listed results in the same way that many papers in this field list them. Normalising the results (by whatever mechanism you suggest/clarify) is certainly possible, but this would also require additional space should we wish to keep unnormalised scores for easy comparison to other papers in the field. We prefer the latter, but are open to a discussion about this.
>
> Regarding your question:
>  - The DAM is always univariate (although we are working on a version that can use information across variables). Inference simply involves sampling time-value pairs from the HSR using the inference time series, fitting initial coefficients to those context points, a forward pass through the DAM, and an assessment of the resultant basis composition for some query time. Figure 10 in Appendix G essentially answers your question regarding the reliance on context size. For the updated model (see the updated paper PDF), a longer context helps in most cases, but this seems to be highly dataset dependent. Thankfully the DAM can be tuned accordingly (HSR tuning) using some held out set (validation in the case of the paper results). It seems that the inference procedure is not immediately obvious (based on question and another), so we will endeavour to make this fully clear in an updated version of the paper.

---

> ### Author Response · Authors · 2023-11-13
> **Ablation study**
>
> We have performed an ablation study as per your recommendation. The results averaged across 4 horizons on the ETT datasets are listed below. We will include this table and a discussion in an update to the paper before the rebuttal period closes. Your suggestion was incredibly useful and enabled an insight into the model functionality that was missing - thank you.
>
> |  | **Nothing** | **Nothing** | **Self-Attn** | **Cross-Attn** | **$FF_{TV}$** | **$FF_{B}$** | **$FF_{B,cross}$** | **ToME** |
> |-------|-------------|-------------|---------------|----------------|-------------|------------|-----------------|----------|
> |       | MSE         | MAE         | MSE           | MAE            | MSE         | MAE        | MSE             | MAE      | MSE   | MAE   | MSE    | MAE   | MSE   | MAE   |
> | ETTh1 | 0.405       | 0.404       | 0.439         | 0.469          | 0.606       | 0.539      | 0.483           | 0.493    | 0.599 | 0.561 | 31.099 | 2.797 | 0.423 | 0.478 |
> | ETTh2 | 0.355       | 0.371       | 0.393         | 0.437          | 0.469       | 0.479      | 0.349           | 0.416    | 0.434 | 0.432 | 13.411 | 1.688 | 0.359 | 0.409 |
> | ETTm1 | 0.352       | 0.385       | 0.411         | 0.439          | 0.642       | 0.528      | 1.354           | 0.498    | 0.706 | 0.531 | 51.323 | 3.519 | 0.364 | 0.425 |
> | ETTm2 | 0.240       | 0.300       | 0.335         | 0.392          | 0.574       | 0.473      | 0.258           | 0.380    | 0.316 | 0.387 | 18.481 | 2.090 | 0.241 | 0.366 |
> ___
> ### Crucial components
> Specifically, it is clear that $FF_{B,cross}$ is the most crucial component. This part of the architecture is a weighted connection that acts across B-tokens to update them at each step (i.e., it is not a transformer or attention mechanism). The second most impactful component is the cross-attention, which suggests that the DAM has learnt to extract additional information from TV-tokens that are useful in updating B-tokens, aside from that information that is captured in $\mathbf{\theta}_0$.
>
> ### The impact of ToME
> Finally, and perhaps most interestingly, removing token merging (ToME) is detrimental to performance, suggesting that a concise collection of TV-tokens is advantageous. This behaviour is the inverse of what was observed by the authors of ToME, although the technique was designed for image transformers.
> ___
> Please note that we were able to perform this ablation study without requiring any additional training. We simply 'switched off' parts of the model to compute these results. This is possible because the latent dimension of the model, $D_{model}$, is consistent throughout the forward pass.

---

> ### Author Response · Authors · 2023-11-20
> **Rebuttal changes**
>
> Good day.
>
> Please note that we have updated the paper as per your commentary and requests. There is a summary entitled "Rebuttal Summary - After addressing reviewers' comments and questions".
>
> Of particular note for your consideration:
> 1. The ablation study (now Section 5.2)
> 2. Normalised MSE and MAE (Appendix I.1.)
> 3. Description of inference process (Section 3.5)
>
> Please let us know if you have any more questions or requests that we can address before the end of the rebuttal period.
>
> Thank you again for your review.

---

> ### Author Response · Authors · 2023-11-23
> **Final change: TOWARDS a foundation model**
>
> Good day,
>
> Please note that we have decided to adopt the term "**toward a foundation model**" throughout the paper. This is based on reviews and discussions with two other reviewers (wavr and PhNp), but worth pointing out to you. See the recent general comment for details of this.
>
> We believe that we have addressed your concerns (particularly regarding the complex architecture and the new ablation study) and have improved the paper in a number of meaningful ways.
>
> **Do you believe that the changes we have made go even a little way towards addressing some of your concerns? An answer from you would be extremely appreciated.** Since there is little over 1 hour left of the rebuttal period, we would very much appreciate it if you decided to alter your score at this stage, given the considerable effort we have put in during this period, and your statement of "I am willing to increase my score if the concerns below are addressed". We believe that the DAM is good work and deserves attention.
>
> Many thanks again for all of your hard work and considerations.

---

### Author Response · Authors · 2023-11-10
**Summary of changes to the paper**

1) We increased the number of basis functions from 219 to 437 and retrained a better model. We altered the text, results, and all figures to reflect these changes. Most notably, the DAM is now winning across 39 of 80 benchmarks (and PatchTST in second place with 28 wins)
2) We sourced, collected, and processed an additional 15 datasets to train the above model. Appendix H2 lists the details of the new datasets.
3) Adjusted Table 1 to count placements across both variants of the DAM.
4) Added additional information in Table 1's caption to list ToME reduction.
5) Improved the HSR tuning (Appendix G) to be finer grained for this new model.
6) Adjusted the results and discussion on long-term forecasting to reflect the new results, highlighting comparisons against PatchTST.
7) Updated the figures into a cleaner digital format.
8) Improved the interpretability demonstration to show self and cross attention for all heads and all layers.
9) Updated hyper parameters in Appendix F.
10) Small rewordings and respacing to be within page limit
11) Minor grammar and spelling fixes.
20) Mentioned the number of heads in the model structure paragraph.
11) Appendix H2 for details of more data.

Additional updates to the paper, addressing concerns and questions for reviewers, will follow in the coming days.

---

### Author Response · Authors · 2023-11-12
**A note on inference and the univariate nature of the DAM**

The DAM is an entirely **univariate model**, even though we tested it on the more challenging multivariate  use-case. PatchTST is also effectively univariate because of its ‘channel independence’. While we list results for multivariate forecasting, these results are computed in a univariate mode, where the predictions on a single variable depend only on the context of that variable. We will make the fully univariate nature of the DAM clear in an incoming updated version of the paper. Further, future work will entail building general multivariate capability into the DAM.

Multiple reviewers pointed out that the inference process was not clear. To this end, we will include a short section discussing how inference is accomplished for new data.

Inference for a given time, $t$, on an input time series, $\mathbf{v}$, where $x$ is the time step (effectively the index), involves the following steps:
1. Use the HSR centred about the queried index ($x$), to define index sample probabilities, $p_{hsr}(x)$. For any missing data, set the probability of sampling at that index to zero and normalise accordingly. Note that $p_{hsr}(x)$ is independent of the series itself.
2. Sample context indices using $p_{hsr}$. Using python and numpy, this is exactly: `np.random.choice(np.arange(len(v)), p=phsr, replace=False)`.
3. Index values from $\mathbf v$ using sampled indices, and compute associated times based on indices and sample resolution (e.g., an index of -3 at a sample resolution of 10 minutes will result in a time of -30 minutes), setting the current time to zero. Ensure time units are in days.
4. Compute initial basis coefficients, $\mathbf{\theta}_0$ from context. Note that this is an extremely lightweight operation (given only 437 basis functions) and is necessary in order to initialise B-tokens. The strength of the DAM is that it produces coefficients on a case-by-case basis. $\mathbf{\theta}_0$ is not part of the model itself and is not learned via gradient descent; instead, it is an input to the model.
5. Forward pass through the DAM to produce updated coefficients, $\mathbf{\theta}$.
6. For any query/prediction time (back or forward in time), assess resultant basis composition.

---

### Author Response · Authors · 2023-11-20
**Rebuttal Summary - After addressing reviewers' comments and questions**

We have made a number of important and comprehensive changes. While we have responded to each reviewers' comments individually, we will summarise here.

## Is the DAM foundational?
Some of the reviewers have questioned whether our experiments were sufficient evidence that the DAM is foundational. To this end, we have made extensive updates to the paper, positing the DAM more clearly as a foundation model. In summary, these experiments are:
1. We have increased the number of training datasets to 25, including an additional 2 domains and 2 resolutions, improving performance throughout.
2. Every dataset-horizon combination **can be considered a unique task**, given that existing methods require models suited to each combination. Considering evaluations in Tables 1,2, and 3, we have demonstrated the DAM's performance across 10x4 (long-term) + 1 (very-long-term) + 8x4 (held-out) + 7 (imputation) =80 tasks in total.
3. Regarding held-out experiments:
    - We now evaluate across 8 datasets, including 3 of which that require short-term forecasting (Illness, Temperature Rain, and Web traffic; reviewer wavr).
    - We compare against 3 SoTA baselines in zero-shot mode and **when these are trained on the transfer datasets**. In some cases the DAM even outperforms baselines trained from scratch, and is far superior to these models in zero-shot mode (winning across 14/16 evaluations).
    - We included a fine-tuned variant of the DAM for comprehensive comparison.

We have altered the text throughout to account for the new and superior performance owing to updates to the model and experiments, making the advantages and strengths of the DAM clear.
___

## Experimental analysis
1. Trained a new model using 437 basis functions and more data, improving performance across all experiments.
2. Section 4.1 discussion now clarifes the DAM's superior performance (39/80 $1^{\text{sts}}$) compared to PatchTST (28/80 $1^{\text{sts}}$); this section now relies less on appendices (as per reviewer PhNp's comment).
3. Extensive zero-shot experiments (see above).
4. Improved very-long-term forecasting experiment:
    - Clarified the practical significance on the Weather dataset.
    - Computed results across 9 datasets, with baselines trained for those horizons.
    - Recomputed results using new and improved DAM.
5. Clarified the trend we observe for flexible inference cost (reviewer PhNP).
6. Included an ablation study of architecture components (reviewer yXbk).
7. Improved interpretability analysis, using both self- and cross-attention for each head, and discussed the high coefficients at low frequencies (reviewer wavr). We also combined Sections 5.2 and 5.3.
8. New Appendix N for reviewer PhNP, demonstrating initial experiments on scaling laws.
___
## Presentation
1. We redid all figures to a cleaner digital format and removed Figure 1 (for space). The backbone figure provides better clarity of the method. The HSR figure now shows far clearer how the DAM's context, while the same size as regular context, is able to provide a far more global perspective of the underlying signal. We also included only 1 example in Figure 7 owing to space.
2. Updated Section 3.4 with new training settings.
3. Added Section 3.5 to describe the inference processes since reviewers yXbk, wavr and PhNp were unsure how inference was accomplished. We have also included an additional comment during this rebuttal to describe this in detail, and clarified in the paper that the DAM is always univariate (even though it is assessed in the more challenging multivariate setting).
4. Reordered Section 5 for better flow.
5. We included normalised MSE and MAE results in an additional appendix (I.1) because of reviewer yXbk's request. We chose to keep the standard scores in the paper because (1) this is how other SoTA methods present results and (2) the datasets on which results are computed are already normalised (using training dataset statistics) as is the norm in this field.
6. Reworked the limitations paragraph, removing the discussion on shorter horizons because the current version of the DAM does better, and noting that the DAM may benefit from multivariate information.
7. Extended Appendix 8 for details of new training and held-out datasets.
8. Minor grammar, spelling, and ordering changes (for space).
___
## Supplementary video
We have included a video version of Figure 5 (tracking quantities as time progresses) as additional supplementary material, because we found that a dynamic version of this demonstrates more clearly how the DAM pays attention.

---

### Author Response · Authors · 2023-11-23
**DAM: TOWARDS a foundation model for time series forecasting**

Dear reviewers.

Please note that we have decided to adopt the term "**towards a foundation model**" throughout the paper. Based on reviews and discussions with some of you (notably **wavr** and **PhNp**), it has become clear that our claim that the DAM is already a foundation model is not fully realised by the experiments in the paper. This is due, in part, to disagreements regarding nomenclature and what is required for a model to be considered foundational. Providing incontestable evidence that a model is indeed foundational is extremely difficult because of this.

Given that the DAM is the first of its kind - a single model that performs remarkably well across many datasets and particularly on held-out datasets, designed to *ingest cheaply data from the distant past* and forecast efficiently to *arbitrary horizons*, and that is naturally *interpretable* - we believe that it is a significant and novel contribution to time series forecasting. Further, we hope that the idea of working with **random histories** (viz. the HSR) that offer a cheap global perspective of the temporal dynamics, as opposed to fixed-length contexts, inspires practitioners in this space.

That all said, by using the term "**towards foundation...**" (reworking several instances throughout the paper; see the updated version and latest comment to reviewer PhNp) we thus reduce our claim to that of a significant step in the right direction. We ask you please to reconsider your scores and reviews accordingly. Thank you again for all of your time and considerations.
___
Please Note: changing the title and abstract will not reflect in this openreview setting, but rather only on the paper itself. This will need to be accounted for for the camera ready version, and we will do that.

---

### Meta-Review · Area_Chair_ycYS · 2023-12-08

**Metareview:**

The paper presents the DAM foundation-model architecture for time series forecasting, and evaluate it in both a multi-task setting and a zero-shot setting. The  reviewers noted the model's interesting design choices such as the use of history-sampling-regime  and basis decomposition functions, which enable it to adapt to varying context and horizon lengths. The evaluation results and datasets used in the paper were significantly enhanced after the discussion phase, and the results (especially zero-shot) now look convincing, Overall, this is a nice contribution in the very timely problem space of foundation models for forecasting

**Justification For Why Not Higher Score:**

-The evaluation results in the paper is performed on a  much smaller scale of datasets (both training and evaluation) than desirable, and the main results in the multi-task (non zero-shot) regime do not really show much gains over the existing state-of-the-art methods (PatchTST)

**Justification For Why Not Lower Score:**

The paper has some novel design elements (HSR and continuous basis functions for forecasting) which enable it to adapt to varying/irregular context and horizon lengths. The new evaluation results after the discussion period seem comprehensive and thorough, and this paper makes nice empirical progress towards the design of large scale foundation models in time series.

---

### Decision · Program_Chairs · 2024-01-16

Accept (poster)